# Decoupling the Class Label and the Target Concept in Machine Unlearning

## Abstract

Machine unlearning as an emerging research topic for data regulations, aims to adjust a trained model to approximate a retrained one that excludes a portion of training data. Previous studies showed that class-wise unlearning is effective in forgetting the knowledge of a training class, either through gradient ascent on the forgetting data or fine-tuning with the remaining data. However, while these methods are useful, they are insufficient as the class label and the target concept are often considered to coincide. In this work, we expand the scope by considering the label domain mismatch and investigate three problems beyond the conventional *all matched* forgetting, e.g., *target mismatch*, *model mismatch*, and *data mismatch* forgetting. We systematically analyze the new challenges in restrictively forgetting the target concept and also reveal crucial forgetting dynamics in the representation level to realize these tasks. Based on that, we propose a general framework, namely, *TARget-aware Forgetting* (TARF). It enables the additional tasks to actively forget the target concept while maintaining the rest part, by simultaneously conducting annealed gradient ascent on the forgetting data and selected gradient descent on the hard-to-affect remaining data. Empirically, various experiments under our newly introduced settings are conducted to demonstrate the effectiveness of our TARF.

## 1 Introduction

In response to data regulations [26, 49], machine unlearning [5, 52] has emerged to eliminate the influence of training data from a trained model [59]. The intuitive goal is to forget the specific data as if the model had never used it during training [4]. To achieve that, a direct way [52] is to retrain the model from scratch by excluding the data to be unlearned, termed *exact unlearning*. Considering the intensive computational cost, much attention has been paid to *approximate unlearning* [17, 29, 6, 11], which adjusts the trained model for approximating the behaviors of the retrained one. Prior works [29, 55] tried to conduct random-wise forgetting to unlearn uniformly sampled instances, while it conflicts with the goal of model generalization and is philosophically infeasible [51]; and feature-wise forgetting [58] to revoke sensitive features, but is limited to tabular data [46]. Focusing target granularity as semantic clusters, recent studies [37, 30, 6, 11] showed that *class-wise* unlearning is effective in forgetting the knowledge of a training class, either through reverse optimization [54, 29] on the class data or fine-tuning on the remaining data [17] to realize catastrophic forgetting [2, 31].

Despite the promising achievements, the previously studied scenario [58, 17, 6, 30, 11] mainly assumed the target concept to coincide with the class label, overlooking that the practical unlearning request [3, 23, 34] may violate the taxonomy of the pre-training tasks. Raised by the model users, the reported cases to be unlearned can involve different concerns from original tasks, spanning from privacy, fairness, copyright, or the hazardous capabilities [39]. The conventional matched scenario is that all the identified forgetting data correspond to one pre-training class. However, those fully identified cases may be only a semantic subset within a class, for which the model developer needs to unlearn the small set considering reserving model utility on the other parts. Nevertheless, sometimes the user would identify limited cases of the target concept. With a conservative attitude for protecting the reputation of serving [3, 39] (e.g., IP conflicts), the developer tends to unlearn a larger semantic cluster when those instances are from the same class or, more critically, across different classes.

In this work, we decouple the target concept with the class label, to model the unlearning scenarios for research explorations. To be specific, we consider the different label domains of the forgetting

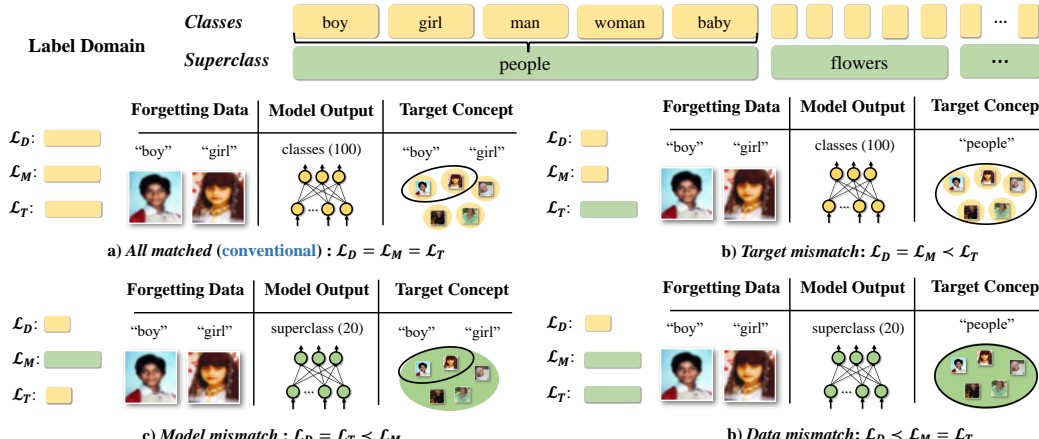

Taking the *CIFAR-100* [35] dataset with its classes and superclass information, we instantiate four unlearning tasks given the same forgetting data with the class labels of "boy" and "girl": a) *all matched forgetting* (conventional scenario): unlearn "boy" and "girl" with the model trained on the classes; b) *target mismatch forgetting*: unlearn "people" with the model trained on the classes; c) *model mismatch forgetting*: unlearn "boy" and "girl" with the model trained on the superclass; d) *data mismatch forgetting*: unlearn "people" with the model trained on the superclass.

Figure 1: Illustrations of decoupling the class label and the target concept.

data $\mathcal{L}_D$, the model output $\mathcal{L}_M$, and the target concept $\mathcal{L}_T$ in unlearning. We introduce two relations between two label domains, i.e., $\mathcal{L}_1$ matches $\mathcal{L}_2$ ($\mathcal{L}_1 = \mathcal{L}_2$) and $\mathcal{L}_1$ is the subclass domain of $\mathcal{L}_2$ ($\mathcal{L}_1 \prec \mathcal{L}_2$)[1] , then identify scenarios corresponding to the target concept being larger or smaller than the class unit. Assuming that the reported forgetting data should be included in the target concept, e.g., $\mathcal{L}_D \preceq \mathcal{L}_T$, we have *all matched* $\mathcal{L}_D = \mathcal{L}_T = \mathcal{L}_M$; *target mismatch* $\mathcal{L}_D = \mathcal{L}_M \prec \mathcal{L}_T$; *model mismatch* $\mathcal{L}_D = \mathcal{L}_T \prec \mathcal{L}_M$; and *data mismatch* $\mathcal{L}_D \prec \mathcal{L}_T = \mathcal{L}_M$. We further illustrated in Figure 1 using the CIFAR-100 [35] dataset to instantiate four unlearning tasks with the classes and superclass.

Given the aforementioned tasks, we identify new challenges with the mismatched label domains (refer to Figure 2). Unlike the accurate unlearning approximation in the conventional all matched task [17, 30, 6], the representative unlearning methods [58, 54] exhibit different performance gap with the retrained reference in the other tasks. Specifically, the under-entangled feature representation (when $\mathcal{L}_M \prec \mathcal{L}_T$) or the under-representative forgetting data (when $\mathcal{L}_D \prec \mathcal{L}_T$) results in insufficient forgetting, while the entangled feature representation (when $\mathcal{L}_T \preceq \mathcal{L}_M$) prevents the decomposition of target concept with the retaining part. The former requires target identification in the remaining dataset, while the latter requires explicit target separation over the entangled feature representation. Through exploration of forgetting dynamics (refer to Figure 3), we demonstrate the feature distance reflected by representation gravity is a crucial factor for the feasibility of these unlearning tasks.

Based on the above analysis, we propose a novel framework, namely, *TARget-aware Forgetting* (TARF), for unlearning. In general, we consider two parts (refer to Eq. 5), i.e., annealed forgetting and target-aware retaining, which collaboratively enable the target identification and separation for these forgetting tasks. Specifically, the algorithmic framework (refer to Figure 4) incorporates an annealed gradient ascent and target-aware gradient descent in a dynamical manner, which can be viewed as three phases. The first actively unlearns the identified forgetting data, and constructs the contrast information to filter out the remaining data which is hard to be affected. Then, simultaneously learning the selected retaining data with gradient descent deconstructs the entangled feature representation. Ultimately, the learning objective can progressively approach standard retraining using the aligned retaining data (refer to Figure 5). We present comprehensive experiments of four unlearning tasks across different datasets to demonstrate the performance of TARF, and show its application on concept forgetting with stable diffusion. The main contributions of our work can be summarized as,

- Conceptually, we introduce new settings that decouple the class label and the target concept, which investigate the label domain mismatch in class-wise unlearning (in Section 3.1).

- Empirically, we systematically reveal the challenges of restrictive unlearning with the mismatched label domains, and demonstrate that the representation gravity in forgetting dynamics is critical for achieving the forgetting target in the new tasks (in Section 3.2).

---

[1]$\mathcal{L}_1 \prec \mathcal{L}_2$: For any label $y \in \mathcal{L}_1$, there exist label $y' \in \mathcal{L}_2$ that instance being labeled with $y$ can also being labeled with $y'$, but not all instance being labeled with $y'$ can be labeled with $y$.

- Technically, we propose a general framework, namely, *TARF*, to realize the target identification and separation in unlearning. It consists of annealed forgetting and target-aware retaining which collaboratively approximate retraining on the retaining data (in Section 3.3).

- Experimentally, we conduct extensive explorations to validate the effectiveness of our framework and perform various ablations to characterize algorithm properties (in Section 4).

## 2 PRELIMINARIES

In this section, we briefly introduce the problem settings of class-wise machine unlearning, and compare the differences between ours and the conventional setting considered in the previous research works. More details about the unlearning baselines considered in our work can refer to Appendix C.1.

**Problem setup.** Following the literature [52, 59], we mainly consider the multi-class classification [18] as the original training task for class-wise unlearning. Let $\mathcal{X} \subset \mathbb{R}^d$ denote the input space and $\mathcal{Y} = \{1, \ldots, C\}$ denote the label space, where $C$ is the number of classes, the training dataset $\mathcal{D} = \{(x_i, y_i)\}_{i=1}^N$ consists of two subsets in machine unlearning, e.g., the forgetting dataset $\mathcal{D}_f$ and the retaining dataset $\mathcal{D}_r = \mathcal{D} \backslash \mathcal{D}_f$. Building upon the model $f_{\theta^*} : \mathcal{X} \to \mathcal{Y}$ trained on $\mathcal{D}$ with the loss function $\ell$, the general goal of this problem is to find an unlearned model $\theta_{un}^*$, which approximates the behaviors of the model $\theta^r$ that retrained on $\mathcal{D}_r$ from scratch,

$$\theta_{un}^* = \arg\min_\theta \frac{1}{|\mathcal{D}|} \sum_{(x,y)\sim\mathcal{D}} \mathcal{R}(\theta, \theta^r, x, y) \quad \text{s.t.} \quad \theta^r = \arg\min_\theta \underbrace{\frac{1}{|\mathcal{D}_r|} \sum_{(x,y)\sim\mathcal{D}_r} \ell(f_\theta(x), y)}_{L_{\text{retrain}}}, \quad (1)$$

where $\mathcal{R}$ indicates a general risk measure for model behavior consistency [17, 52], which can be instantiated by comprehensive evaluation metrics [30, 11] (e.g., unlearning accuracy (UA), retaining accuracy (RA), and others related to privacy) in experiments to pursue the unlearning efficacy and the model utility [59]. The specific definition of evaluation metrics can be referred to in Section 4.1.

**Dataset partition in mismatched setting.** As the target concept is decoupled from the class label, we adopt $\mathcal{D}_t$ to indicate the dataset of the target concept, $\mathcal{D}_f$ to indicate the *given forgetting* dataset, and summarize the scenarios in Table 1. We can find that the previous assumptions of $\mathcal{D}_f = \mathcal{D}_t$ and $\mathcal{D}_r = \mathcal{D} \backslash \mathcal{D}_f$ only hold in all matched setting. In model mismatch forgetting, the former is still

Table 1: Training set data partition.

| Scenario | Data Partition | |
|---|---|---|
| All matched | $\mathcal{D}_f = \mathcal{D}_t$ | $\mathcal{D}_{un} = \mathcal{D}_r$ |
| Target mismatch | $\mathcal{D}_f \subset \mathcal{D}_t$ | $\mathcal{D}_{un} = \mathcal{D}_{fr} \cup \mathcal{D}_r$ |
| Model mismatch | $\mathcal{D}_f = \mathcal{D}_t$ | $\mathcal{D}_{un} = \mathcal{D}_r$ |
| Data mismatch | $\mathcal{D}_f \subset \mathcal{D}_t$ | $\mathcal{D}_{un} = \mathcal{D}_{fr} \cup \mathcal{D}_r$ |

held while we notice that there exists *affected retaining* data (like the left of Figure 2) in $\mathcal{D}_{ar}$ having the same class label with that in $\mathcal{D}_f$; in target/data mismatch forgetting, $\mathcal{D}_f \subseteq \mathcal{D}_t$ and the remaining dataset $\mathcal{D}_{un} = \mathcal{D} \backslash \mathcal{D}_f$ include both true retaining dataset $\mathcal{D}_r \subseteq \mathcal{D}_{un}$ and the *false retaining* dataset (like the right of Figure 2) $\mathcal{D}_{fr} = \mathcal{D}_t \backslash \mathcal{D}_f$, where the data belong to the target concept but have not been unidentified. Considering task feasibility, we assume that the number of classes in $\mathcal{D}_{un}$ belonging to the target concept is known in target mismatch forgetting, and the retrained models are all trained using $\mathcal{D}_r = \mathcal{D} \backslash \mathcal{D}_t$. Details of scenario construction and class information can refer to Appendix D.4.

**Different focus from prior methods.** Existing studies [30, 6] generally assume that $\mathcal{D}_f = \mathcal{D}_t$ and $\mathcal{D}_r = \mathcal{D} \backslash \mathcal{D}_f$. The common approximation unlearning methods either focus on retaining or forgetting objectives. The former, represented by Fine-tuning (FT) [58], fine-tunes the model $\theta^o$ on $\mathcal{D}_r$ to induce catastrophic forgetting over $\mathcal{D}_f$. Later advances assign random labels [17] on $\mathcal{D}_f$ to enforce forgetting or adopt $L_1$-norm [30] to infuse weight sparsity in approximation. The latter, represented by gradient ascent (GA), reverse gradient updates on $\mathcal{D}_f$. And another line of works [29] utilizes the influence function [33] to erase the influence. More recently, adversarial perturbation [6] on $\mathcal{D}_f$ is employed to shrink the decision boundary for the target class. From a different perspective, we explore the label domain mismatch that relaxes the previous assumption. More discussions are in Appendixes B and C.

## 3 TARF: *TARget-aware Forgetting*

In this section, we first introduce the motivation and reveal the challenges of restrictive unlearning when the class label and the target concept are decoupled (Section 3.1). Second, we present systematic

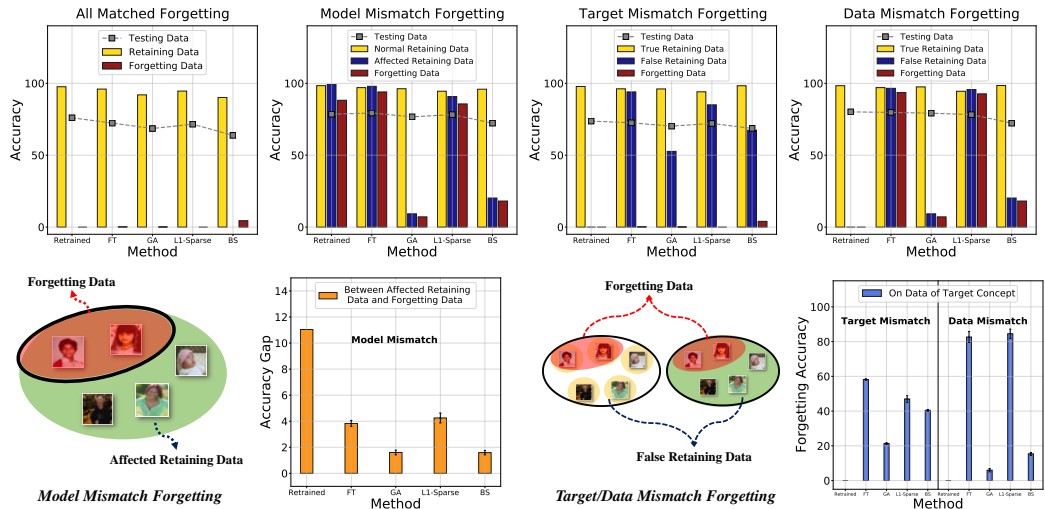

We conduct various unlearning methods for the four tasks. In conventional all-matched forgetting, all the methods can perform similarly to Retrained. In contrast, we can find that model mismatch forgetting can be affected by the trained model, coupling the behaviors on the forgetting and affected retaining (under the same superclass) data, leaving less accuracy gap between them. In target or data mismatch forgetting, the class labels can not fully represent the target concepts, leaving the non-zero accuracy on the false retaining data (belongs to the target concept).

Figure 2: The challenges of restrictive unlearning with the mismatched label domains.

exploration through the perspectives of feature representation and forgetting dynamics (Section 3.2). Lastly, we propose our novel framework, i.e., *TARget-aware Forgetting* (TARF) (Section 3.3).

### 3.1 MOTIVATION: EXPLORING MISMATCHED TAXONOMY IN UNLEARNING

Given its technical nature of mitigating the data influence from a trained model, unlearning is given broader significance in the context of trustworthiness [3], where the requests can be varied beyond the withdrawal from data owner [51], and may be applied in mitigating bias [60] to improve fairness, erasing harmful content [39] to ensure safety usage, or removing inappropriate content [14] for social good. Recently, a series of studies [17, 58, 30, 11, 6] have several proposals on forgetting a training class of the models, and demonstrated it can be successfully achieved by partially scrubbing the class data or fine-tuning on the retaining data to realize catastrophic forgetting [13, 19]. However, a general scenario considered in previous works is that the target concept is aligned with the taxonomy of the pre-training tasks, which may not always hold in practical scenarios with the previous meanings (we leave more discussion in Appendix D.5). This naturally motivates the following research question,

*What if the class labels and target concept do not coincide in unlearning?*

To investigate it, we consider different label domains on $\mathcal{L}_D$ of forgetting data, $\mathcal{L}_M$ of model outputs, and $\mathcal{L}_T$ of target concepts. Assuming that $\mathcal{L}_D \preceq \mathcal{L}_T$, we have either $\mathcal{L}_D = \mathcal{L}_T$ or $\mathcal{L}_D \prec \mathcal{L}_T$. When $\mathcal{L}_D = \mathcal{L}_T$, we have all matched if $\mathcal{L}_D = \mathcal{L}_M$ (e.g., forgetting "boy" and "girl" with the model trained on classes) and model mismatch if $\mathcal{L}_T \prec \mathcal{L}_M$ (e.g., forgetting "boy" and "girl" with the model trained on superclass); When $\mathcal{L}_D \prec \mathcal{L}_T$, we have target mismatch if $\mathcal{L}_D = \mathcal{L}_M$ (e.g., forgetting "people" with the model trained on classes given "boy" and "girl") and data mismatch if $\mathcal{L}_T = \mathcal{L}_M$[2] (e.g., forgetting "people" with the model trained on superclass given "boy" and "girl").

In Figure 2, we conduct the unlearning on the four forgetting tasks as instantiated in Figure 1. As a result, those unlearning methods, e.g., the representative methods FT, GA, and the recent $L_1$-sparse [30] and BS [6] show different performance gaps compared with the retrained models except in the conventional all matched setting. It can be found that the *false retaining* data (which belong to the target concept but are not identified) are under-represented by the given forgetting data when $\mathcal{L}_D \prec \mathcal{L}_T$, as demonstrated by the non-zero accuracy on target concept in the right-bottom of Figure 2; and the *affected retaining* data (which is under the same superclass as the model trained

---

[2]Note that we also provide a detailed discussion in appendix D for all the potential cases, while some (e.g., $\mathcal{L}_T \prec \mathcal{L}_D$) are impractical and some (e.g., $\mathcal{L}_M \prec \mathcal{L}_T$) are similar to the major scenarios considered here.

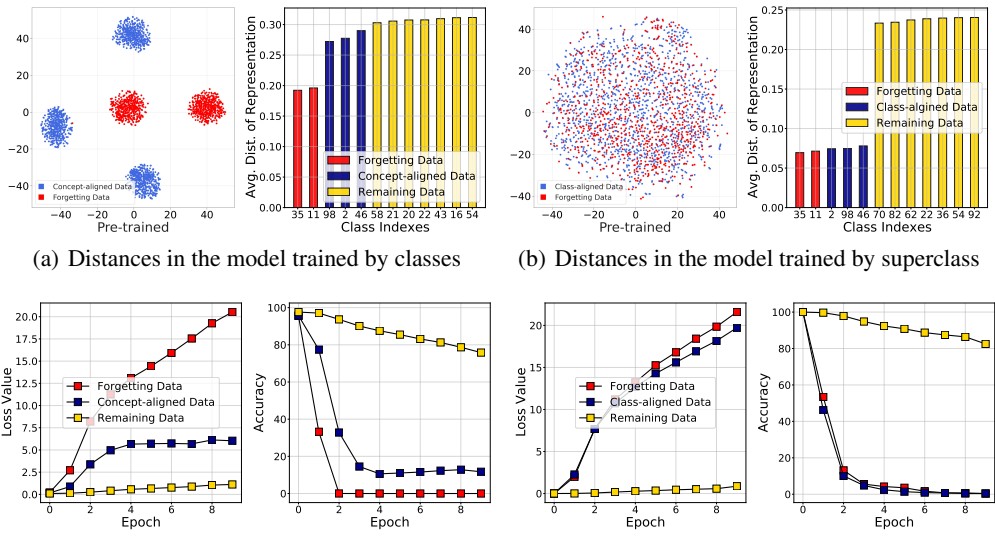

(a) Distances in the model trained by classes   (b) Distances in the model trained by superclass

(c) Forgetting with under-entangled representation   (d) Forgetting with entangled representation

We present the tSNE visualization of the learned features from the pre-trained model trained by (a) classes and (b) superclass, and their averaged Euclidean distance to cluster centers of the forgetting data. At the bottom, we show the averaged loss/accuracy value of forgetting data, concept/class-aligned data, and the remaining data during GA on the two representations. Note that we only show the 5 classes in tSNE due to the large number of remaining classes, we also provide the full results of the unlearned representations in Appendix F.2.

Figure 3: Forgetting dynamics on entangled/under-entangled feature representations.

on) are entangled with the forgetting part when $\mathcal{L}_T \prec \mathcal{L}_M$, as demonstrated by the less accuracy gap between forgetting and affected retaining data than that of Retrained in the left-bottom of Figure 2.

## 3.2 SYSTEMATIC EXPLORATION ON FORGETTING DYNAMICS

The mismatch of label domains affects the construction of model representation in unlearning, which requires us to explore it further to understand the underlying mechanism of the performance gaps. In Figure 3, we take a closer look at the relation between the representation and forgetting dynamics, for which we use the analogy "gravity", originating in physics, as an intuitive inspiration for describing the mutual influence on forgetting data and the other data, specializing in the following ability of unlearning behaviors with the representation similarity, similar to the mutual attraction with objects.

**Target or data mismatch.**   In both target and data mismatch forgetting, we have $\mathcal{L}_D \prec \mathcal{L}_T$, which means the forgetting data is a subset of the target concept, i.e., $\mathcal{D}_f \subset \mathcal{D}_t$. As indicated in Figures 3(a) and 3(c), partially relying on the forgetting or remaining data can not fully represent the target concept, and leaves non-zero accuracy on false retaining data. We can summarize the following observation,

**Observation 3.1** (Insufficient representation). Given $\mathcal{L}_D \prec \mathcal{L}_T$ that indicates $\mathcal{D}_f \subset \mathcal{D}_t$, and a cluster center $h^*$ of feature representations $h_{x \sim \mathcal{D}_f}(x)$ extracted at the pre-trained model $\theta$, as well as a distance measure $d(\cdot, \cdot)$, the sample $(x^u, y^u) \sim (\mathcal{D}_t \backslash \mathcal{D}_f)$ exhibits weak gravity following the sample $(x, y) \sim \mathcal{D}_f$ on the forgetting dynamic $\epsilon = \mathbb{E}(\ell(f_\theta(x), y) - \ell(f_{\theta^t}(x), y))$ with a large value $\zeta_1$, under the observation interval $t$ from $\theta$ to the unlearned model $\theta^t$,

$$(d(h(x^u), h^*) > \sup_{x \sim \mathcal{D}_f} d(h(x), h^*)) \Rightarrow |(\ell(f_\theta(x^u), y^u) - \ell(f_{\theta^t}(x^u), y^u)) - \epsilon| > \zeta_1. \quad (2)$$

**Model mismatch forgetting.**   In this task, we have $\mathcal{L}_D = \mathcal{L}_T$ while $\mathcal{L}_T \prec \mathcal{L}_M$. Regarding the model trained by the superclass, it can be found in Figures 3(b) and 3(d) that the features of forgetting data and affected retaining data are closely entangled, showing that the unlearning of the forgetting data can unavoidably affect the representation of the other part. In contrast, it is also notable in the left-bottom of Figure 2 that the accuracy gap between forgetting data and affected retaining data is expected to be large in the retrained reference. We summarize the observation as follows,

**Observation 3.2** (Decomposition lacking). Given $\mathcal{L}_T \prec \mathcal{L}_M$ that indicates the broader representation region for $\mathcal{D}_z := \mathcal{D}_{ar} \cup \mathcal{D}_t$ within the same class $z$, where the sample $x^m \sim (\mathcal{D}_z \backslash \mathcal{D}_t)$ exhibits strong

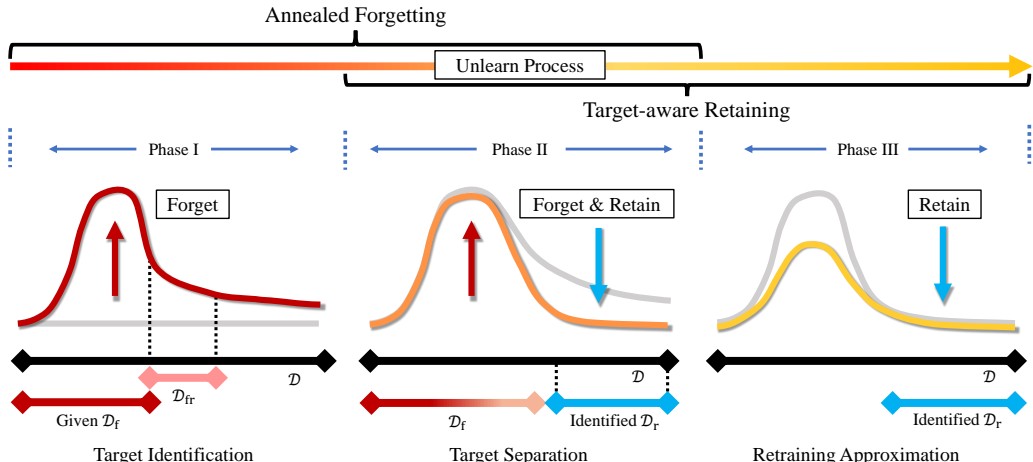

The overall framework consists of two objective parts, e.g., annealed forgetting and target-aware retaining, which can be regarded as three phases to enable all the class-wise unlearning tasks through the view of the unlearning process. (a) Phase I utilizes the gradient ascent to construct dynamic information for all class data; (b) Phase II simultaneously considers gradient ascent on forgetting data and gradient decent on remaining data that is hard to affect to separate target concept; (c) Phase III conducts gradient descent on the selected data to approximate the retraining.

Figure 4: Overview of the proposed framework TARF.

gravity following the forgetting dynamic $\epsilon = \mathbb{E}(\ell(f_\theta(x), y) - \ell(f_{\theta^t}(x), y))$ with a small value $\zeta_2$,

$$(d(h(x^m), h^*) \leq \sup_{x \sim \mathcal{D}_z} d(h(x), h^*)) \Rightarrow |(\ell(f_\theta(x^m), y^m) - \ell(f_{\theta^t}(x^m), y^m)) - \epsilon| < \zeta_2. \quad (3)$$

**Forgetting dynamics with representation distance.** Despite the issues revealed by previous observations under label domain mismatch, the forgetting performance varying obviously on different representations also provides clues on addressing them. Notably, we can find that GA achieves better forgetting efficacy on the data mismatch forgetting as the feature representation of the forgetting data and false retaining data is entangled. Through the effect of actively forgetting the given data on the other parts of data, we have the following that reveals a crucial factor for achieving these tasks,

**Definition 3.3** (Representation gravity). Given the empirically demonstrated Observation 3.1 and Observation 3.2, we can have an indicator $I_{con}(x, y, \theta)$ to reflect the representation similarity $d(h(x), h^*)$ in the model $\theta^t$, which is a crucial factor for the feasibility of unlearning under mismatch,

$$I_{con}(x, y, \theta) = |\ell(f_\theta(x), y) - \ell(f_{\theta^t}(x), y)|. \quad (4)$$

It is empirically demonstrated in Figure 3 where we present the average representation distance to the cluster center of forgetting data, the corresponding changes in accuracy and loss values show that the smaller the distance in representation level, the similar forgetting dynamics the model would have on prediction. As the Observations 3.1 and 3.2 reveal the issues of insufficient representation and decomposition missing, we can utilize the representation gravity to identify the other unidentified forgetting data in the remaining set, and reveal the needs of deconstructing the pre-entangled representation by simultaneously considering the forgetting and retaining objectives.

### 3.3 ALGORITHM FRAMEWORK OF TARF

Based on the previous analysis, we introduce the whole framework of *TARget-aware Forgetting* (TARF), to enable the four class-wise unlearning tasks. Given the identified forgetting data, we illustrate the overall process in Figure 4, and introduce its dynamic learning objective as follows:

$$L_{TARF} = \underbrace{k(t) \cdot \left( -\frac{1}{|\mathcal{D}_f|} \sum_{(x,y) \sim \mathcal{D}_f} \ell(f(x), y) \right)}_{\textbf{Annealed Forgetting } L_f(k)} + \underbrace{\frac{1}{|\mathcal{D}_{un}|} \sum_{(x,y) \sim \mathcal{D}_{un}} \ell(f(x), y) \cdot \tau(x, y, t)}_{\textbf{Target-aware Retaining } L_u(\tau)}, \quad (5)$$

where $k(t)$ serves as an annealing strategy to control the strength of the forgetting part. Along with training, we expect the overall objective to approximate the retraining ones $L_{TARF} \rightarrow L_{retrain}$ through,

$$L_f(k) \rightarrow 0, \quad L_u(\tau) \rightarrow L_{retrain}, \quad (6)$$

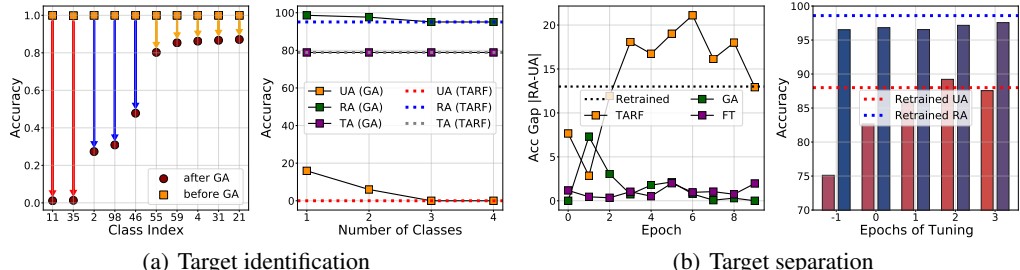

(a) Target identification  (b) Target separation

We show (a) accuracy changes in target identification in target mismatch forgetting, unlearning performance using different forgetting classes in data mismatch forgetting; (b) accuracy gap of retaining and forgetting part of the same class, as well as the need of reconstruction.

Figure 5: Target identification and target separation for unlearning under mismatch.

given the initially provided forgetting data $\mathcal{D}_f$ and the remaining set $\mathcal{D}_{un}$. Specifically, we design the two dynamic hyperparameters $k(t)$ and $\tau(x, y, t)$ as follows to achieve that,

$$k(t) = \frac{k \cdot (T - t - t_0)}{T}, \quad t \in [0, T]; \quad \tau(x, y, t) = \begin{cases} 0 & I_{con}(x, y, \theta_{t_1}) > \beta \text{ or } t < t_1, \\ 1 & I_{con}(x, y, \theta_{t_1}) < \beta \text{ and } t \geq t_1, \end{cases} \quad (7)$$

where $T$ indicates the total training time (e.g., epochs), and the value of $k(t)$ decreases with the training process, $\beta$ can be estimated by the information of forgetting dynamics about the specific unlearning request and the rank of loss/accuracy change at $t_1$, $t_0$ and $t_1$ respectively control the end time of active forgetting and the begin time of retaining part. The overall process can be regarded as,

**Phase I: Target Identification.** Before $t_1$, since $\tau(x, y, t) = 0$, Eq. 5 can be formalized as, $L_{\text{TARF-Phase-I}} = k(t) \cdot (-\frac{1}{|\mathcal{D}_f|} \sum_{(x,y)\sim\mathcal{D}_f} \ell(f(x), y))$, in which the retaining part is waiting for the dynamic information revealed by this phase. As shown in Figures 3, the false retaining data in $\mathcal{D}_{fr}$ can be identified due to the similar forgetting dynamics with the forgetting data. To conduct the specific selection, we utilize the class label information in our main tasks, for which we detail the implementation of controlling $\beta$ in Appendix E. In Figure 5(a), we show the selected classes via the accuracy variance and validate identification efficacy with varied given forgetting classes.

**Phase II: Target Separation.** After phase I, the retaining part is engaged with the forgetting part with the identified data $\mathcal{D}_{fr}$ and the remaining retaining data $\mathcal{D}_r$. By simultaneously considering the forgetting and retaining part as Eq. 5, $L_{\text{TARF-Phase-II}}$ encourages the model to deconstruct the target concept and reconstruct the feature representation of the retaining part, which can effectively decouple the pre-entangled feature in the model mismatch forgetting. In the left of Figure 5(b), we compare the accuracy gap on the retaining and forgetting part, demonstrating the necessity of considering two parts of objectives to separate the entangled feature representation in model mismatch forgetting.

**Phase III: Retraining Approximation.** After $t_0$, we focus on retaining in the current phase, which approximates the retraining objective as follows, $L_{\text{TARF-Phase-III}} = \frac{1}{|\mathcal{D}_{un}|} \sum_{(x,y)\sim\mathcal{D}_{un}} \ell(f(x), y) \cdot \tau(x, y, t)$, where we use $\tau$ at $t_1$ to indicate the identified hard-to-effect retaining data, and continually reconstruct the representations. Since the general goal of unlearning considered in our work is similar to retraining, this phase can prevent excessive forgetting. In the right of Figure 5(b), we compare the performance using different lengths of this phase for approximating the retrained reference.

## 4 EXPERIMENTS

In this section, we present the comprehensive experimental results. To begin with, we introduce the basic setups for the unlearning and the evaluation (Section 4.1). Then, we validate the effectiveness of our method in four unlearning scenarios with the decoupled class label and the target concept (Section 4.2). To better understand its properties, we conduct various experiments on the ablation study and provide further discussions (Section 4.3). More details and results are provided in Appendix F.

Table 2: Main Results (%). Comparison with the unlearning baselines. All methods are trained on the same backbone, i.e., the basis of unlearning initialization is the same (except for the reference Retrained from scratch). Bold numbers are superior results. ↓ indicates smaller values are better. (The complete results under multiple runs are summarized with mean and std values in Appendix F.7)

| Type / $\mathcal{D}$ | Dataset | CIFAR-10 | | | | | | CIFAR-100 | | | | | |
|---|---|---|---|---|---|---|---|---|---|---|---|---|---|
| | Method / Metrics | UA | RA | TA | MIA | Gap↓ | TIME↓ | UA | RA | TA | MIA | Gap↓ | TIME↓ |
| **All matched** | Retrained (Ref.) | 0.00 | 99.51 | 94.69 | 100.00 | - | 43.3 | 0.00 | 97.85 | 76.03 | 100.00 | - | 43.2 |
| | FT [58] | 1.07 | 98.62 | 92.36 | 100.00 | 1.07 | 4.43 | 0.67 | 96.32 | 72.34 | 100.00 | 1.47 | 5.02 |
| | RL [56] | 4.13 | 97.65 | 91.23 | 100.00 | 2.36 | 4.88 | 1.00 | 96.09 | 72.00 | 100.00 | 1.70 | 4.96 |
| | GA [28] | 0.49 | 95.24 | 88.17 | 99.78 | 2.88 | **0.25** | 1.33 | 94.74 | 68.56 | 99.89 | 3.01 | **0.06** |
| | IU [29] | 0.22 | 88.15 | 82.38 | 99.96 | 5.99 | 0.45 | 0.00 | 37.61 | 29.58 | 100.00 | 26.67 | 0.51 |
| | BS [6] | 25.04 | 87.94 | 80.90 | 88.67 | 15.43 | 0.82 | 4.60 | 90.18 | 63.66 | 99.55 | 6.27 | 0.78 |
| | $L_1$-sparse [30] | 0.00 | 94.20 | 89.77 | 100.00 | 2.56 | 4.39 | 0.00 | 94.60 | 71.57 | 100.00 | 1.93 | 4.39 |
| | SalUn [11] | 0.00 | 91.32 | 86.87 | 100.00 | 4.00 | 5.65 | 0.00 | 75.34 | 62.14 | 100.00 | 9.10 | 5.75 |
| | SCRUB [37] | 0.00 | 99.94 | 91.00 | 100.00 | 1.03 | 2.88 | 0.00 | 99.98 | 76.75 | 100.00 | **0.71** | 3.23 |
| | **TARF** (ours) | 0.00 | 98.23 | 91.95 | 100.00 | **1.01** | 4.21 | 0.00 | 96.90 | 72.53 | 100.00 | 1.11 | 4.68 |
| **Model mismatch** | Retrained (Ref.) | 87.76 | 99.58 | 95.91 | 20.57 | - | 43.8 | 88.22 | 98.58 | 78.50 | 25.78 | - | 43.8 |
| | FT [58] | 94.67 | 98.53 | 93.56 | 9.56 | 5.33 | 4.29 | 92.67 | 95.02 | 79.34 | 16.33 | 4.58 | 4.86 |
| | RL [56] | 53.69 | 97.85 | 92.39 | 96.60 | 28.84 | 4.82 | 80.11 | 95.83 | 79.83 | 99.00 | 21.35 | 4.93 |
| | GA [28] | 5.76 | 86.99 | 82.20 | 94.98 | 45.68 | **0.25** | 6.78 | 94.83 | 76.96 | 97.78 | 39.68 | **0.06** |
| | IU [29] | 23.69 | 87.34 | 82.57 | 89.87 | 39.74 | 0.44 | 34.67 | 96.83 | 79.08 | 86.44 | 29.14 | 0.49 |
| | BS [6] | 10.29 | 50.77 | 49.39 | 95.96 | 62.05 | 0.79 | 18.11 | 95.90 | 72.28 | 95.22 | 37.14 | 0.89 |
| | $L_1$-sparse [30] | 93.11 | 94.76 | 91.63 | 14.44 | 5.15 | 4.24 | 90.22 | 94.78 | 78.81 | 18.88 | 3.25 | 5.00 |
| | SalUn [11] | 8.91 | 93.95 | 84.38 | 99.32 | 43.69 | 6.04 | 66.33 | 78.83 | 70.78 | 77.00 | 25.15 | 5.97 |
| | SCRUB [37] | 95.14 | 99.81 | 94.22 | 15.38 | 3.61 | 3.06 | 91.44 | 99.74 | 79.23 | 21.11 | 2.45 | 4.12 |
| | **TARF** (ours) | 91.11 | 97.49 | 92.49 | 17.82 | **2.90** | 4.31 | 86.67 | 97.05 | 80.07 | 26.00 | **1.21** | 4.81 |
| **Target mismatch** | Retrained (Ref.) | 0.00 | 99.38 | 93.85 | 100.00 | - | 52.1 | 0.00 | 97.85 | 73.72 | 100.00 | - | 53.2 |
| | FT [58] | 50.43 | 98.47 | 91.65 | 50.44 | 25.78 | 4.38 | 58.18 | 96.32 | 72.53 | 46.76 | 28.54 | 5.00 |
| | RL [56] | 51.25 | 97.56 | 90.90 | 56.23 | 24.95 | 4.79 | 58.89 | 96.05 | 72.20 | 46.98 | 28.81 | 4.93 |
| | GA [28] | 40.82 | 97.01 | 89.51 | 64.32 | 20.80 | **0.26** | 21.38 | 96.64 | 70.22 | 90.67 | 8.86 | **0.05** |
| | IU [29] | 44.51 | 88.07 | 81.80 | 58.73 | 27.29 | 0.44 | 30.62 | 37.19 | 29.58 | 63.69 | 42.93 | 0.50 |
| | BS [6] | 53.62 | 88.65 | 75.39 | 76.33 | 26.62 | 0.82 | 40.44 | 98.32 | 68.66 | 85.16 | 15.20 | 0.97 |
| | $L_1$-sparse [30] | 49.47 | 93.61 | 88.83 | 51.24 | 27.26 | 4.38 | 56.09 | 94.63 | 72.00 | 48.04 | 28.25 | 4.78 |
| | SalUn [11] | 46.63 | 91.08 | 86.31 | 60.94 | 25.38 | 5.90 | 59.64 | 75.52 | 62.37 | 65.96 | 27.35 | 5.81 |
| | SCRUB [37] | 49.98 | 99.94 | 92.10 | 50.18 | 25.53 | 2.89 | 59.64 | 99.99 | 75.32 | 44.89 | 29.90 | 3.52 |
| | **TARF** (ours) | 0.06 | 97.57 | 90.81 | 100.00 | **1.23** | 4.23 | 0.31 | 97.35 | 73.68 | 100.00 | **0.21** | 4.85 |
| **Data mismatch** | Retrained (Ref.) | 0.00 | 99.54 | 95.56 | 100.00 | - | 52.1 | 0.00 | 98.50 | 80.15 | 100.00 | - | 53.2 |
| | FT [58] | 96.79 | 98.49 | 93.26 | 6.48 | 48.41 | 4.32 | 82.62 | 95.66 | 79.77 | 37.24 | 37.15 | 4.93 |
| | RL [56] | 76.47 | 97.68 | 91.93 | 49.81 | 33.04 | 4.76 | 89.78 | 96.82 | 79.90 | 70.76 | 30.49 | 4.97 |
| | GA [28] | 8.69 | 96.41 | 90.78 | 93.03 | 5.89 | **0.25** | 6.00 | 97.65 | 79.23 | 98.04 | 2.43 | **0.05** |
| | IU [29] | 22.84 | 95.50 | 89.54 | 88.57 | 11.08 | 0.44 | 31.51 | 98.96 | 78.20 | 88.09 | 11.46 | 0.48 |
| | BS [6] | 16.70 | 61.21 | 49.76 | 92.24 | 22.37 | 0.82 | 15.38 | 98.50 | 72.28 | 96.22 | 6.76 | 0.96 |
| | $L_1$-sparse [30] | 95.76 | 94.31 | 91.08 | 9.52 | 48.99 | 4.78 | 88.31 | 94.91 | 79.02 | 22.49 | 42.64 | 5.03 |
| | SalUn [11] | 51.77 | 93.87 | 90.46 | 63.52 | 24.75 | 5.72 | 72.93 | 78.87 | 71.04 | 54.13 | 36.89 | 5.72 |
| | SCRUB [37] | 97.13 | 99.89 | 95.03 | 10.99 | 46.76 | 2.94 | 95.50 | 99.79 | 79.68 | 15.11 | 45.54 | 3.68 |
| | **TARF** (ours) | 0.00 | 98.17 | 93.09 | 100.00 | **0.96** | 4.22 | 0.00 | 95.01 | 78.98 | 100.00 | **1.17** | 4.78 |

## 4.1 EXPERIMENTAL SETUP

**Datasets, models, and baselines.** In our experiments, we explore machine unlearning for conventional image classification tasks. Since the introduced unlearning settings need a coarse-to-fine label structure, we adopt the benchmarked dataset, e.g., CIFAR-10/CIFAR-100 [35] with their superclass information in the major experiments. Specifically, we train two models based on the original classes and its superclass respectively, and instantiate four tasks (as illustrated in Figure 1) of unlearning with the decoupled class label and the target concept. The detailed information is summarized in Appendix D.4. Following previous works [58, 30, 11], we use ResNet-18 [24] as the major architecture to obtain the original trained models with standard learning [18], and then set it to be the basis for unlearning. In addition, we also adopt TinyImageNet [38], ImageNet [36] for large-scale experiments. As for comparison, we consider four representative baselines with the retrained model (Retrained), e.g., FT [58, 18], RL [56], GA [28], IU [29], and also consider four recent advanced methods, e..g, BS [6], $L_1$-sparse [30], SalUn [11], and SCRUB [37]. The method details are in Appendix C.1.

**Evaluation metrics.** The general target of class-wise unlearning considered in this work is to approximate the Retrained model [58]. To give a comprehensive evaluation, we adopt 5 specific evaluation metrics in classification tasks following previous works [30, 11]. We utilize Unlearning Accuracy (UA) to evaluate the accuracy of the unlearning targeted subset; Retaining Accuracy (RA) to evaluate the accuracy of the retaining subset; Testing Accuracy (TA) to evaluate the generalization ability of the model; Membership Inference Attack (MIA) to evaluate the efficacy of unlearning by the confidence-based predictor. Note that any single indicator does not represent optimally in the approximation of a Retrained reference. All the above will be compared with that of the Retrained model and summarized in a "Gap" value (averaged gap with Retrained) to indicate the overall performance (the lower the better), and we also adopt TIME to present the computational time. The implementation of evaluation metrics in different unlearning scenarios is detailed in Appendix C.2.

Table 3: Results (%). Comparison with the baselines on TinyImageNet-200 trained on a larger model structure, i.e., ResNet101. (More results on large-scale dataset can refer to Appendix F.5)

| Type / $\mathcal{D}$ | Dataset | All matched | | | | | | Model mismatch | | | | | |
|---|---|---|---|---|---|---|---|---|---|---|---|---|---|
| | Method / Metrics | UA | RA | TA | MIA | Gap↓ | TIME↓ | UA | RA | TA | MIA | Gap↓ | TIME↓ |
| | Retrained (Ref.) | 0.00 | 74.32 | 63.13 | 100.00 | - | 217.0 | 34.80 | 71.26 | 64.29 | 66.90 | - | 256.14 |
| | FT [58] | 3.80 | 77.66 | 62.98 | 97.30 | 2.50 | 30.41 | 59.30 | 77.26 | 62.92 | 38.00 | 15.19 | 37.44 |
| | RL [56] | 73.20 | 69.87 | 60.49 | 18.40 | 40.47 | 225.13 | 84.10 | 68.53 | 60.63 | 8.00 | 28.64 | 226.79 |
| | GA [28] | 5.70 | 63.26 | 57.09 | 87.50 | 8.83 | 0.34 | 6.30 | 63.17 | 58.04 | 90.70 | 16.66 | 0.34 |
| | $L_1$-sparse [30] | 3.70 | 76.63 | 62.55 | 97.50 | 2.28 | 40.79 | 59.40 | 76.30 | 62.80 | 38.80 | 14.81 | 37.05 |
| | SCRUB [37] | 0.00 | 75.06 | 63.82 | 100.00 | **0.36** | 66.69 | 37.70 | 73.89 | 64.20 | 57.30 | 3.81 | 58.53 |
| | **TARF** (ours) | 0.00 | 75.47 | 62.79 | 100.00 | 0.37 | 28.22 | 34.00 | 74.28 | 62.60 | 65.00 | **1.85** | 28.21 |
| **Tiny ImageNet** | Dataset | Target matched | | | | | | Data mismatch | | | | | |
| | Method / Metrics | UA | RA | TA | MIA | Gap↓ | TIME↓ | UA | RA | TA | MIA | Gap↓ | TIME↓ |
| | Retrained (Ref.) | 0.00 | 72.83 | 65.12 | 100.00 | - | 213.05 | 0.00 | 71.37 | 65.76 | 100.00 | - | 252.62 |
| | FT [58] | 29.67 | 75.94 | 62.97 | 69.30 | 16.41 | 30.41 | 64.33 | 75.45 | 62.96 | 30.60 | 35.15 | 37.44 |
| | RL [56] | 68.93 | 69.97 | 60.55 | 22.00 | 38.59 | 225.13 | 84.27 | 68.64 | 60.59 | 7.86 | 46.08 | 226.79 |
| | GA [28] | 11.33 | 63.63 | 57.26 | 81.00 | 11.85 | 0.34 | 7.33 | 63.44 | 58.24 | 89.80 | 8.25 | 0.34 |
| | $L_1$-sparse [30] | 28.93 | 75.18 | 62.55 | 69.60 | 16.06 | 40.79 | 63.90 | 74.80 | 62.80 | 31.30 | 34.75 | 37.05 |
| | SCRUB [37] | 25.67 | 75.31 | 63.85 | 73.80 | 13.90 | 66.69 | 44.07 | 74.02 | 64.25 | 46.93 | 25.33 | 58.53 |
| | **TARF** (ours) | 5.07 | 75.78 | 62.72 | 97.53 | **3.22** | 28.81 | 0.00 | 74.85 | 62.59 | 100.00 | **1.66** | 27.92 |

## 4.2 PERFORMANCE EVALUATION

In this part, we present the main comparison results with those considered baselines in the four unlearning tasks, evaluated with the four detailed metrics and the overall performance gap with retrained references. We also report results under multiple runs with mean and std values in Appendix F.7.

As the performance reference, all the retrained models (termed Retrained) are trained with the fully aligned retaining data. Note that the UA of Retrained (Ref.) in the model mismatch scenario is not equal to 0 since it is evaluated with superclass label. In Table 2, we can find the previous unlearning methods achieved satisfactory performance on the conventional all matched forgetting, but did not perform well on the other three newly considered tasks with the label domain mismatch. Specifically, since the previous methods partially rely on forgetting data or remaining data, it results in ineffective or excessive forgetting due to the insufficient representation or decomposition missing. For example, FT can retain a similar RA with the Retrained but be less effective in forgetting, while GA reaches the lowest UA across different tasks but sacrifices too much model performance on the retaining dataset. In contrast, our TARF can consistently perform better (or comparable with the best method) through simultaneous gradient ascent and target-aware gradient descent to restrict the forgetting regions on the four unlearning tasks.

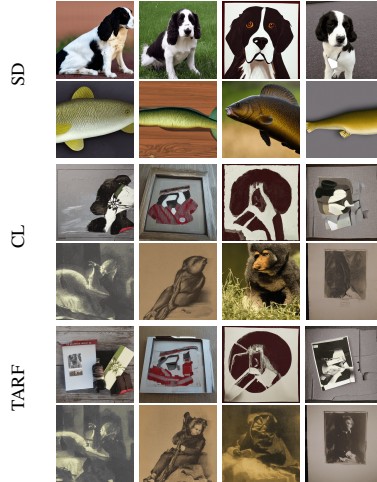

Figure 6: Image generation results of original and unlearned stable diffusion. More results are in Tables 15 and 16.

To verify TARF in a large-scale dataset, we also conduct unlearning on Tiny-ImageNet trained on ResNet-101 in Table 3, and also in ImageNet-1k as well as forgetting multiple classes in Appendix F. The results again show that our TARF can achieve satisfactory performance regarding the overall gap with Retrained. In addition, we also conduct a case study to unlearn the concept in stable diffusion [25], considering the practical data mismatch forgetting where users report some undesirable examples to represent the unwanted concept. In Figure 6, we show the efficacy of unlearning the "springer" and "tench", compared with the original generation results and certain label (CL) mismatching, TARF can identify the potential samples with similar features on the target concept and perform more thorough forgetting, e.g., generating less feature related to "dog" and "fish" in Figure 6. The full results are provided in Appendix E.1.

## 4.3 ABLATIONS AND FURTHER EXPLORATION

In this part, we provide further exploration of the three class-wise unlearning tasks and conduct various ablation studies to characterize TARF. More results and discussion are provided in Appendix F.

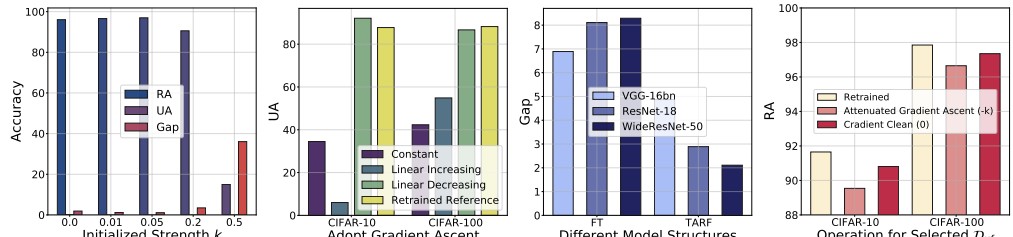

Figure 7: Ablation studies: *Left:* performance using different initialized $k$ on all matched forgetting; *middle-left:* effects of constant or different dynamic gradient ascent controlled by $k(t)$; *middle-right:* comparison of forgetting with different model structures; *right:* comparison of using different operations on the selected forgetting data. More experimental details can refer to Appendix F.

**Weighted control on annealed gradient ascent.**   To analyze the annealed gradient ascent, we present the results on the left of Figure 7 to show the effects of initialized strength $k$ on the all matched forgetting task using the CIFAR-100 dataset. The results show that an appropriate $k$ (e.g., about 0.05) can help the model to achieve a satisfactory performance. However, the larger $k$ results in lower retaining performance and higher Gap value as the strength increases on the feature deconstruction.

**Constant or dynamic gradient ascent for forgetting.**   In the middle-left of Figure 7, we study whether we need the learning-rate-reduced $k$ for the forgetting part. Specifically, we compare it with using constant $k$ and learning-rate-increased $k$ on two model mismatch forgetting tasks. The results demonstrate that annealed gradient ascent can achieve more similar performance with the Retrained on forgetting data. The gradient ascent is considered simultaneously with gradient descent for restricting the forgetting region, while we adopt the annealed one since the unlearning target is to approximate the retrained model instead of continually maximizing the loss of forgetting data.

**Unlearning on models trained by different structures.**   In the middle-right of Figure 7, we investigate forgetting on the models pre-trained using different structures, e.g., ResNet-18 [24], VGG-16bn [53], and WideResNet-50 [63]. The results of TARF on the model mismatch forgetting demonstrate that our TARF can achieve the lower performance gap than FT, evaluated with the retrained reference. With the increasing model capacity on the original training tasks, we can also find the model with a smaller capacity makes it harder to decompose the entangled feature representation for achieving the unlearning target, which increases the representation complexity.

**Differernt operations on the selected forgetting data.**   In the right of Figure 7, we present the ablation on the specific gradient operation on the selected forgetting data $\mathcal{D}_{\mathrm{fr}}$. We compare using the gradient ascent ($-k(t)$) and cleaning (0) with the Retrained reference in target mismatch forgetting. Except for the similar forgetting efficacy achieved by the three trials, major differences exist in the performance evaluated by RA. The results show that gradient cleaning may be a better choice for $\mathcal{D}_{\mathrm{fr}}$ to not deconstruct the feature representation too much and affect the retaining accuracy.

## 5 CONCLUSION

In this work, we decouple the class label and target concept in class-wise unlearning. By introducing the label domain mismatch among forgetting data, model output, and target concept, we uncover three additional tasks beyond the conventional all matched forgetting, e.g., target mismatch, model mismatch, and data mismatch forgetting. We identify the insufficient representation and decomposition lacking of restrictively forgetting the target concept, and reveal the crucial forgetting dynamics in the representation level for the feasibility of these unlearning requests. Based on that, we propose the TARF that assigns an annealed gradient ascent on the identified forgetting data and the normal gradient descent on the selected retaining data. By collaboratively considering the forgetting/retaining target, TARF is more accurate in unlearning while maintaining the rest. We hope our work can provide new insights and draw more attention toward the practical scenarios of machine unlearning.

## ETHICS STATEMENT

This paper does not raise any ethical concerns. This study does not involve any human subjects, practices to data set releases, potentially harmful insights, methodologies and applications, potential conflicts of interest and sponsorship, discrimination/bias/fairness concerns, privacy and security issues, legal compliance, and research integrity issues.

## REPRODUCIBILITY STATEMENT

We provide the link to our source codes to ensure the reproducibility of our experimental results: https://anonymous.4open.science/r/TARF-83B5/. Below we summarize some critical aspects to facilitate reproducible results:

- **Datasets.** The datasets we used are all publicly accessible, which is introduced in Section 4.1. For our newly introduced unlearning scenarios, we provide the specific dataset construction in our code, implemented as described in Section 4.1 and Appendix D.4.
- **Assumption.** Following the previous work [58, 30, 11], We set our experiments to a tuning scenario where a well-trained model is available, and all the training samples are available but limited samples are labeled as "to be unlearned".
- **Open source.** The code repository will be available in an anonymous repository for the reviewing purposes. We provide a series of unlearning methods considered in our work and also the pre-trained model for unlearning.
- **Environment.** All experiments are conducted with multiple runs on NVIDIA Tesla V100-SXM2-32GB GPUs with Python 3.8 and PyTorch 1.8. More detailed requirements can also refer to the environment descriptions in our aforementioned source codes.

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

APPENDIX

The whole Appendix is structured in the following manner. In Appendix A, we provide the anonymous link to our source code and introduce the critical aspects of reproducibility. In Appendix B, we provide a detailed discussion with related works of machine unlearning and other aspects. In Appendix C, we review the representative baseline methods in machine unlearning, which are considered in our experimental comparisons. In Appendix D, we introduce the complete scenarios considering the mismatch issues in machine unlearning, going beyond the four basic scenarios presented in the main text. In Appendix E, we formally present the algorithm implementation of our proposed TARF with its variant, and further explanation of the rationality of TARF in unlearning. In Appendix F, we provide additional experimental results to characterize forgetting dynamics and the properties of TARF. In Appendix G, we discuss the potential broader impact and limitations of our work.

## A    REPRODUCIBILITY STATEMENT

We provide the link to our source codes to enhance the reproducibility of our experimental results: https://anonymous.4open.science/r/TARF-83B5/. Below we summarize some critical aspects to facilitate reproducible results:

- **Datasets.** The datasets we used are all publicly accessible, which is introduced in Section 4.1. For our newly introduced unlearning scenarios, we provide the specific dataset construction in our code, implemented as described in Section 4.1 and Appendix D.4.
- **Assumption.** Following the previous work [58, 30, 11], We set our experiments to a tuning scenario where a well-trained model is available, and all the training samples are available but limited samples are labeled as "to be unlearned".
- **Open source.** The code repository will be available in an anonymous repository for the reviewing purposes. We provide a series of unlearning methods considered in our work and also the pre-trained model for unlearning.
- **Environment.** All experiments are conducted with multiple runs on NVIDIA Tesla V100-SXM2-32GB GPUs with Python 3.8 and PyTorch 1.8. More detailed requirements can also refer to the environment descriptions in our aforementioned source codes.

## B    DISCUSSION ABOUT RELATED WORK

In this section, we discuss the related literature on machine unlearning, and provide more detailed comparisons of some work with their approaches and motivations.

### B.1    MACHINE UNLEARNING

Machine unlearning targets to adjust a trained model to scrub the data influence [33, 52, 59]. It is initially proposed to protect data privacy [5, 4, 16], and a series of studies explore probabilistic methods through the differential privacy [16, 21, 47, 57, 50]. Although having the provable guarantee on the unlearning errors, the strong algorithmic assumptions hinders the practical effectiveness [30]. Current research [17, 55, 54, 11, 6, 60, 14, 64] focus more on developing more effective and efficient unlearning methods to approximate the Retrained model, with the given trained model. As for the assumption on data generation, prior works [17, 58, 30, 6] mainly consider all matched forgetting targets, with similar features on the original training tasks. As for the assumption on label generation, most prior works [4, 20, 54, 30, 11, 12] assume the accessibility on the fully identified forgetting dataset, and the complementary is the remaining dataset. One recent work [61] considers unlearning with only a few forgetting samples but requires another generative model to generate approximated data. Our work considers a more practical scenario in which we can conduct mismatched forgetting and use limited identified forgetting data with the remaining set. More details and the intuition of related baseline methods are introduced in Appendix C.

### B.2    POSITIVE-UNLABELED LEARNING

Positive-unlabeled learning [10, 45] tries to learn a binary classifier from a few labeled positive samples with the rest unlabeled ones. A series of PU algorithms [40, 8, 9] are developed to train

an accurate binary classifier, and can be roughly divided into two categories [1]. The first branch is cost-sensitive learning, which is related to importance weighting [41]. Given the estimated class prior, these methods [9, 32, 7] can develop an unbiased or consistent risk estimator for PU learning. Another branch of PU learning adopts two heuristic steps to perform binary classification. Such methods [40, 62] first identify reliable negative and positive examples from the unlabeled data, and then conduct semi-supervised learning. The model trained using cost-sensitive learning can also be a recognizer for positive or negative samples [27]. Different from PU learning focusing on binary classification tasks, our work tries to enable more practical scenarios in class-wise unlearning [52] where the class labels and target concepts are decoupled, and we consider the label domain mismatch.

## C  DETAILS ABOUT CONSIDERED BASELINES AND METRICS

In this section, we provide details about the considered representative baselines for machine unlearning methods, as well as their general intuitions with specific objectives. For the specific hyperparameters adopted in different methods, we keep the same setting with previous related works [30, 11], and the specific values are listed in detail in our source codes. In addition, we introduce the evaluation metrics in detail, corresponding to the implementations in different unlearning scenarios.

### C.1  UNLEARNING METHODS

**Finetune (FT).**  Utilizing the catastrophic forgetting [31] in the model (e.g., existed in the continual learning), FT [58] fine-tunes the given trained model partially on $\mathcal{D}_r$ with few training epochs to obtain the $\theta_{un}^*$ with the following objective function,

$$L_{FT} = \frac{1}{|\mathcal{D}_r|} \sum_{(x,y) \sim \mathcal{D}_r} \ell(f(x), y). \tag{8}$$

**Gradient Ascent (GA).**  Different from the normal gradient descent, GA reverses the gradient signal on $\mathcal{D}_f$ to conduct maximization with ascended gradients, resulting in the increasing loss of the forgetting data to obtain the $\theta_{un}^*$. The objective is given as follows,

$$L_{GA} = -\frac{1}{|\mathcal{D}_f|} \sum_{(x,y) \sim \mathcal{D}_f} \ell(f(x), y). \tag{9}$$

With reverse optimization to maximize the loss on the specific data, the model can approximate $\theta^*$ by directly forgetting the learned knowledge represented by the forgetting data.

**Random Label (RL).**  Similar to GA, RL [17] assign the random labels $Y^*$ on the forgetting data in $\mathcal{D}_f$ and fine-tune the given model with it to obtain the unlearned model $\theta_{un}^*$,

$$L_{RL} = \frac{1}{|\mathcal{D}_f|} \sum_{(x,y) \sim \mathcal{D}_f} \ell(f(x), y^*). \tag{10}$$

Instead of using the original training label on the forgetting data in $\mathcal{D}_f$, RL can destroy the learned feature by using the random label $y^*$ on $\mathcal{D}_f$, which violate the minimized loss value.

**Influence Unlearning (IU).**  IU adopts the influence function [33] to estimate the change if the training point is removed from the training loss. It is designed for random data unlearning [52] with the provable guarantee on the unlearning effects. In general, IU estimates the change in model parameters of $\theta_{un}^* - \theta$ and adds the weight perturbation to the given model to obtain the unlearned one. However, it usually requires additional model information and training assumptions for the theoretical guarantee and may suffer hyperparameter tuning with inaccurate hessian estimation [30, 11].

**Boundary Shrink (BS).**  BS [6] is recently proposed for class-wise unlearning, especially on the all matched forgetting. It focuses on the decision spaces [18] of the given trained model. The critical idea is to shift the original decision boundary to imitate the decision behavior of the model retrained from scratch. Motivated by adversarial attacks [43], it proposes a neighbor searching method to identify the nearest but incorrect class labels $y_{near}$ for $\mathcal{D}_f$ to guide the model to unlearn the existing

class and shift the decision boundary. Using the adversarial attack to find the nearest incorrect label, the objective of BS can be formulated as follows,

$$L_{\text{BS}} = \frac{1}{|\mathcal{D}_{\text{f}}|} \sum_{(x,y) \sim \mathcal{D}_{\text{f}}} \ell(f(x), y_{\text{near}}), \tag{11}$$

where $y_{\text{near}}$ is obtained by first perturbing the forgetting data and getting the newly predicted result as,

$$x' = x + \epsilon \cdot \text{sign}(\nabla \ell(f(x), y))$$
$$y_{\text{near}} \leftarrow \text{softmax}(f(x')) \tag{12}$$

$L_1$**-sparse.** Developed based on the conventional FT, $L_1$-sparse [30] investigate the model sparsity on machine unlearning. It figures out that model sparsification can benefit the unlearning performance on different perspectives via first pruning and then conducting unlearning. By carrying out pruning and unlearning simultaneously, $L_1$-sparse proposes the sparsity-aware unlearning utilizing the $L_1$ norm-based penalty. The objective is as follows with a hyperparameter $\gamma$,

$$L_{L_1\text{-sparse}} = \frac{1}{|\mathcal{D}_{\text{r}}|} \sum_{(x,y) \sim \mathcal{D}_{\text{r}}} \ell(f(x), y) + \gamma ||\theta^*||, \tag{13}$$

and the general sparsity-aware penalty can also be added to different unlearning methods. In this work, we mainly compare the $L_1$-sparse FT as the previous work [30, 11] considered.

**SalUn.** With the concern on unlearning stability and cross-domain applicability, SalUn [11] introduces the concept of weight saliency in machine unlearning. This innovation directs the attention of unlearning into specific model weights for specific data that need to be unlearned. In general, it first generates the gradient-based weight saliency map inspired by model sparsification [30] with gradient-value thresholding, where the specific generation method is defined as,

$$\text{m}_s = \mathbf{1}(|\nabla_\theta \ell(\theta; \mathcal{D}_f)|_\theta = \theta_o| \geq \gamma), \quad \theta_u = \text{m}_s \odot (\delta\theta + \theta_o) + (1 - \text{m}) \odot \theta_o, \tag{14}$$

in which $\mathbf{1}(g \geq \gamma)$ is an element-wise indicator function that yields a value of $\mathbf{1}$ for the i-th element if and 0 otherwise, $|\cdot|$ is an element-wise absolute value operation, and $\gamma > 0$ is a hard threshold. and then conducts saliency-based unlearning using the generated saliency map. Specifically, SalUn adopts RL [17] to fine-tune the forgetting data in $\mathcal{D}_{\text{f}}$ on the salience map, and the extended objective is given as follows,

$$L_{\text{SalUn}} = \frac{1}{|\mathcal{D}_{\text{f}}|} \sum_{(x,y) \sim \mathcal{D}_{\text{f}}} \ell_{\theta_u}(f(x), y^*) + \alpha \frac{1}{|\mathcal{D}_{\text{r}}|} \sum_{(x,y) \sim \mathcal{D}_{\text{r}}} \ell(f(x), y), \tag{15}$$

More detailed operations can refer to [11], and we keep the same hyperparameter used in [11] to conduct the class-wise unlearning tasks.

**SCRUB.** SCRUB is a newly proposed unlearning algorithm based on a novel casting of the problem into a teacher-student framework [37]. It is designed to meet the desiderata of unlearning: efficiently forgetting without hurting the model utility. As the general target of SCRUB in forgetting is application-dependent, it is proposed with a recipe that works across applications: SCRUB is first to strive for maximal forget error, which is desirable in some scenarios like removing bias or restricted contents but not in others like user privacy protection. To address the latter case, SCRUB is integrated with a rewinding procedure that can reduce the forget set error appropriately when required.

Given the original model $\theta^o$ as the teacher model, the goal of SCRUB is formatting as training a student model $\theta^u$ that selectively obeys the teacher. The overall objective can be divided into two folds, the first is to remember $\mathcal{D}_{\text{r}}$ under the teacher model's guide while the second is to forget $\mathcal{D}_{\text{f}}$ by disobeying the teacher model's guide. To measure the degree to which the student model obeys the teacher model, SCRUB utilizes the following distance measure,

$$d(x; \theta^u) = D_{\text{KL}}(p(f(x; \theta^o)) || p(f(x; \theta^u))), \tag{16}$$

where $D_{\text{KL}}$ is the KL-divergence and the overall measures of the distance between the student model's and teacher model's prediction distribution. With the aforementioned distance, the objective of SCRUB is as follows,

$$L_{\text{SCRUB}} = \min_{\theta^u} \frac{\alpha}{N_r} \sum_{x_r \in \mathcal{D}_r} d(x_r; \theta^u) + \frac{\gamma}{N_r} \sum_{(x_r,y_r) \in \mathcal{D}_r} \ell(f(x_r; \theta^u), y_r) - \frac{1}{N_f} \sum_{x_f \in \mathcal{D}_f} d(x_f; \theta^u), \tag{17}$$

where the first two parts can be regarded as a variant of distillation from a teacher model on $\mathcal{D}_r$ and the third part is encouraging the student model to disobey the teacher model to forget the target data.

Due to the objective design and implementation tricks adopted in SCRUB, we find it may fail to conduct effectively unlearning when the forgetting target consists of a large amount of data (e.g., refer to its good results on Tiny-ImageNet or CIFAR-100 cases compared with that on CIFAR-10). We further debugged the failures in Tabel 4. We conjecture it may be due to its specific objective of requiring different prediction results from the original pre-trained model. As there is no hyperparameter that directly controls the forgetting part, the method needs further adjustment when being adopted in unlearning a relatively larger amount (e.g., 4500 samples in one class of CIFAR-10) of target than the original experiments (e.g., 25 or 100 samples) conducted in the paper [37], e.g., sometimes need using lower learning rate for avoiding excessive forgetting.

Table 4: Unlearning using SCRUB with different hyperparameters in the all matched forgetting task on CIFAR-10. *$L = \alpha \cdot d_{kl}(x_r; \theta^u) + \gamma \cdot \ell(f(x_r), y_r) - d_{kl}(x_f; \theta^u)$, recommending with $\gamma$=0.99, $\alpha$=0.001 [37] and lr = 0.01.

| Method | Hyperparameter | UA | RA | TA | MIA | Gap$\downarrow$ |
|---|---|---|---|---|---|---|
| Retrained | - | 0.00 | 99.51 | 94.69 | 100.00 | - |
| TARF (ours) | - | 0.00 | 98.23 | 91.95 | 100.00 | **1.01** |
| SCRUB* | $\gamma$=0.99 $\alpha$=0.001 | 0.00 | 12.92 | 12.92 | 0.00 | 67.09 |
| | $\gamma$=0.99 $\alpha$=0.01 | 0.00 | 18.30 | 18.52 | 0.00 | 64.35 |
| | $\gamma$=0.99 $\alpha$=0.1 | 0.00 | 16.99 | 17.07 | 0.00 | 65.06 |
| | omit $-d_{kl}(x_f; \theta^u)$ | 41.51 | 99.87 | 94.44 | 99.91 | 10.55 |
| | $\gamma$=0.99 $\alpha$=1.0 | 0.00 | 13.71 | 13.23 | 99.95 | 41.82 |
| | $\gamma$=0.99 $\alpha$=10.0 | 0.00 | 20.18 | 20.04 | 100.00 | 38.46 |

## C.2 EVALUATION METRICS REGARDING DIFFERENT SCENARIOS

In this part, we summarize the following list and tables of the evaluation metrics (adopted from the previous work [30, 11]) and the used labels in different unlearning scenarios,

- Unlearning Accuracy (**UA**): the accuracy of the unlearned model $\theta_u$ on the dataset of target concept $D_t$.
- Retaining Accuracy (**RA**): the accuracy of the unlearned model $\theta_u$ on retaining dataset $D_r$.
- Testing Accuracy (**TA**): the accuracy of the unlearned model $\theta_u$ on test dataset $D_{test}$ excluding the data belonging to the target concept to be forgotten.
- Model Inversion Attack (**MIA**): the MIA success rate by a confidence-based MIA predictor of the model $\theta_u$ on the dataset of target concept $D_t$. We follow [30] to implement it to find how many samples in $D_t$ can be correctly predicted as a non-training sample by the MIA predictor against $\theta_u$. First, we sample a balanced dataset from the retaining dataset $D_r$ and the test dataset excluding the forgetting data to train the MIA predictor, then it is used to count the rate of true negative predictions for forgetting data of the target concept.

Generally, in the evaluation phase, we adopt the same labels used in pre-training to measure the unlearned model. Note that in the model mismatch forgetting, as the model is trained with superclass labels, the UA is also calculated using the superclass label. Hence, the UA of the Retrained reference is not equal to 0 as indicated in Table 2, and we compare the methods mainly on the averaged performance "Gap" (calculated based on the previous four metrics) to the Retrained reference.

In Table 5, we summarize the specific label used in different unlearning scenarios. To provide an intuitive example that corresponds to the instantiated unlearning tasks like Figure 1, we present Table 6 to give overall information about the data and labels considered in each metric.

## D FULL DISCUSSION ABOUT LABEL DOMAIN MISMATCH

In this section, we discuss the full scenarios of label domain mismatch in class-wise unlearning [58, 17, 6, 30, 11]. Specifically, we will start by why focusing on class-wise unlearning, and then discuss

Table 5: The label used in evaluation metrics on different forgetting scenarios.

| Used Label | All matched | Target mismatch | Model mismatch | Data mismatch |
|---|---|---|---|---|
| UA | Class Label | Class Label | Superclass Label | Superclass Label |
| RA | Class Label | Class Label | Superclass Label | Superclass Label |
| TA | Class Label | Class Label | Superclass Label | Superclass Label |
| MIA | Class Label | Class Label | Superclass Label | Superclass Label |

Table 6: The evaluation data (label number) of different forgetting scenarios with CIFAR-100.

| Data (classes number) | All matched | Target mismatch | Model mismatch | Data mismatch |
|---|---|---|---|---|
| UA ($D_\text{t}$) | "boy", "girl" (2) | "boy", "girl", "man", "woman", "baby" (5) | part of "people" (1), which is data of "boy" and "girl" but with superclass label | "people" (1) |
| RA ($D_\text{r}$) | Other classes (98) | Other classes (95) | other part of "people" (1) with the rest superclasses (19) | Other superclasses (19) |
| TA ($D_\text{test}$) | Other classes (98) | Other classes (95) | other part of "people" (1) with the rest superclasses (19) | Other superclasses (19) |
| MIA ($D_\text{t}$) | "boy", "girl" (2) | "boy", "girl", "man", "woman", "baby" (5) | part of "people" (1), which is data of "boy" and "girl" but with superclass label | "people" (1) |

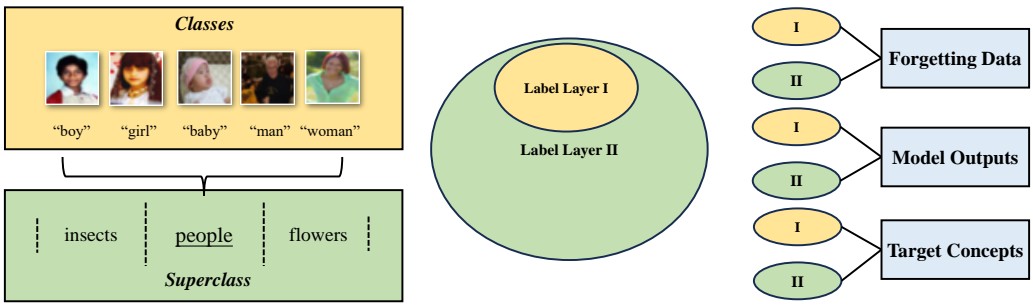

The left panel shows an example of two-layer label domains; The middle panel is the Venn diagram to show the hierarchical relation; The right panel illustrates the potentials of three critical class-wise unlearning aspects.

Figure 8: **Label domain mismatch with the two-layer illustration.**

Table 7: Mismatching in the label domain of three critical aspects with a two-layer label structure.

| No. | Forgetting data | Model output | Target concept | Comment |
|---|---|---|---|---|
| 1 | Class label | Class label | Class label | **All matched** |
| 2 | Class label | Class label | Superclass | **Target mismatch** |
| 3 | Class label | Superclass | Class label | **Model mismatch** |
| 4 | Class label | Superclass | Superclass | **Data mismatch** |
| 5 | Superclass | Class label | Class label | Impractical since $\mathcal{L}_D \succ \mathcal{L}_T$ |
| 6 | Superclass | Class label | Superclass | Similar to all matched |
| 7 | Superclass | Superclass | Class label | Impractical since $\mathcal{L}_D \succ \mathcal{L}_T$ |
| 8 | Superclass | Superclass | Superclass | **All matched** |

the motivation for investigating its label domain mismatch, with the newly introduced setting being friendly for empirical analysis and further research. In addition, we provide detailed information on our instantiated four tasks using the benchmarked datasets [35]. Finally, we discuss the commonalities of mismatch forgetting scenarios and the general principle of unified framework design.

To begin with, machine unlearning [5, 55, 59, 52] is originally proposed in response to "the right to be forgotten" to protect the data privacy, and recently deep machine unlearning is a timely research topic associated with foundation models which use massive of data to train [37, 3]. The ensuing data regulation concerns have also expanded the original privacy-protecting goal to more general needs and scenarios [60, 44, 14]. As stated in [51, 37, 30], unlearning a subset of the training set has received increasing attention (like removing sensitive information, and inappropriate content). However, the previous scenarios mainly consider the coinciding class labels with the target concept to be unlearned. Although achieving promising results in forgetting, it is still not enough in practice.

Considering the problem setups of unlearning, we have three critical aspects, e.g., the well-trained machine learning model $\theta$, and the reported data $\mathcal{D}_f$ to be unlearned, as well as the target concept. In previous studies, the three aspects are mainly considered to be under the same label taxonomy. In other words, the unlearning tasks are aligned with the pre-training task, where the latter trains a multi-class classification model, and the former aims to unlearn a training class. However, in practice, the unlearning request may violate the taxonomy of the pre-training tasks, while the specific target concepts always exhibit a unified property for specific forgetting data. It naturally motivates us to consider different label domains of the three aspects of unlearning. As listed in Table 8, the label domain of data $\mathcal{L}_D$, the label domain of model output $\mathcal{L}_M$, and the label domain of target concept $\mathcal{L}_T$. To begin with, we introduce the relations between two label domains, i.e., $\mathcal{L}_1$ matches $\mathcal{L}_2$ ($\mathcal{L}_1 = \mathcal{L}_2$), $\mathcal{L}_1$ is the subclass domain of $\mathcal{L}_2$ ($\mathcal{L}_1 \prec \mathcal{L}_2$)[3] and $\mathcal{L}_1$ is the superclass domain of $\mathcal{L}_2$ ($\mathcal{L}_1 \succ \mathcal{L}_2$)[4], and we have a practical assumption on the relation between label domains of forgetting data and target concept, i.e., $\mathcal{L}_D \preceq \mathcal{L}_T$, indicating that the reported forgetting data should be included in the target concept (as intuitively illustrated in the middle panel of Figure 8). Considering $\mathcal{L}_D = \mathcal{L}_T$, we can have two possibilities on $\mathcal{L}_M$, e.g., $\mathcal{L}_M = \mathcal{L}_T$ and $\mathcal{L}_M \neq \mathcal{L}_T$, where the former is regarded as all matched when $\mathcal{L}_D = \mathcal{L}_M = \mathcal{L}_T$ and the latter is the model mismatch. To be more specific, we consider model mismatch forgetting as $\mathcal{L}_D = \mathcal{L}_T \prec \mathcal{L}_M$, since $\mathcal{L}_M \prec \mathcal{L}_T$ will have no additional effects on the unlearning when $\mathcal{L}_D = \mathcal{L}_T$ and we can regard it as similar to the all matched case. Considering $\mathcal{L}_D \prec \mathcal{L}_T$, we can have the target mismatch forgetting when $\mathcal{L}_D = \mathcal{L}_M$ and data mismatch forgetting when $\mathcal{L}_M = \mathcal{L}_T$.

We summarize the mainly considered mismatch cases in Table 8, which can serve as a general reference for further research on constructing the unlearning tasks. In the following, we further explain the procedure of task instantiating and discuss the other potential scenarios with the typical two-layer label structure considered in the main text and an additional three-layer label structure.

| Label Domain $\mathcal{L}$ | Relation of Data $\mathcal{L}_D$, Model $\mathcal{L}_M$, and Target $\mathcal{L}_T$ | | | | |
|---|---|---|---|---|---|
| All matched | $\mathcal{L}_D$ | = | $\mathcal{L}_T$ | = | $\mathcal{L}_M$ |
| Target mismatch | $\mathcal{L}_M$ | = | $\mathcal{L}_D$ | $\prec$ | $\mathcal{L}_T$ |
| Model mismatch | $\mathcal{L}_D$ | = | $\mathcal{L}_T$ | $\prec$ | $\mathcal{L}_M$ |
| Data mismatch | $\mathcal{L}_D$ | $\prec$ | $\mathcal{L}_T$ | = | $\mathcal{L}_M$ |

Table 8: considering **label domain** relations of three critical aspects in class-wise unlearning.

## D.1 A TWO-LAYER LABEL STRUCTURE OF MISMATCH

In Figure 8, we first show the illustration of a two-layer label structure and the three aspects of unlearning, i.e., forgetting data, model outputs, and target concept. Without losing generality, we utilize the class labels and superclass information (refer to the official information in CIFAR-100 [35]) for consideration. Then we have a two-layer label structure representing different knowledge regions.

Given two potential label domains in each aspect, we can totally get the 8 scenarios list in Table 7. The first 4 scenarios are mainly considered and detailedly introduced in the main text. For the rest 4 scenarios (i.e., No. 5-8), we consider some (i.e., No. 5 and No. 7) to be impractical as the label domain of forgetting data is larger than the target concept, which means that the unlearning requests identify more forgetting data than the true target concept. It should be more reasonable that only limited forgetting data are identified by server users or internal examiner [34] in real-world applications. Therefore, we mainly consider the forgetting data $\mathcal{D}_f$ belongs a part of or equals to the overall data $\mathcal{D}_t$ of the target concept. As for No. 6 and No. 8 cases, the former is similar to the conventional all matched forgetting since the forgetting data has the same label domains with the

---

[3]$\mathcal{L}_1 \prec \mathcal{L}_2$: For any label $y \in \mathcal{L}_1$, there exist label $y' \in \mathcal{L}_2$ that instance being labeled with $y$ can also being labeled with $y'$, but not all instance being labeled with $y'$ can be labeled with $y$.

[4]$\mathcal{L}_1 \succ \mathcal{L}_2$: For any label $y \in \mathcal{L}_2$, there exist label $y' \in \mathcal{L}_1$ that instance being labeled with $y$ can also being labeled with $y'$, but not all instance being labeled with $y'$ can be labeled with $y$.

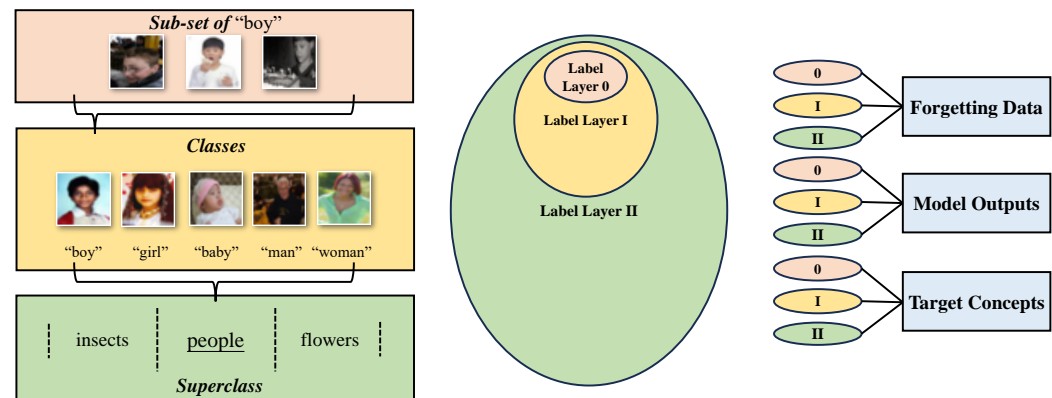

The left panel shows an example of three-layer label domains extended from the ordinary setting considered in our main text, where the sub-set is sampled from the "boy" class; The middle panel is the Venn diagram to show the hierarchical relation; The right panel illustrates the potentials of three critical class-wise unlearning aspects.

Figure 9: **Label domain mismatch with the three-layer illustration.**

target concept while the model output has a fine-grained label domain (e.g., class label) that will not affect the unlearning, and the latter is exactly same as the all matched forgetting.

### D.2 A THREE-LAYER LABEL STRUCTURE OF MISMATCH

Since in more extreme cases, some unlearning requests would exhibit only several instances of forgetting an abstract concept not aligned with the pre-training tasks. We then consider an extra label layer (e.g., the sub-set level inside a class) to construct a three-layer structure beyond the previous one. In Figure 9, we illustrate it with some samples and a Venn diagram.

Considering each aspect can have three potential label domains, we can totally get 27 scenarios in Table 9. In general, we have three rough categories for analysis. First, due to the aforementioned constraint that the target concept should include the forgetting data, we consider several cases (e.g., No. 14, 17, 18, 20, 22-24, and 26-27) to be impractical. Second, the three-layer structure also includes a group of scenarios that also existed in the two-layer structure, so No. 1-4 and 19 are the same as the four scenarios (i.e., all matched, target mismatch, model mismatch, and data mismatch). Third, for the rest scenarios, we regarded them to be novel cases than those considered in the main text.

To be more specific, there are two groups of cases in the third part. For No. 5, 6, and 7, since they also can be represented using a two-layer structure, the forgetting dynamics are similar to that in target, model, and data mismatch forgetting. By contrast, in No. 8, 9, and 16, all three label domains exist in the three aspects of class-wise unlearning, which is worthy of further discussion.

### D.3 FURTHER EXPLORATION ON THE OTHER 6 DIFFERENT SCENARIOS

In this part, we further discuss the 6 different scenarios discovered by constructing the three-layer label structure. We illustrated these forgetting tasks in Figure 10 and discuss them as follows,

- **No. 5&16** In the two scenarios, the model output has the most fine-grained label domain (e.g., sub-set as illustrated in Figure 9) for representation. At the same time, the target concept is broader than both model output and identified forgetting data. Different from the aforementioned target mismatch, the mismatch degree of this task is larger (e.g., superclass level) than the previous one (e.g., class level). In other words, the model output further loses the entanglement of feature representation of the samples belonging to target concept (compared with the original setups of target mismatch). To simulate the case, we employ the same model pre-trained by class in target mismatch, but enlarge the target concept (consists of 7 classes with similar semantic features, instead of the original 5) and change the forgetting data (2 class as the given forgetting data in No.5 and 3 classes in No.16).

- **No. 8&7.** Similar to the previous No. 5, the target concept in these tasks is also broader than the label domains of the identified forgetting data. However, in these two scenarios, the model output is

Table 9: Mismatching in the label domain of three critical aspects with a three-layer label structure.

| No. | Forgetting data | Model output | Target concept | Comment |
|---|---|---|---|---|
| 1 | Sub-set | Sub-set | Sub-set | **All matched** |
| 2 | Sub-set | Sub-set | Class label | **Target mismatch** |
| 3 | Sub-set | Class label | Sub-set | **Model mismatch** |
| 4 | Sub-set | Class label | Class label | **Data mismatch** |
| 5 | Sub-set | Sub-set | Superclass | Different |
| 6 | Sub-set | Superclass | Sub-set | Different |
| 7 | Sub-set | Superclass | Superclass | Different |
| 8 | Sub-set | Class label | Superclass | Different |
| 9 | Sub-set | Superclass | Class label | Different |
| 10 | Class label | Class label | Class label | **All matched** |
| 11 | Class label | Class label | Superclass | **Target mismatch** |
| 12 | Class label | Superclass | Class label | **Model mismatch** |
| 13 | Class label | Superclass | Superclass | **Data mismatch** |
| 14 | Class label | Sub-set | Sub-set | Impractical since $\mathcal{L}_D \succ \mathcal{L}_T$ |
| 15 | Class label | Sub-set | Class label | Similar to all matched |
| 16 | Class label | Sub-set | Superclass | Different |
| 17 | Class label | Class label | Sub-set | Impractical since $\mathcal{L}_D \succ \mathcal{L}_T$ |
| 18 | Class label | Superclass | Sub-set | Impractical since $\mathcal{L}_D \succ \mathcal{L}_T$ |
| 19 | Superclass | Superclass | Superclass | **All matched** |
| 20 | Superclass | Class label | Class label | Impractical since $\mathcal{L}_D \succ \mathcal{L}_T$ |
| 21 | Superclass | Class label | Superclass | Similar to all matched |
| 22 | Superclass | Superclass | Class label | Impractical since $\mathcal{L}_D \succ \mathcal{L}_T$ |
| 23 | Superclass | Sub-set | Sub-set | Impractical since $\mathcal{L}_D \succ \mathcal{L}_T$ |
| 24 | Superclass | Sub-set | Class label | Impractical since $\mathcal{L}_D \succ \mathcal{L}_T$ |
| 25 | Superclass | Sub-set | Superclass | Similar to all matched |
| 26 | Superclass | Class label | Sub-set | Impractical since $\mathcal{L}_D \succ \mathcal{L}_T$ |
| 27 | Superclass | Superclass | Sub-set | Impractical since $\mathcal{L}_D \succ \mathcal{L}_T$ |

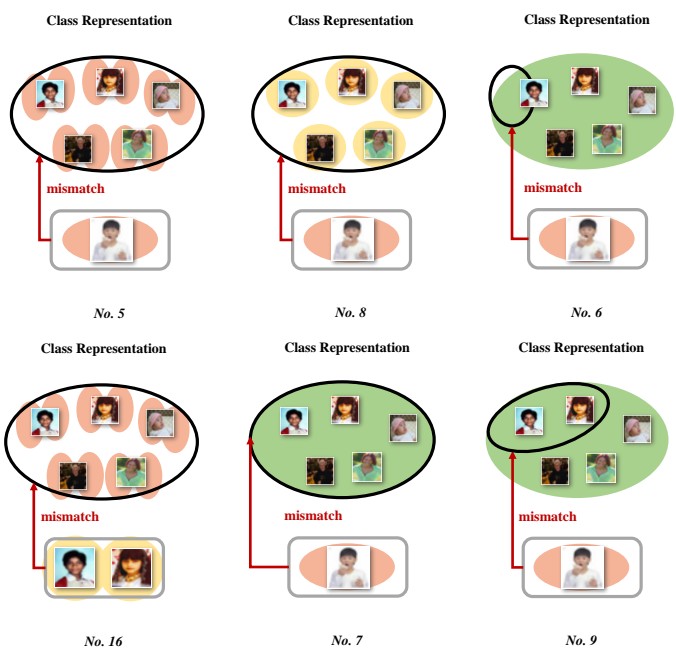

Figure 10: Illustration of 6 scenarios different from the four major tasks according to Table 9.

varied which controls the entanglement of target samples. To construct these two forgetting tasks, we respectively adopt the models pre-trained by class labels and superclass, and use the same forgetting data (with 1 class) to investigate the performance change using our TARF and other baselines.

Table 10: Results (%) of unlearning with different model structures. All methods are trained on the same backbone, i.e., the basis of unlearning initialization is the same (except for retraining from scratch). Values are percentages. Bold numbers are superior results. ↓ indicates smaller are better.

| CIFAR-100 | Metric | UA | RA | TA | MIA | Gap↓ | Metric | UA | RA | TA | MIA | Gap↓ |
|---|---|---|---|---|---|---|---|---|---|---|---|---|
| Retrained | | 0.00 | 97.85 | 73.72 | 100.00 | - | | 0.00 | 97.85 | 73.72 | 100.00 | - |
| FT [58] | | 67.52 | 96.43 | 72.96 | 41.14 | 32.72 | | 53.11 | 94.64 | 71.23 | 52.70 | 26.53 |
| RL [56] | No. 5 | 68.57 | 96.12 | 72.58 | 41.17 | 33.15 | No. 16 | 53.90 | 96.94 | 73.07 | 53.56 | 25.48 |
| GA [28] | | 38.03 | 97.00 | 70.98 | 76.92 | 16.75 | | 32.24 | 95.73 | 69.99 | 77.62 | 15.12 |
| **TARF** (ours) | | 0.00 | 96.58 | 72.03 | 100.00 | **0.74** | | 0.00 | 96.98 | 72.87 | 100.00 | **0.43** |
| Retrained | | 0.00 | 97.85 | 73.72 | 100.00 | - | | 0.00 | 98.50 | 80.15 | 100.00 | - |
| FT [58] | | 74.09 | 97.19 | 74.01 | 36.71 | 34.58 | | 95.16 | 94.98 | 78.68 | 13.06 | 46.77 |
| RL [56] | No. 8 | 76.04 | 96.76 | 72.88 | 36.00 | 35.49 | No. 7 | 91.51 | 96.98 | 80.11 | 47.24 | 36.46 |
| GA [28] | | 49.47 | 98.92 | 72.94 | 77.96 | 18.34 | | 15.91 | 98.64 | 80.27 | 93.82 | 5.59 |
| **TARF** (ours) | | 0.00 | 96.22 | 72.43 | 100.00 | **0.73** | | 0.00 | 96.54 | 79.23 | 100.00 | **0.65** |
| Retrained | | 88.22 | 98.52 | 84.42 | 22.22 | - | | 88.22 | 98.58 | 78.50 | 25.78 | - |
| FT [58] | | 94.33 | 95.00 | 78.77 | 13.67 | 5.96 | | 91.78 | 95.02 | 78.90 | 18.44 | 3.72 |
| RL [56] | No. 6 | 84.22 | 96.96 | 80.18 | 65.77 | 13.34 | No. 9 | 96.97 | 70.22 | 80.24 | 94.67 | 26.94 |
| GA [28] | | 18.44 | 96.06 | 78.20 | 92.67 | 37.23 | | 19.11 | 95.27 | 77.56 | 91.56 | 34.79 |
| **TARF** (ours) | | 92.21 | 98.43 | 82.32 | 19.17 | **2.31** | | 89.12 | 97.23 | 79.21 | 24.32 | **1.11** |

- **No. 6&9.** In the last two scenarios, the forgetting tasks are more similar to the previous model mismatch forgetting. However, the distinguishable difference is that the label domain of model outputs can be much different from the identified forgetting data. In the No. 6 task, the target concept is aligned with the identified forgetting data, while since the remaining data is more than the original model mismatch forgetting, the task separation could be harder than the previous. In the No. 9 task, we can find that it is a complex scenario where the target concept is broader than the forgetting data but included in the same superclass. In both tasks, we use 1 class data as the forgetting data.

To further understand the properties of unlearning in these tasks, we conducted additional experiments and summarized the results in Table 10. We can find the empirical results well demonstrate the conceptual conjectures in the previous discussion, and the representative baselines exhibit varied performance gap with the Retrained reference. Among them, our TARF can consistently achieve the better performance regarding to the Gap.

## D.4 Specific Information of the Instantiated Tasks

For the four major scenarios (i.e., conventional all matched forgetting, target mismatch forgetting, model mismatch forgetting, and data mismatch forgetting) considered in our work, we provide the dataset construction and partition details in this section. Note that we focus on class-wise unlearning in this work, which is different from random data forgetting that uniformly samples the forgetting target of all classes in the training dataset.

**Forgetting target.** In previous works [58, 6], the target concept to be forgotten is mainly considered as all matched where $\mathcal{D}_t = \mathcal{D}\{y = y_f\}$ has the same label domains (exactly same labels) with the pre-training task and forgetting data $\mathcal{D}_f = \mathcal{D}\{y = y_f\}$. In contrast, we assume that the target concept can be decoupled from the class label in practical unlearning requests. As illustrated in Figure 1, we further instantiate with three forgetting tasks given $\mathcal{D}_f = \mathcal{D}\{y = y_f\}$ with the superclass labels $\mathcal{Y}'$ of $\mathcal{Y}$ (classes): i) model mismatch forgetting, e.g., $\mathcal{D}_t = \mathcal{D}\{y = y_t\}$ and $y_t \subseteq y'_f$ where $y'_f \in \mathcal{Y}'$ given the model trained on $\mathcal{Y}'$; ii) target mismatch forgetting, e.g., $\mathcal{D}_t = \mathcal{D}\{y = y'_f\}$ given the model trained on $\mathcal{Y}$; iii) data mismatch forgetting, e.g., $\mathcal{D}_t = \mathcal{D}\{y = y'_f\}$ given the model trained on $\mathcal{Y}'$.

To ease the research investigation and empirical verification, we adopt the commonly used [37, 30, 11, 12] benchmark CIFAR-10 and CIFAR-100 for constructing the pre-training task for unlearning. Specifically, the official class labels are kept as classes for ordinary setup, and we provide the superclass information referring to the pre-defined lists [35] of CIFAR-100. Since there is no official superclass information for CIFAR-10 dataset, we manually grouped the classes of CIFAR-10 according to their semantic feature similarity and finalized 5 superclass clusters consisting of 2 classes in each. The full structured label layers information is summarized in Tables 12 and 13. For all the unlearning scenarios where the label domain of model output is the superclass, we will first use the

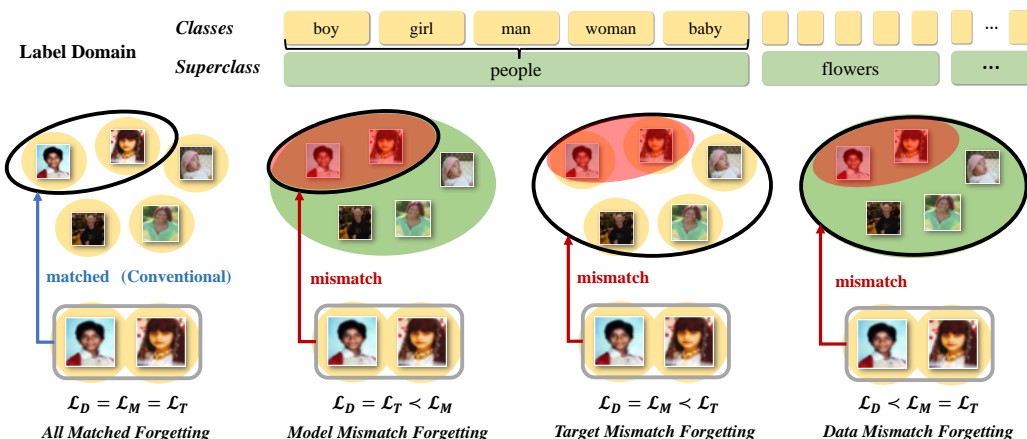

Taking the *CIFAR-100* [35] dataset, we instantiate four unlearning tasks given the same forgetting data with the class labels of "boy" and "girl": a) *all matched forgetting* (conventional scenario): unlearn "boy" and "girl" with the model trained on the classes; b) *target mismatch forgetting*: unlearn "people" with the model trained on the classes; c) *model mismatch forgetting*: unlearn "boy" and "girl" with the model trained on the superclass; d) *data mismatch forgetting*: unlearn "people" with the model trained on the superclass.

Figure 11: Illustrations of class representation with the four unlearning scenarios.

superclass information to train the 20-class and 5-class classification models respectively. For the specific data partition in unlearning requests, we randomly sampled two classes in CIFAR-100 and one class in CIFAR-10 as forgetting data and kept the setup across the four forgetting tasks as well as other experiments. For other additional experimental setups, we will state them at the near positions.

Table 11: Basic setup about unlearning scenarios. More illustrations can be found in Appendix D.4.

| Dataset | Forgetting Data | Setup | All matched | Model mismatch | Target mismatch | Data mismatch |
|---|---|---|---|---|---|---|
| CIFAR-10 | "automobile" | Training Class | 10 | 5 | 10 | 5 |
| | | Target Concept | "automobile" | "automobile" | "vehicle" | "vehicle" |
| CIFAR-100 | "boy", "girl" | Training Class | 100 | 20 | 100 | 20 |
| | | Target Concept | "boy", "girl" | "boy", "girl" | "people" | "people" |

Table 13: Full list of the 5-class classification on CIFAR-10 with its manually set superclass [35].

| Superclass (5) | Classes (2 for each superclass) |
|---|---|
| 1 | airplane, bird |
| 2 | automobile, truck |
| 3 | cat, dog |
| 4 | deer, frog |
| 5 | horse, ship |

Table 14: Specific training set data partition corresponding to four major forgetting tasks.

| Forgetting Tasks | Identified | Unidentified |
|---|---|---|
| All matched | $\mathcal{D}_f = \mathcal{D}_t$ | $\mathcal{D}_{un} = \mathcal{D}_r$ |
| Target mismatch | $\mathcal{D}_f \subset \mathcal{D}_t$ | $\mathcal{D}_{un} = \mathcal{D}_{fr} \cup \mathcal{D}_r$ |
| Model mismatch | $\mathcal{D}_f = \mathcal{D}_t$ | $\mathcal{D}_{un} = \mathcal{D}_r$ |
| Data mismatch | $\mathcal{D}_f \subset \mathcal{D}_t$ | $\mathcal{D}_{un} = \mathcal{D}_{fr} \cup \mathcal{D}_r$ |

## D.5 DISCUSSION ON THE PRACTICALITY OF LABEL DOMAIN MISMATCH

Machine unlearning is originally proposed in response to data regulations [5, 26], which are primarily motivated by a desire to protect data owners' right to withdraw from the learning process. However, regarding its technical nature of mitigating the data influence from a trained model [59], unlearning is actually given broader significance in the context of trustworthy AI [4,5], like studies for mitigating bias and unfairness [6], addressing safety issues [7], erasing the NSFW generation [8,9]. It is worth noting that these trustworthy requirements may generally exhibit different concerns from the original training tasks. Motivated by the research problem raised in Section 3.1, our work focuses on a critical problem from the assumption view, i.e., the unlearning request may have a different taxonomy from the original tasks, for which we model the three mismatched scenarios for systematical exploration.

Table 12: Full list of the 20-class classification on CIFAR-100 with its official superclass labels [35].

| Superclass (20) | Classes (5 for each superclass) |
| --- | --- |
| aquatic mammals | beaver, dolphin, otter, seal, whale |
| fish | aquarium fish, flatfish, ray, shark, trout |
| flowers | orchids, poppies, roses, sunflowers, tulips |
| food containers | bottles, bowls, cans, cups, plates |
| fruit and vegetables | apples, mushrooms, oranges, pears, sweet peppers |
| household electrical devices | clock, computer keyboard, lamp, telephone, television |
| household furniture | bed, chair, couch, table, wardrobe |
| insects | bee, beetle, butterfly, caterpillar, cockroach |
| large carnivores | bear, leopard, lion, tiger, wolf |
| large man-made outdoor things | bridge, castle, house, road, skyscraper |
| large natural outdoor scenes | cloud, forest, mountain, plain, sea |
| large omnivores and herbivores | camel, cattle, chimpanzee, elephant, kangaroo |
| medium-sized mammals | fox, porcupine, possum, raccoon, skunk |
| non-insect invertebrates | crab, lobster, snail, spider, worm |
| people | baby, boy, girl, man, woman |
| reptiles | crocodile, dinosaur, lizard, snake, turtle |
| small mammals | hamster, mouse, rabbit, shrew, squirrel |
| trees | maple, oak, palm, pine, willow |
| vehicles 1 | bicycle, bus, motorcycle, pickup truck, train |
| vehicles 2 | lawn-mower, rocket, streetcar, tank, tractor |

Here, we discuss some practical use cases for the three newly proposed settings. For example, 1) the label domain mismatch may exist in some recommendation tasks [15] or other generative tasks like image generation [14] with diversified user feedback (for which we have presented a case study on concept-forgetting on Stable Diffusion in Appendix E). 2) When the users raise unlearning requests for some representative disliked item with a message of "don't recommend this kind of thing", it is similar to target mismatch forgetting. In addition, 3) when debugging the pre-trained model with some spurious correlation or safety concerns [3, 14, 39], it is similar to model or data mismatch forgetting as we only have the forgetting cases (e.g., some figures including NSFW content or adversarial features) that may not be aligned with the taxonomy of the pre-training task. We hope our exploration can provide insights for further consideration of specific practical applications.

### D.6  DISCUSSION ON THE SCENARIO COMMONALITIES AND FRAMEWORK PRINCIPLES

The three mismatch scenarios, i.e., target mismatch forgetting, data mismatch forgetting, and model mismatch forgetting, share the common challenge of representation mismatch between the pre-trained model, the identified forgetting data, and the target concept to be forgotten. It breaks the assumption in all-matched scenarios that the three are matched [51, 59, 11, 30] and can result in extra-/ineffective-forgetting in unlearning tasks, as demonstrated in our Figure 2. Specifically, in the target/data mismatch forgetting, the target concept can be wider than the identified forgetting data; while in the model mismatch forgetting, it can be smaller than the coarse-grained model representation.

To build a unified framework like our TARF, it requires considering the aforementioned two issues, i.e., insufficient representation (refer to the observation 3.1) in target/data mismatch, and decomposition lacking (refer to the observation 3.2) in the model mismatch. The former requires a flexible controller for forgetting strength while and latter requires a simultaneous consideration on forgetting and retaining. Thus, based on the general equation in Eq. (5), we set two sub-objectives (annealed forgetting and target-aware retaining) to decompose the learned representation and control the forgetting strength by the instance-wise weighting mechanism which selects the targeted-aware forgetting data. Then, TARF becomes a unified framework for the three mismatched scenarios. Note that in our presentation, TARF is illustrated with three phases to better explain its functionality, while it is not three independent parts but unified in a general objective with specific parameters.

# E  ALGORITHM IMPLEMENTATION AND EXPLANATION

In this section, we present the pseudo-code of our proposed TARF and its variant, as well as additional discussions to enhance the understanding of our methods. Here we summarize the detailed procedure of algorithm implementation in Algorithm 1 and Algorithm 2. In detail, Algorithm 1 identifies the potential target using the class labels, while Algorithm 2 can use the instance level information.

As introduced in Section 3.3, the objective of our TARF is defined as follows,

$$L_{\text{TARF}} = k(t) \cdot \underbrace{\left( - \frac{1}{|\mathcal{D}_{\text{f}}|} \sum_{(x,y) \sim \mathcal{D}_{\text{f}}} \ell(f(x), y) \right)}_{\text{Annealed Forgetting } L_{\text{f}}(k)} + \underbrace{\frac{1}{|\mathcal{D}_{\text{un}}|} \sum_{(x,y) \sim \mathcal{D}_{\text{un}}} \ell(f(x), y) \cdot \tau(x, y, t)}_{\text{Target-aware Retaining } L_{\text{u}}(\tau)}, \tag{18}$$

where $k(t)$ and $\tau(x, y, t)$ are two training-time-related hyperparameters to deal with the mismatch issues raised in our new settings. Specifically, we set a learning-rate-reduced $k(t)$ as,

$$k(t) = k \cdot (T - t - t_0)/T, \quad t \in [0, T], \tag{19}$$

where $T$ indicates the total training time (e.g., epochs), and the value of $k(t)$ decreases with the training process. On the other hand, we have the following indicator to measure the model prediction consistency with the training data $I_{\text{con}}(x, y, \theta) = |\ell_{f_\theta}(x, y) - \ell_{f_{\theta*}}(x, y)|$, with which we set $\tau(x, y, t)$ as follows,

$$\tau(x, y, t) = \begin{cases} 0 & I_{\text{con}}(x, y, \theta_{t_1}) > \beta \text{ or } t < t_1 \quad {}^*\text{Unconf. Retain,} \\ 1 & I_{\text{con}}(x, y, \theta_{t_1}) < \beta \text{ and } t \geq t_1 \quad {}^*\text{Conf. Retain,} \end{cases} \tag{20}$$

where $t_1$ is a time stamp to control the start of pursuing the retaining part. The overall two dynamic hyperparameters can divide the whole unlearning process into three phases as illustrated in Figure 4.

In the implementation, the $\beta$ value is estimated with the ranked loss difference of each class and the prior information about the target concept to control the forgetting strength in unlearning. For the task feasibility, we will generally assume the amount of false remaining data or classes is known at our target/data mismatch forgetting, following a similar setup in learning from label noise [22] where the noise rate can be estimated and utilized as prior information.

**Annealed Forgetting.**  For the forgetting target, we adopt the gradient ascent on the given forgetting data to unlearn it. However, to approximate the retrained model, the intuition is not to pursue the maximization of the risk on this part of the data but to destroy the learned feature on the given model. So we introduce a learning-rate-reduced $k(t)$ to realize the annealed gradient ascent where $t_0 = 1$ is adopted for target or data mismatch forgetting, and the value of $k(t)$ decreases with the training process. Resulting in destroyed features, gradient ascent on this part of data also constructs the dynamic information for differentiating the data of different consistency on its loss values, making the risks of the false retaining data higher than the rest, and helping to filter retaining data.

**Target-aware Retaining.**  For the retaining part, we need to selectively learn the data from the remaining set, since the complementary dataset may be biased with unidentified forgetting data. Compared with other remaining data, the false retaining data is easy to be affected by similar feature representation as indicated in Figure 3(a). Thus, we can have $\tau(x, y, t)$ where we can divide the remaining set into unconfident/confident parts to note the estimated retaining data like Figure 5(a). $t_1 = 2$ is adopted at target and data mismatch tasks, and $\beta$ can be estimated by the prior information about the specific unlearning request and the rank of loss values. By simultaneously conducting gradient ascent on forgetting data and selective gradient descent on confident retaining data, we can better restrict the forgetting region and deconstruct the entangled feature representation (refer to the middle of Figure 5(b) where we reveal the feature decomposition in deeper layers of model structure using ResNet). Finally, with the partial objective of retaining, it can approximate the retrained reference (refer to the right of Figure 5(b)).

## E.1  CASE STUDY FOR UNLEARNING GENERATION CONCEPT

To demonstrate the compatibility, we also extend the idea of this work and investigate the performance of TARF on the specific text-to-image generation task with stable diffusion [14, 11], and presented the generated images by the original model and unlearned model in Tables 15 and 16.

---

**Algorithm 1** TARF

---

**Input:** Training dataset $D = \{(x_i, y_i, s_i)\}_{i=1}^n$, where $s_i = 1$ indicates the identified forgetting dat, otherwise the data is recognized to be unlabeled for unlearning, learning rate $\eta$, number of epochs $T$, batch size $m$, number of batches $M$, data $x \in \mathcal{X}$, label $y \in \mathcal{Y}$, original trained model $\theta$, loss function $\ell$, initialized indicator value $\tau$ with the threshold $\beta$.

**Output:** model $\theta^T$;

1: **for** mini-batch $= 1, \ldots, M$ **do**
2:     Sample a mini-batch $\{(x_i, y_i)\}_{i=1}^m$ from $D$
3:     $\{\ell(x_i, y_i)\}_{i=1}^m \leftarrow \theta.\text{forward}(f_\theta, \{(x_i, y_i)\}_{i=1}^m)$,
4:     Collect the initial training accuracy in each class based on $\{\ell(x_i, y_i)\}_{i=1}^m$,
5: **end for**
6: **for** epoch $= 1, \ldots, T$ **do**
7:     Update $k(t)$ according to Eq. 7,
8:     **if** epoch $< t_0$ **then**
9:         $\tau \leftarrow 0$
10:     **else**
11:         compute $\beta$ in Eq. 7 according to the rank of class accuracy difference, and update $\tau$.
12:     **end if**
13:     **for** mini-batch $= 1, \ldots, M$ **do**
14:         Sample a mini-batch $\{(x_i, y_i, s_i)\}_{i=1}^m$ from $D$
15:         Assign different weights for identified target samples and the rest retaining data,
16:         $L_{\text{TARF}} = k(t) \cdot \left( - \frac{1}{|\mathcal{D}_f|} \sum_{(x,y) \sim \mathcal{D}_f} \ell(f(x), y) \right) + \frac{1}{|\mathcal{D}_{\text{un}}|} \sum_{(x,y) \sim \mathcal{D}_{\text{un}}} \ell(f(x), y) \cdot \tau(x, y, t)$,
17:         $\theta \leftarrow \theta - \eta \nabla_\theta L_{\text{TARF}}(D, D_f, f, \tau)$
18:     **end for**
19: **end for**

---

In detail, we aim to unlearn the image generation of a concept with its specific prompt like "a photo of a tench". To simulate the practical unlearning request (e.g., the user raises the request of unlearning a specific concept with some identified generation examples, and the developer needs to adjust the model to forget the concept), we construct the given dataset consisting of limited forgetting data and the unidentified remaining data for unlearning, which corresponds to the data mismatch forgetting task. Then we compare the image generation on the original stable diffusion, the unlearned model with certain label (CL) mismatching [11], and that with our TARF. Note that here we recognize ESD [14] as a performance upper bound and do not compare it, since it is the same for all matched settings with fully identified forgetting data (as it directly encourages the model to unlearn the concept from text semantics). For this exploration of TARF, we adopt the instance-wise identification during the forgetting process as described in Algorithm 2, to unlearn the target concept with the given limited forgetting data and pursue retaining the selected remaining data with lower loss values.

The results in Tables 15 and 16 demonstrate that our TARF can achieve better forgetting results given the limited identified forgetting data, with proper target identification in the remaining set, while CL using only identified forgetting data can not unlearn the concept well as the generated examples still maintain some semantic features belongs to the target concept (like "tench" or "English springer").

E.2   DISCUSSION ON THE ALGORITHM COMPUTATION COST

We would acknowledge that TARF may require more time in unlearning compared with some methods like GA, which only uses the forgetting data (which sometimes can be extremely limited than other retaining data) for unlearning, while those methods may suffer from excessive forgetting and results in inaccurate unlearning across different scenarios. Regarding the metric "TIME", it originally means to avoid some methods that consume too much time compared with that of Retrained (Ref.). From this perspective, these current methods and TARF actually fall in the acceptable time range, and the efficiency gap between existing explorations in that range is indeed not a bottleneck based on Table 1.

From the methodology perspective, the three separately presented phases are integrated in a unified framework, instead of adding extra phases before and after Phase II. Compared to other approximate unlearning methods, the unique operation is target identification by comparing the output information

---

**Algorithm 2** TARF-I: generalized version on instance-wise identification

---

**Input:** Training dataset $D = \{(x_i, y_i, s_i)\}_{i=1}^n$, where $s_i = 1$ indicates the identified forgetting dat, otherwise the data is recognized to be unlabeled for unlearning, learning rate $\eta$, number of epochs $T$, batch size $m$, number of batches $M$, data $x \in \mathcal{X}$, label $y \in \mathcal{Y}$, original trained model $\theta$, loss function $\ell$, initialized indicator value $\tau$ with the threshold $\beta$.

**Output:** model $\theta^T$;

1: **for** mini-batch $= 1, \ldots, M$ **do**
2:     Sample a mini-batch $\{(x_i, y_i)\}_{i=1}^m$ from $D$
3:     $\{\ell(x_i, y_i)\}_{i=1}^m \leftarrow \theta.\text{forward}(f_\theta, \{(x_i, y_i)\}_{i=1}^m)$,
4:     Collect the initial loss values in each training samples based on $\{\ell(x_i, y_i)\}_{i=1}^m$,
5: **end for**
6: **for** epoch $= 1, \ldots, T$ **do**
7:     Update $k(t)$ according to Eq. 7,
8:     **if** epoch $< t_0$ **then**
9:       $\tau \leftarrow 0$
10:     **else**
11:       compute $\beta$ in Eq. 7 according to the rank of difference in instance loss values, and update $\tau$.
12:     **end if**
13:     **for** mini-batch $= 1, \ldots, M$ **do**
14:       Sample a mini-batch $\{(x_i, y_i, s_i)\}_{i=1}^m$ from $D$
15:       Assign different weights for identified target samples and the rest retaining data,
16:       $L_{\text{TARF}} = k(t) \cdot \left( - \frac{1}{|\mathcal{D}_{\text{f}}|} \sum_{(x,y) \sim \mathcal{D}_{\text{f}}} \ell(f(x), y) \right) + \frac{1}{|\mathcal{D}_{\text{un}}|} \sum_{(x,y) \sim \mathcal{D}_{\text{un}}} \ell(f(x), y) \cdot \tau(x, y, t)$,
17:       $\theta \leftarrow \theta - \eta \nabla_\theta L_{\text{TARF}}(D, D_{\text{f}}, f, \tau)$
18:     **end for**
19: **end for**

---

of the unlearned model with the original model for weight assignment, which has similar or less computation than other advanced designs that consider the sparse regularization [30] or compute the gradient mask for the original model [11].

### E.3 DISCUSSION ON TARF WITH LIMITED CLASS INFORMATION

The phase 1 of TARF is for target identification in the target mismatch forgetting where the target concept is wider than the given forgetting data (e.g., forgetting "people" given "boy" and "girl"). The class information may affect the accurate identification of the target concept but not the rationality of our framework. In our experimental setup, the class information is available in TARF as the class labels are used in pre-training, while the information of the target concept is given by the number of extra classes instead of the superclass label. Regarding the unavailable or implicit class information, first, if the class (i.e., the subclasses w.r.t. target concept) is not available, TARF may also utilize the model prediction to obtain the pseudo labels to conduct the task; Second, if the extra forgetting target beyond the identified data is not restricted as classes, it may require that given forgetting data can well represent the target concept (e.g., the false retaining data should be easier affected than the other retaining data). We acknowledge that both scenarios would lead to a larger performance gap with Retrained reference, as it is a generally more challenging scenario affecting the task achievability to all of the approximate unlearning methods. We believe it worth future effort to explore.

Table 15: Image generation results of unlearned Stable Diffusion in the **Data mismatch forgetting**, compared with the original stable diffusion, certain label (CL) unlearning [11], and our **TARF**. The specific prompt used in the image generation is "a photo of tench".

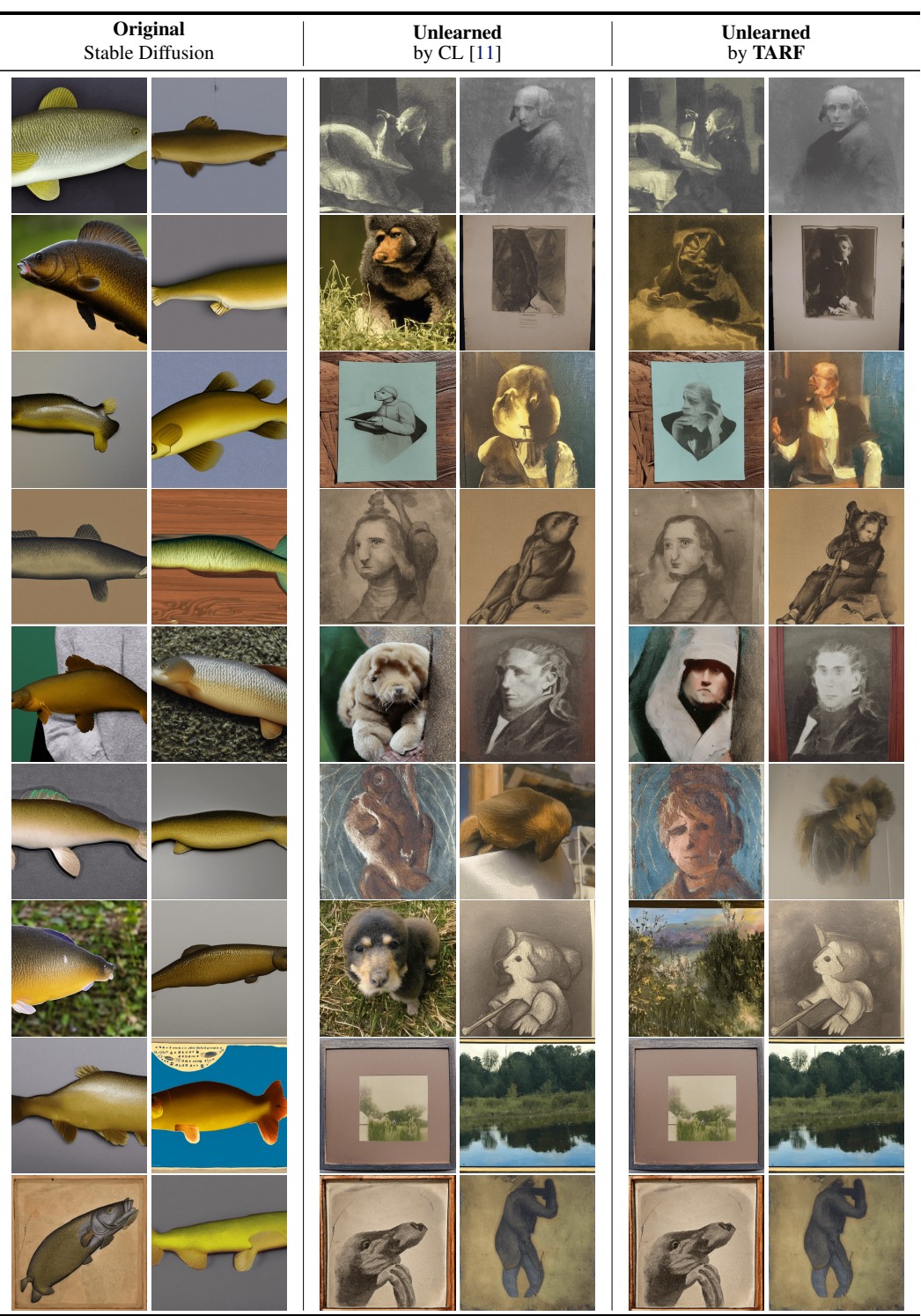

Table 16: Image generation results of unlearned Stable Diffusion in the **Data mismatch forgetting**, compared with the original stable diffusion, certain label (CL) unlearning [11], and our **TARF**. The specific prompt used in the image generation is "a photo of English springer".

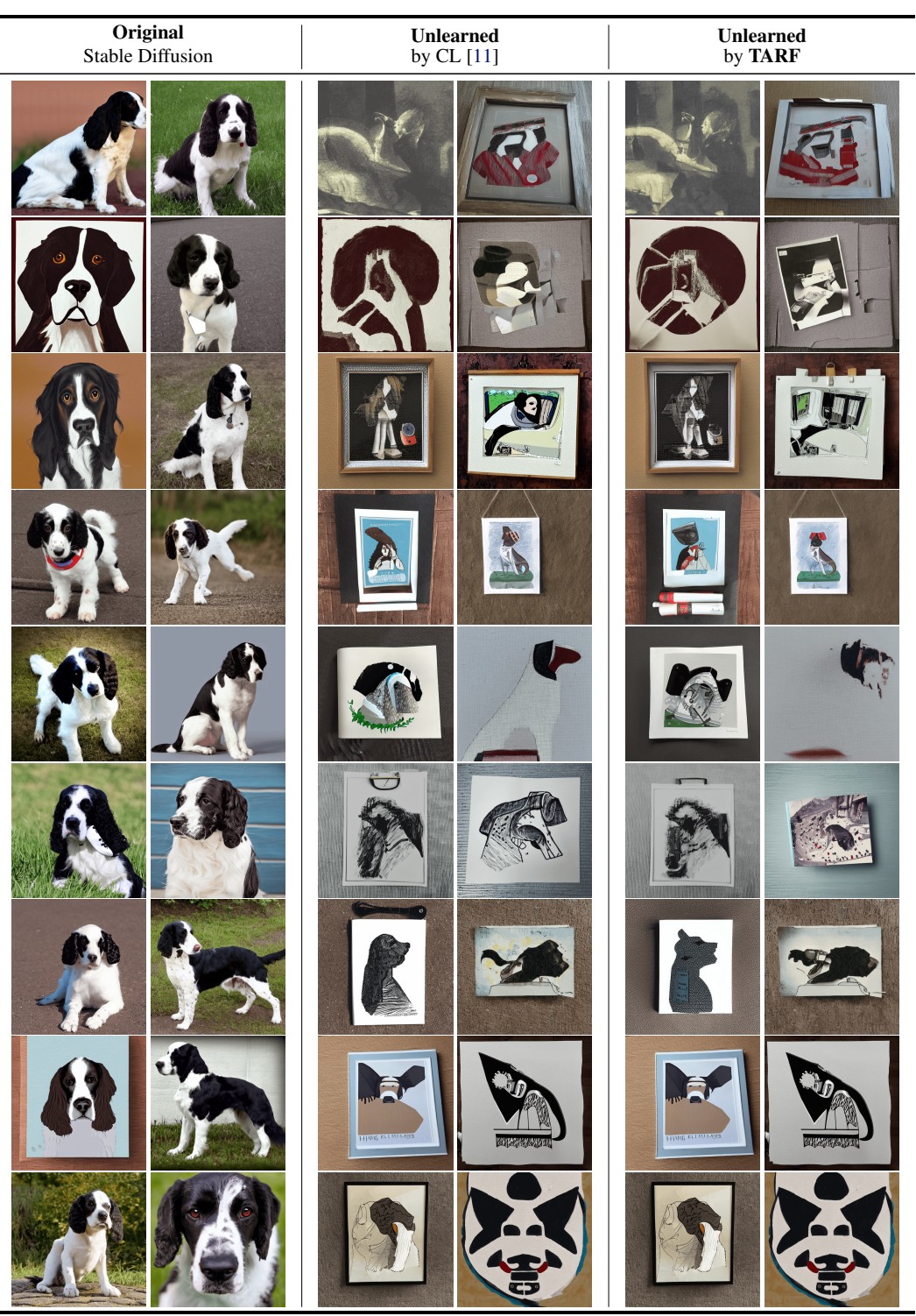

# F  ADDITIONAL EXPERIMENTAL RESULTS

In this section, we provide additional experimental results of our work.

- In Appendix F.1, we summarize the additional experimental setups.

- In Appendix F.2, we discuss the crucial target identification in unlearning.

- In Appendix F.3, we discuss and compare TARF with the advanced method in all matched scenario.

- In Appendix F.4, we discuss potential ways to extend unlearning to the scenario without class labels.

- In Appendix F.5, we verify unlearning on large-scale datasets trained with large models.

- In Appendix F.6, we present unlearning with different model structures.

- In Appendix F.7, we present the full results under multiple runs with the four forgetting tasks.

## F.1  EXTRA EXPERIMENTAL SETUPS

We introduce additional experimental details in the specific unlearning tasks. In our TARF, In general, we set $t_1 = 1$ for all the target identification parts, and we adopt $k = 0.04$, $t_0 = 2$ in model mismatch forgetting, and $k = 0.02$, $t_0 = 2$ for all matched, target mismatch and data mismatch forgetting in the unlearning request on CIFAR-10 classification task; for the CIFAR-100 classification task, we adopt $k = 0.5$, $t_0 = 2$ in model mismatch forgetting, and $k = 0.05$, $t_0 = 2$ for all matched, target mismatch and data mismatch forgetting. For the other hyperparameters, we follow the previous works [30, 37, 11] to set the specific values. All the forgetting trails use 10 epochs for the total unlearning process except for GA (use 5 epochs) and IU (use the specific fixed step for optimization). The specific parameters and the pre-trained models (unlearn base) are provided in our source codes.

## F.2  DISCUSSION ABOUT TARGET IDENTIFICATION IN UNLEARNING

In this part, we further discuss the important factors for the achievability of the unlearning tasks. To be more specific, for the target or data mismatch forgetting, the scenario assumes that the identified forgetting data is part of the whole samples belonging to the target concept, which means there are other forgetting data included in the remaining set that need to be found. Thus, target identification is important for effective unlearning. As demonstrated in Section 3.2, the representation gravity can be a useful clue in forgetting dynamics to identify the other false retaining data. An implicit assumption is that those false retaining data have similar semantic features to the initially provided forgetting data, which has smaller representation distance than the retaining part of data as illustrated in Figure 12. Empirically, the model can have similar prediction changes on those false retaining data with the initial forgetting data. However, not all of the superclasses officially defined for the CIFAR-100 dataset are suitable for constructing the unlearning request, as some superclasses are not semantically separable like "aquatic mammals" and "fish". It can be found in Figure 14, where we check the Top-10 classes with the most accuracy changes after gradient ascent for each superclass in the CIFAR-100 dataset, some false retaining data (class-level indicated by blue arrows) are not easily identified given the two initially provided forgetting data classes (indicated by red arrows). One interesting future problem can be how to handle the spurious correlation given the insufficient representative samples.

## F.3  DISCUSSION ABOUT TARF ON ALL MATCHED SCENARIO

Regarding the all matched scenario, there is no need for the target identification part to identify extra forgetting data in the all-matched scenario as the target concept matches the forgetting data, then TARF degenerates into a general framework using the given forgetting data to forget, and the rest to retain. The performance of TARF is comparable to the existing best counterpart like SCRUB regarding the "Gap↓" in Table 2. It can be found that the overall performance of the unlearned models has already closely approximated the Retrained reference. Furthermore, since TARF is a general framework, we can also adopt the KL divergence loss with the original model as designed in SCRUB to further improve the performance, for which we present the comparison in Table 17.

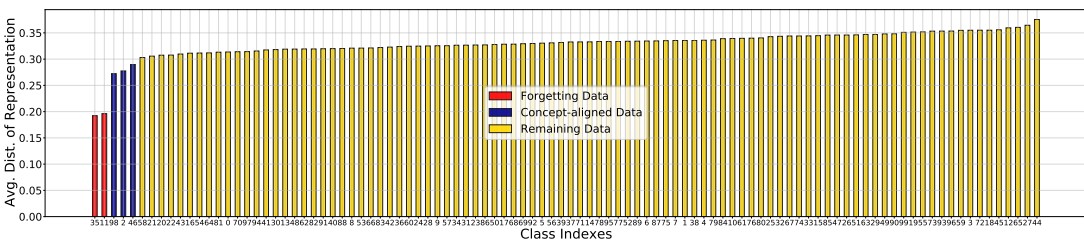

(a) Inter-classes distance in the model trained by classes

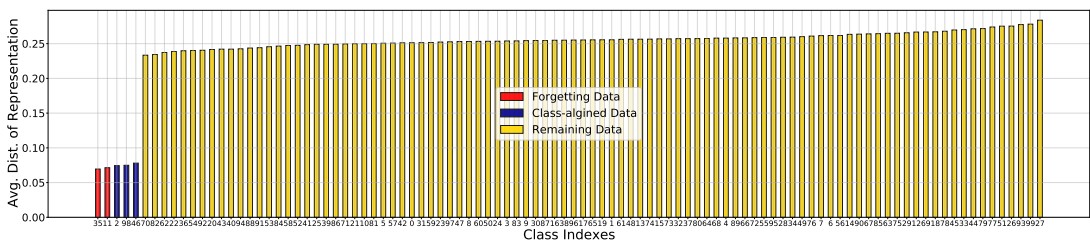

(b) Inter-superclass distance in the model trained by superclass

The distance is calculated at the feature representation extracted from the penultimate layer of the model for each class, which measures the averaged Euclidean distance to the cluster center (averaged by the forgetting data).

Figure 12: Inter-class distance and Inter-superclass distance for the unlearning assumption.

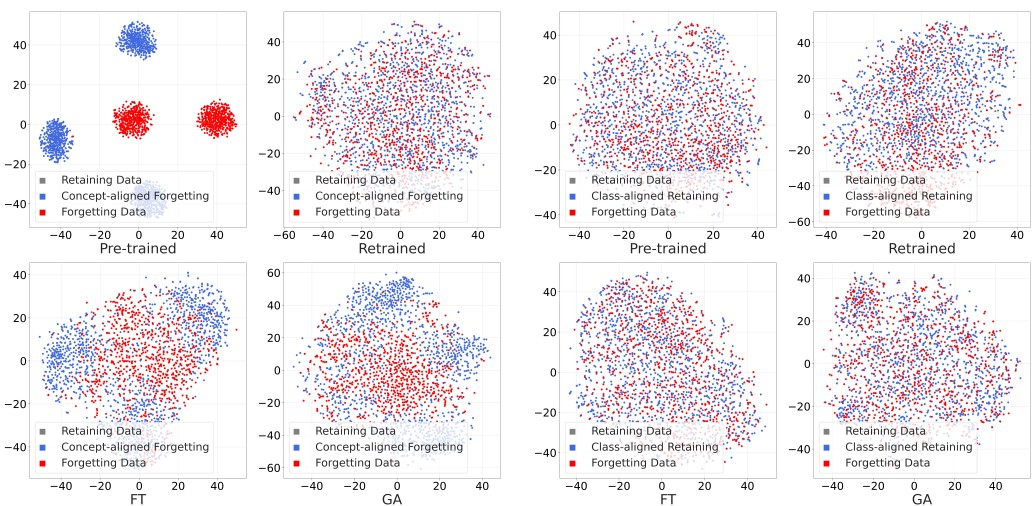

We present the tSNE visualization [42] of the learned features, using two representative unlearning methods, i.e., finetune (FT) [58] and gradient ascent (GA) [54] with the pre-trained and retrained ones.

Figure 13: The entangled/under-entangled feature representations visualized by tSNE.

## F.4 DISCUSSION ABOUT TARF ON WEAKLY-SUPERVISED SCENARIO

Our current work mainly focus on expanding the scope of conventional class-wise unlearning. Regarding the existing approximate unlearning studies [51, 37, 6, 30], considering the all matched forgetting scenario with full supervision, we push it towards more practical settings via decoupling the class labels and the target concept. For machine unlearning under weak supervision, there are limited studies [51] to our best knowledge, and we believe it is worth an in-depth exploration in future work.

Given that if a model is trained with semi-supervised or other weak supervision, we can obtain the pseudo labels by the model prediction for its unlearning phase. Instead of using the predicted label,

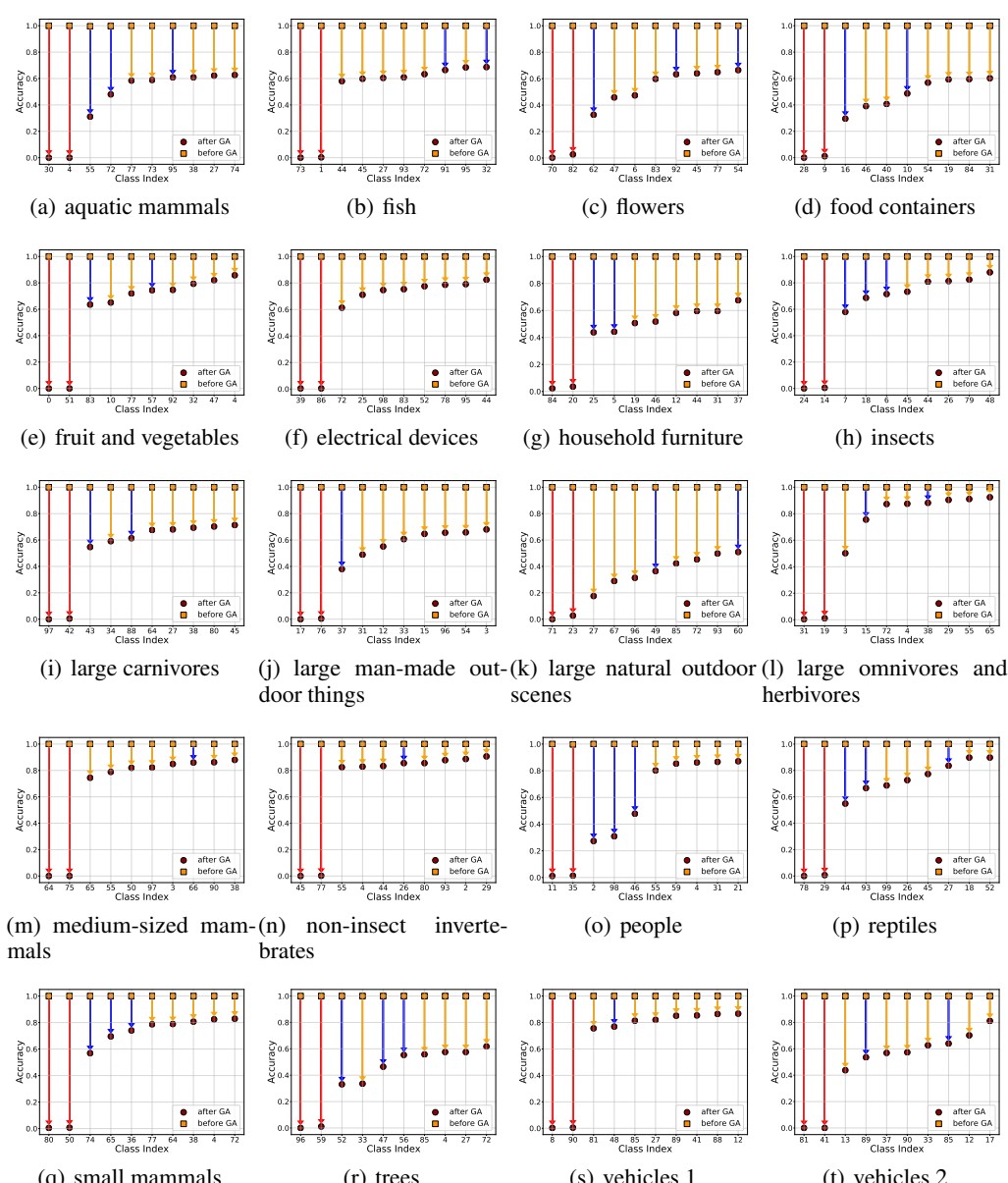

(a) aquatic mammals  (b) fish  (c) flowers  (d) food containers

(e) fruit and vegetables  (f) electrical devices  (g) household furniture  (h) insects

(i) large carnivores  (j) large man-made outdoor things  (k) large natural outdoor scenes  (l) large omnivores and herbivores

(m) medium-sized mammals  (n) non-insect invertebrates  (o) people  (p) reptiles

(q) small mammals  (r) trees  (s) vehicles 1  (t) vehicles 2

Target identification results with different unlearning requests and the minimum identified forgetting data on the CIFAR-100 dataset. Note that some target concepts are not successfully identified by the identified data.

Figure 14: Task Identification using the CIFAR-100 dataset for target mismatch forgetting.

we can also utilize the distillation objective to encourage the unlearned model's output to be far away from (or close to) the original ones. With the guide of model prediction, the data belonging to the same superclass with the forgetting data can be figured out to constrain the unlearning target. In Table 18, we present the results of our methods when only the given forgetting data are labeled, demonstrating our framework can be extended to achieve satisfactory performance.

## F.5 FORGETTING IN THE LARGE-SCALE DATASET

In this part, we present more experiments conducted on large-scale dataset like ImageNet-1k in Table 19, and also unlearning multiple classes in the large-scale datasets in Table 20.

Table 17: Performance comparison in the all matched scenario when TARF with CE loss/KL divergence (refer to Eq. (16) with the original model for the retaining part.

| Type / $\mathcal{D}$ | Dataset | CIFAR-10 | | | | | | CIFAR-100 | | | | | |
|---|---|---|---|---|---|---|---|---|---|---|---|---|---|
| | Method / Metrics | UA | RA | TA | MIA | Gap↓ | TIME↓ | UA | RA | TA | MIA | Gap↓ | TIME↓ |
| | Retrained (Ref.) | 0.00 | 99.51 | 94.69 | 100.00 | - | 43.3 | 0.00 | 97.85 | 76.03 | 100.00 | - | 43.2 |
| | FT [58] | 1.07 | 98.62 | 92.36 | 100.00 | 1.07 | 4.43 | 0.67 | 96.32 | 72.34 | 100.00 | 1.47 | 5.02 |
| Semi-supervised | SCRUB [37] | 0.00 | 99.94 | 91.00 | 100.00 | 1.03 | 2.88 | 0.00 | 99.98 | 76.75 | 100.00 | 0.71 | 3.23 |
| Scenarios | **TARF** (with CE) | 0.00 | 98.23 | 91.95 | 100.00 | 1.01 | 4.21 | 0.00 | 96.90 | 72.53 | 100.00 | 1.11 | 4.68 |
| | **TARF** (with KL) | 0.00 | 98.81 | 93.33 | 100.00 | **0.52** | 4.32 | 0.00 | 96.95 | 74.98 | 100.00 | **0.49** | 4.89 |

Table 18: A case study (%) on the unlearning on CIFAR-100 under the weakly-supervised scenario (e.g., using the pseudo-label generated by model prediction to handle unlabeled retaining data).

| Type / $\mathcal{D}$ | Dataset | Model mismatch | | | | | | Data mismatch | | | | | |
|---|---|---|---|---|---|---|---|---|---|---|---|---|---|
| | Method / Metrics | UA | RA | TA | MIA | Gap↓ | TIME↓ | UA | RA | TA | MIA | Gap↓ | TIME↓ |
| | Retrained | 88.22 | 98.58 | 78.50 | 25.78 | - | 43.8 | 0.00 | 98.50 | 80.15 | 100.00 | - | 53.2 |
| | FT | 92.67 | 95.02 | 79.34 | 16.33 | 4.58 | 4.86 | 82.62 | 95.66 | 79.77 | 37.24 | 37.15 | 4.93 |
| | RL | 80.11 | 95.83 | 79.83 | 99.00 | 21.35 | 4.93 | 89.78 | 96.82 | 79.90 | 70.76 | 30.49 | 4.97 |
| Semi-supervised | GA | 6.78 | 94.83 | 76.96 | 97.78 | 39.68 | 0.06 | 6.00 | 97.65 | 79.23 | 98.04 | 2.43 | 0.05 |
| Scenarios | BS | 18.11 | 95.90 | 72.28 | 95.22 | 37.14 | 0.89 | 15.38 | 98.50 | 72.28 | 96.22 | 6.76 | 0.96 |
| | $L_1$-sparse | 82.11 | 85.17 | 75.22 | 20.00 | 7.15 | 5.00 | 84.53 | 85.13 | 75.22 | 17.02 | 46.45 | 5.03 |
| | **TARF** (full labels) | 86.67 | 97.05 | 80.07 | 26.00 | **1.21** | 4.81 | 0.00 | 95.01 | 78.98 | 100.00 | **1.17** | 4.78 |
| | **TARF** (unlabeled retain) | 90.22 | 96.58 | 80.01 | 22.54 | 2.17 | 4.84 | 1.33 | 95.30 | 78.12 | 99.34 | 1.45 | 4.85 |

## F.6 Forgetting with Different Model Structures

In this part, we further check the unlearning performance of our TARF on different pre-trained model structures compared with several baselines. We choose CIFAR-100 as the pre-training classification task and conduct all matched forgetting and model mismatch forgetting. The results are summarized in Table 21. The results validate that our TARF can robustly achieve better unlearning performance across different model structures.

## F.7 Full Results with Different Forgetting Tasks

In this section, we provide the full results of Table 2, which is conducted by setting different random seeds (for multiple runs) with the original trails and reported as the mean and std values for each evaluation metric. Tables 22 to 25 presents the performance of unlearning on CIFAR-10, and Tables 26 to 29 presents the performance of unlearning on CIFAR-100. The performance comparison of our TARF with other baseline across the four forgetting tasks (i.e., all matched, target mismatch, model mismatch, and data mismatch) demonstrated the general effectiveness of our algorithm framework.

## G Broader Impact

In this work, we explore the label domain mismatch in class-wise unlearning, which aims to enhance the flexibility of data regulation with the increasing concern about the trustworthiness of machine learning. Pushing forward the practical usage of machine unlearning, our research provides a broader consideration of real-world unlearning scenarios and offers significant positive social impacts. It can enhance data privacy protection by allowing individuals to effectively remove their data, ensuring some sensitive data is not used for analysis. In addition, unlearning can remove bias or discrimination by correcting flawed datasets, promoting the development of fairness or other ethical considerations. This feature also enables enterprises to adhere to data protection standards such as GDPR [49] and CCPA [48], therefore promoting confidence among users. Our newly introduced unlearning setting, which decouples the class label and the target concept, is more general and discusses the achievability of various unlearning requests, which may often be different from the taxonomy of pre-training tasks.

Although we take a step forward in more practical class-wise unlearning by considering the label domain mismatch scenarios, it is not the end of this direction and there are still many problems to be addressed. Following the previous works [58, 17, 30, 6], our work mainly focuses on the class-wise unlearning with the classification model for the exploration, future efforts can also be paid in the unlearning problem of the emerging and powerful generative models. On the technical level, although those compared unlearning methods and our framework can achieve the forgetting target, it all requires extra computational cost, and how to make it more efficient can be further studied.

Table 19: Results (%). Comparison with the unlearning baselines on ImageNet-1k. All matched forgetting: unlearn 1 class; Target mismatch forgetting: unlearn three classes belonging to "fish".

| Type / $\mathcal{D}$ | Dataset | All matched | | | | | | Target mismatch | | | | | |
|---|---|---|---|---|---|---|---|---|---|---|---|---|---|
| | Method / Metrics | UA | RA | TA | MIA | Gap↓ | TIME↓ | UA | RA | TA | MIA | Gap↓ | TIME↓ |
| | Retrained | 0.00 | 79.77 | 77.64 | 100.00 | - | 7075.48 | 0.00 | 80.09 | 77.54 | 100.00 | - | 7777.54 |
| | FT | 0.00 | 70.18 | 71.98 | 100.00 | 3.82 | 608.11 | 0.79 | 70.26 | 72.07 | 100.00 | 4.02 | 608.62 |
| | RL [56] | 81.38 | 70.22 | 71.79 | 19.46 | 44.29 | 969.44 | 79.69 | 69.98 | 71.77 | 23.03 | 43.14 | 972.02 |
| | GA | 0.00 | 66.25 | 67.36 | 100.00 | 5.95 | 8.76 | 0.00 | 31.21 | 37.74 | 0.00 | 47.17 | 17.38 |
| | BS | 0.00 | 31.15 | 36.33 | 100.00 | 22.48 | 9.03 | 0.00 | 21.57 | 27.56 | 99.97 | 27.13 | 23.75 |
| | $L_1$-sparse | 0.00 | 67.98 | 70.70 | 100.00 | 4.68 | 603.21 | 0.00 | 67.24 | 70.28 | 100.00 | 5.03 | 601.27 |
| | SCRUB | 29.77 | 74.92 | 75.66 | 81.77 | 13.71 | 655.42 | 22.44 | 74.87 | 75.60 | 82.77 | 11.71 | 681.53 |
| | **TARF** (ours) | 0.00 | 70.53 | 72.23 | 100.00 | **3.66** | 600.11 | 0.00 | 69.93 | 71.79 | 100.00 | **3.97** | 628.87 |
| ImageNet-1k | Dataset | Model matched | | | | | | Data mismatch | | | | | |
| | Method / Metrics | UA | RA | TA | MIA | Gap↓ | TIME↓ | UA | RA | TA | MIA | Gap↓ | TIME↓ |
| | Retrained | 79.15 | 80.00 | 70.29 | 25.69 | - | 6501.27 | 0.00 | 80.36 | 70.38 | 100.00 | - | 6493.16 |
| | FT [58] | 83.31 | 70.38 | 64.05 | 19.00 | 6.68 | 695.42 | 0.00 | 69.99 | 63.76 | 100.00 | 4.24 | 693.18 |
| | RL [56] | 87.62 | 69.43 | 63.26 | 15.23 | 9.13 | 959.84 | 88.21 | 70.33 | 63.81 | 12.21 | 48.15 | 956.13 |
| | GA [28] | 0.00 | 66.62 | 58.91 | 100.00 | 44.56 | 17.44 | 0.00 | 15.35 | 14.34 | 0.00 | 55.26 | 17.58 |
| | BS [6] | 0.00 | 45.81 | 40.84 | 100.00 | 54.28 | 19.69 | 0.00 | 13.00 | 12.10 | 100.00 | 31.41 | 23.70 |
| | $L_1$-sparse [30] | 82.00 | 67.94 | 62.58 | 19.15 | 7.29 | 1091.29 | 0.00 | 66.37 | 61.03 | 100.00 | 5.84 | 1071.41 |
| | SCRUB [37] | 86.08 | 74.82 | 68.04 | 14.69 | 6.34 | 663.61 | 14.18 | 74.84 | 67.92 | 93.10 | 7.27 | 689.82 |
| | **TARF** (ours) | 80.62 | 70.27 | 64.04 | 19.46 | **5.92** | 601.28 | 0.00 | 70.10 | 63.97 | 100.00 | **4.17** | 602.62 |

Table 20: Results (%). Comparison with the unlearning baselines on TinyImageNet-200/ImageNet-1k with more (10+) forgetting classes in all matched forgetting scenarios.

| Scenarios / $\mathcal{D}$ | Unlearn request | forget 10 classes in Tiny-ImageNet | | | | | | forget 30 classes in Tiny-ImageNet | | | | | |
|---|---|---|---|---|---|---|---|---|---|---|---|---|---|
| | Method / Metrics | UA | RA | TA | MIA | Gap↓ | TIME↓ | UA | RA | TA | MIA | Gap↓ | TIME↓ |
| | Retrained | 0.00 | 71.00 | 60.29 | 100.00 | - | 251.43 | 0.00 | 65.26 | 57.60 | 100.00 | - | 181.13 |
| | FT | 2.04 | 70.63 | 59.04 | 98.26 | 1.35 | 27.10 | 2.79 | 72.41 | 60.36 | 97.38 | 3.71 | 35.00 |
| | GA | 17.76 | 61.74 | 56.12 | 76.90 | 13.57 | 1.37 | 28.95 | 59.72 | 57.54 | 57.06 | 19.37 | 3.49 |
| | **TARF** (ours) | 0.00 | 69.63 | 59.69 | 100.00 | **0.49** | 28.5 | 0.00 | 70.24 | 60.16 | 100.00 | **1.89** | 39.6 |
| All matched Forgetting | Unlearn request | forget 50 classes in Tiny-ImageNet | | | | | | forget 10 classes in ImageNet | | | | | |
| | Method / Metrics | UA | RA | TA | MIA | Gap↓ | TIME↓ | UA | RA | TA | MIA | Gap↓ | TIME↓ |
| | Retrained | 0.00 | 66.26 | 57.88 | 100.00 | - | 161.37 | 0.00 | 51.94 | 56.74 | 100.00 | - | 917.66 |
| | FT | 5.19 | 75.77 | 61.29 | 85.75 | 8.09 | 44.62 | 0.00 | 55.16 | 59.53 | 100.00 | 1.50 | 316.14 |
| | GA | 22.92 | 44.12 | 48.03 | 62.26 | 23.16 | 7.70 | 5.73 | 47.35 | 52.42 | 87.21 | 6.85 | 2.18 |
| | **TARF** (ours) | 0.00 | 71.68 | 60.89 | 100.00 | **2.11** | 46.97 | 0.00 | 50.69 | 55.83 | 100.00 | **0.54** | 353.69 |

Table 21: Results (%) of unlearning with different model structure. All methods are trained on the same backbone, i.e., the basis of unlearning initialization is the same (except for retraining from scratch). Values are percentages. Bold numbers are superior results. ↓ indicates smaller are better.

| CIFAR-100 | Task | All matched | | | | | Model mismatch | | | | |
|---|---|---|---|---|---|---|---|---|---|---|---|
| | Metric | UA | RA | TA | MIA | Gap↓ | UA | RA | TA | MIA | Gap↓ |
| VGG-19 | Retrained | 0.00 | 97.26 | 73.13 | 100.00 | - | 87.44 | 98.22 | 82.12 | 19.89 | - |
| | FT [58] | 0.00 | 90.92 | 66.86 | 100.00 | 3.15 | 95.22 | 95.17 | 77.71 | 7.56 | 6.89 |
| | RL [56] | 0.00 | 90.29 | 66.16 | 100.00 | 3.48 | 96.22 | 95.26 | 77.71 | 98.56 | 23.71 |
| | GA [28] | 0.00 | 79.27 | 56.03 | 100.00 | 8.77 | 0.00 | 93.09 | 74.30 | 100.00 | 45.13 |
| | **TARF** (ours) | 0.00 | 91.96 | 67.94 | 100.00 | **2.62** | 82.67 | 93.71 | 76.24 | 24.22 | **4.87** |
| ResNet-18 | Retrained | 0.00 | 97.85 | 76.03 | 100.00 | - | 88.22 | 98.58 | 78.50 | 25.78 | - |
| | FT [58] | 0.66 | 96.55 | 71.97 | 100.00 | 1.51 | 98.22 | 96.79 | 80.14 | 6.78 | 8.11 |
| | RL [56] | 0.11 | 95.90 | 71.57 | 100.00 | 1.63 | 94.11 | 96.70 | 80.17 | 96.89 | 20.14 |
| | GA [28] | 1.89 | 95.26 | 69.14 | 99.89 | 2.87 | 9.33 | 95.13 | 77.22 | 96.89 | 38.68 |
| | **TARF** (ours) | 0.00 | 96.90 | 71.51 | 100.00 | **1.37** | 86.00 | 96.54 | 74.20 | 22.78 | **2.89** |
| WideResNet | Retrained | 0.00 | 97.71 | 76.95 | 100.00 | - | 88.11 | 98.37 | 83.61 | 23.56 | - |
| | FT [58] | 0.67 | 96.61 | 71.29 | 100.00 | 1.86 | 97.44 | 95.70 | 78.70 | 7.33 | 8.29 |
| | RL [56] | 0.00 | 95.86 | 71.36 | 100.00 | 1.86 | 85.77 | 94.69 | 78.26 | 96.00 | 20.95 |
| | GA [28] | 0.44 | 91.49 | 66.29 | 100.00 | 2.26 | 4.33 | 91.76 | 75.18 | 99.11 | 43.71 |
| | **TARF** (ours) | 0.00 | 96.51 | 71.77 | 100.00 | **1.60** | 88.00 | 95.50 | 79.06 | 22.67 | **2.11** |

Table 22: Main Results (%). Comparison with the unlearning baselines. All methods are trained on the same backbone, i.e., the basis of unlearning initialization is the same (except for retraining from scratch). Values are percentages. Bold numbers are superior results. ↓ indicates smaller are better.

| CIFAR-10 | Metric | UA | | RA | | TA | | MIA | | Gap↓ | |
|---|---|---|---|---|---|---|---|---|---|---|---|
| | Method | mean | std | mean | std | mean | std | mean | std | mean | std |
| All matched | Retrained | 0.00 | - | 99.51 | - | 94.69 | - | 100.00 | - | - | - |
| | FT [58] | 4.66 | 3.59 | 98.58 | 0.04 | 92.42 | 0.06 | 100.00 | 0.00 | 1.96 | 0.89 |
| | RL [56] | 2.23 | 1.90 | 98.30 | 0.65 | 91.97 | 0.74 | 100.00 | 0.00 | 1.54 | 0.82 |
| | GA [28] | 0.34 | 0.16 | 95.48 | 0.24 | 88.52 | 0.35 | 99.88 | 0.10 | 2.67 | 0.21 |
| | IU [29] | 0.11 | 0.05 | 72.50 | 15.65 | 68.28 | 14.10 | 99.98 | 0.02 | 13.39 | 7.41 |
| | BS [6] | 24.72 | 0.32 | 88.91 | 0.97 | 81.84 | 0.94 | 89.23 | 0.56 | 14.74 | 0.70 |
| | $L_1$-sparse [30] | 0.00 | 0.00 | 94.18 | 0.03 | 90.01 | 0.24 | 100.00 | 0.00 | 2.50 | 0.05 |
| | SalUn [11] | 0.48 | 0.46 | 88.66 | 2.67 | 84.48 | 2.40 | 100.00 | 0.00 | 5.39 | 1.38 |
| | SCRUB [37] | 1.23 | 0.58 | 99.92 | 0.02 | 91.23 | 0.56 | 100.00 | 0.00 | 1.28 | 0.23 |
| | **TARF** (ours) | 0.00 | 0.00 | 98.22 | 0.02 | 92.09 | 0.14 | 100.00 | 0.00 | **0.97** | 0.03 |

Table 23: Main Results (%). Comparison with the unlearning baselines. All methods are trained on the same backbone, i.e., the basis of unlearning initialization is the same (except for retraining from scratch). Values are percentages. Bold numbers are superior results. ↓ indicates smaller are better.

| CIFAR-10 | Metric | UA | | RA | | TA | | MIA | | Gap↓ | |
|---|---|---|---|---|---|---|---|---|---|---|---|
| | Method | mean | std | mean | std | mean | std | mean | std | mean | std |
| Model mismatch | Retrained | 87.76 | - | 99.58 | - | 95.91 | - | 20.57 | - | - | - |
| | FT [58] | 94.78 | 0.11 | 98.65 | 0.12 | 93.77 | 0.21 | 10.42 | 0.86 | 5.06 | 0.27 |
| | RL [56] | 48.25 | 5.43 | 98.01 | 0.12 | 93.03 | 0.21 | 98.10 | 0.64 | 30.37 | 1.53 |
| | GA [28] | 6.49 | 0.73 | 86.91 | 0.08 | 82.03 | 0.18 | 94.39 | 0.59 | 45.41 | 0.27 |
| | IU [29] | 15.84 | 7.86 | 85.89 | 1.45 | 81.08 | 1.49 | 93.58 | 3.71 | 43.36 | 3.62 |
| | BS [6] | 14.05 | 3.76 | 53.28 | 2.51 | 51.25 | 1.86 | 94.90 | 1.06 | 59.75 | 2.29 |
| | $L_1$-sparse [30] | 92.25 | 0.87 | 95.01 | 0.25 | 91.67 | 0.04 | 17.40 | 2.86 | 4.14 | 1.00 |
| | SalUn [11] | 16.31 | 7.40 | 92.91 | 1.05 | 86.50 | 2.12 | 99.24 | 0.09 | 41.55 | 2.14 |
| | SCRUB [37] | 93.21 | 1.17 | 99.83 | 0.13 | 93.29 | 0.81 | 14.24 | 0.87 | 3.65 | 0.18 |
| | **TARF** (ours) | 89.91 | 1.20 | 97.73 | 0.24 | 92.66 | 0.17 | 20.36 | 2.54 | **2.45** | 0.46 |

Table 24: Main Results (%). Comparison with the unlearning baselines. All methods are trained on the same backbone, i.e., the basis of unlearning initialization is the same (except for retraining from scratch). Values are percentages. Bold numbers are superior results. ↓ indicates smaller are better.

| CIFAR-10 | Metric | UA | | RA | | TA | | MIA | | Gap↓ | |
|---|---|---|---|---|---|---|---|---|---|---|---|
| | Method | mean | std | mean | std | mean | std | mean | std | mean | std |
| | Retrained | 0.00 | - | 99.38 | - | 93.85 | - | 100.00 | - | - | - |
| | FT [58] | 52.23 | 1.80 | 98.43 | 0.05 | 91.74 | 0.09 | 50.59 | 0.15 | 26.18 | 0.40 |
| | RL [56] | 50.63 | 0.62 | 98.21 | 0.65 | 91.51 | 0.61 | 53.88 | 2.36 | 25.06 | 0.12 |
| | GA [28] | 41.64 | 0.82 | 97.05 | 0.04 | 89.68 | 0.17 | 63.23 | 1.10 | 21.23 | 0.43 |
| Target | IU [29] | 45.32 | 0.81 | 70.25 | 17.82 | 65.67 | 2.76 | 55.98 | 2.76 | 36.66 | 9.37 |
| mismatch | BS [6] | 53.78 | 0.16 | 89.67 | 1.02 | 79.34 | 3.95 | 66.31 | 10.02 | 25.36 | 3.28 |
| | $L_1$-sparse [30] | 49.55 | 0.08 | 93.57 | 0.05 | 89.06 | 0.23 | 51.33 | 0.09 | 27.21 | 0.05 |
| | SalUn [11] | 47.85 | 1.22 | 87.84 | 3.25 | 83.38 | 2.94 | 58.10 | 2.85 | 27.40 | 1.10 |
| | SCRUB [37] | 48.53 | 1.02 | 99.43 | 0.21 | 91.66 | 0.28 | 51.27 | 0.73 | 24.92 | 0.51 |
| | **TARF** (ours) | 0.05 | 0.02 | 97.65 | 0.08 | 91.28 | 0.47 | 100.00 | 0.00 | **1.09** | 0.14 |

Table 25: Main Results (%). Comparison with the unlearning baselines. All methods are trained on the same backbone, i.e., the basis of unlearning initialization is the same (except for retraining from scratch). Values are percentages. Bold numbers are superior results. ↓ indicates smaller are better.

| CIFAR-10 | Metric | UA | | RA | | TA | | MIA | | Gap↓ | |
|---|---|---|---|---|---|---|---|---|---|---|---|
| | Method | mean | std | mean | std | mean | std | mean | std | mean | std |
| | Retrained | 0.00 | - | 99.53 | - | 95.56 | - | 100.00 | - | - | - |
| | FT [58] | 96.85 | 0.06 | 98.62 | 0.13 | 93.47 | 0.21 | 6.93 | 0.45 | 48.23 | 0.18 |
| | RL [56] | 73.62 | 2.86 | 97.90 | 0.22 | 92.59 | 0.66 | 52.04 | 2.23 | 31.55 | 1.49 |
| | GA [28] | 9.82 | 1.13 | 96.14 | 0.28 | 90.46 | 0.33 | 90.46 | 0.95 | 6.56 | 0.67 |
| Data | IU [29] | 15.19 | 7.66 | 94.80 | 0.70 | 89.08 | 0.46 | 92.83 | 4.26 | 8.39 | 2.69 |
| mismatch | BS [6] | 16.72 | 0.02 | 61.01 | 0.21 | 53.81 | 4.05 | 93.47 | 1.24 | 25.88 | 1.27 |
| | $L_1$-sparse [30] | 95.42 | 0.35 | 94.57 | 0.26 | 91.07 | 0.01 | 10.82 | 1.30 | 48.51 | 0.47 |
| | SalUn [11] | 55.52 | 3.76 | 92.68 | 1.19 | 89.25 | 1.22 | 60.23 | 3.30 | 27.12 | 2.37 |
| | SCRUB [37] | 97.06 | 0.52 | 99.16 | 0.23 | 94.72 | 0.56 | 9.98 | 0.43 | 46.98 | 0.21 |
| | **TARF** (ours) | 0.00 | 0.00 | 98.35 | 0.18 | 93.42 | 0.34 | 100.00 | 0.00 | **0.83** | 0.13 |

Table 26: Main Results (%). Comparison with the unlearning baselines. All methods are trained on the same backbone, i.e., the basis of unlearning initialization is the same (except for retraining from scratch). Values are percentages. Bold numbers are superior results. ↓ indicates smaller are better.

| CIFAR-100 | Metric | UA | | RA | | TA | | MIA | | Gap↓ | |
|---|---|---|---|---|---|---|---|---|---|---|---|
| | Method | mean | std | mean | std | mean | std | mean | std | mean | std |
| | Retrained | 0.00 | - | 97.85 | - | 76.03 | - | 100.00 | - | - | - |
| | FT [58] | 0.67 | 0.01 | 96.44 | 0.12 | 72.16 | 0.19 | 100.00 | 0.00 | 1.49 | 0.02 |
| | RL [56] | 0.56 | 0.45 | 96.00 | 0.10 | 71.79 | 0.22 | 100.00 | 0.00 | 1.66 | 0.03 |
| | GA [28] | 1.61 | 0.28 | 95.00 | 0.26 | 68.85 | 0.29 | 99.89 | 0.00 | 2.93 | 0.07 |
| All matched | IU [29] | 0.00 | 0.00 | 39.80 | 2.19 | 31.09 | 1.51 | 100.00 | 0.00 | 25.75 | 0.93 |
| | BS [6] | 4.83 | 0.05 | 90.17 | 0.06 | 64.30 | 0.64 | 99.45 | 0.12 | 6.20 | 0.22 |
| | $L_1$-sparse [30] | 0.00 | 0.00 | 94.25 | 0.57 | 71.35 | 1.27 | 100.00 | 0.00 | 1.92 | 0.46 |
| | SalUn [11] | 0.00 | 0.00 | 77.00 | 1.66 | 63.06 | 0.92 | 100.00 | 0.00 | 8.46 | 0.64 |
| | SCRUB [37] | 0.00 | 0.00 | 99.72 | 0.26 | 76.69 | 0.06 | 100.00 | 0.00 | **0.64** | 0.08 |
| | **TARF** (ours) | 0.00 | 0.00 | 96.67 | 0.24 | 72.40 | 0.14 | 100.00 | 0.00 | 1.21 | 0.09 |

Table 27: Main Results (%). Comparison with the unlearning baselines. All methods are trained on the same backbone, i.e., the basis of unlearning initialization is the same (except for retraining from scratch). Values are percentages. Bold numbers are superior results. ↓ indicates smaller are better.

| CIFAR-100 | Metric | UA | | RA | | TA | | MIA | | Gap↓ | |
|---|---|---|---|---|---|---|---|---|---|---|---|
| | Method | mean | std | mean | std | mean | std | mean | std | mean | std |
| Model mismatch | Retrained | 88.22 | - | 98.58 | - | 78.50 | - | 25.78 | - | - | - |
| | FT [58] | 95.45 | 2.78 | 95.91 | 0.89 | 79.74 | 0.40 | 11.56 | 4.78 | 6.34 | 1.77 |
| | RL [56] | 87.11 | 7.00 | 96.27 | 0.44 | 80.00 | 0.17 | 97.95 | 1.06 | 20.75 | 0.61 |
| | GA [28] | 8.06 | 1.28 | 94.98 | 0.15 | 77.09 | 0.13 | 97.34 | 0.45 | 39.18 | 0.50 |
| | IU [29] | 39.95 | 5.28 | 97.22 | 0.39 | 79.71 | 0.63 | 83.28 | 3.17 | 27.08 | 2.05 |
| | BS [6] | 18.56 | 0.56 | 95.87 | 0.03 | 74.96 | 2.68 | 94.95 | 0.28 | 36.27 | 0.87 |
| | $L_1$-sparse [30] | 91.11 | 5.00 | 94.28 | 0.18 | 77.61 | 0.39 | 15.56 | 4.45 | 5.84 | 1.69 |
| | SalUn [11] | 74.78 | 8.45 | 79.98 | 1.14 | 71.55 | 0.77 | 65.61 | 11.39 | 19.71 | 5.44 |
| | SCRUB [37] | 92.45 | 2.80 | 99.44 | 0.78 | 78.75 | 1.75 | 20.13 | 4.56 | 4.14 | 1.15 |
| | **TARF (ours)** | 84.78 | 1.90 | 97.19 | 0.14 | 80.02 | 0.15 | 28.89 | 2.89 | **2.37** | 1.15 |

Table 28: Main Results (%). Comparison with the unlearning baselines. All methods are trained on the same backbone, i.e., the basis of unlearning initialization is the same (except for retraining from scratch). Values are percentages. Bold numbers are superior results. ↓ indicates smaller are better.

| CIFAR-100 | Metric | UA | | RA | | TA | | MIA | | Gap↓ | |
|---|---|---|---|---|---|---|---|---|---|---|---|
| | Method | mean | std | mean | std | mean | std | mean | std | mean | std |
| Target mismatch | Retrained | 0.00 | - | 97.85 | - | 73.72 | - | 100.00 | - | - | - |
| | FT [58] | 58.58 | 0.40 | 96.42 | 0.10 | 72.31 | 0.22 | 45.94 | 0.83 | 28.87 | 0.34 |
| | RL [56] | 57.76 | 1.14 | 96.00 | 0.10 | 72.04 | 0.16 | 50.67 | 3.69 | 27.66 | 1.15 |
| | GA [28] | 22.07 | 0.69 | 96.87 | 0.24 | 70.52 | 0.30 | 90.45 | 0.23 | 8.95 | 0.10 |
| | IU [29] | 30.80 | 0.18 | 39.44 | 2.25 | 31.00 | 1.42 | 63.83 | 0.14 | 42.03 | 0.91 |
| | BS [6] | 40.91 | 0.47 | 98.36 | 0.04 | 70.04 | 1.38 | 85.00 | 0.16 | 15.03 | 0.18 |
| | $L_1$-sparse [30] | 55.31 | 2.90 | 94.23 | 0.44 | 72.15 | 1.27 | 48.47 | 3.54 | 30.26 | 1.18 |
| | SalUn [11] | 43.29 | 1.60 | 77.15 | 1.63 | 63.30 | 0.93 | 64.63 | 1.34 | 27.45 | 0.10 |
| | SCRUB [37] | 59.56 | 0.09 | 99.74 | 0.26 | 76.14 | 0.82 | 45.45 | 0.56 | 29.60 | 0.02 |
| | **TARF (ours)** | 0.29 | 0.03 | 97.06 | 0.29 | 73.27 | 0.41 | 100.00 | 0.00 | **0.38** | 0.17 |

Table 29: Main Results (%). Comparison with the unlearning baselines. All methods are trained on the same backbone, i.e., the basis of unlearning initialization is the same (except for retraining from scratch). Values are percentages. Bold numbers are superior results. ↓ indicates smaller are better.

| CIFAR-100 | Metric | UA | | RA | | TA | | MIA | | Gap↓ | |
|---|---|---|---|---|---|---|---|---|---|---|---|
| | Method | mean | std | mean | std | mean | std | mean | std | mean | std |
| Data mismatch | Retrained | 0.00 | - | 98.50 | - | 80.15 | - | 100.00 | - | - | - |
| | FT [58] | 90.79 | 5.18 | 96.19 | 0.52 | 79.80 | 0.03 | 20.46 | 16.78 | 43.25 | 5.10 |
| | RL [56] | 93.60 | 3.82 | 96.32 | 0.39 | 79.92 | 0.02 | 65.20 | 5.56 | 32.73 | 2.24 |
| | GA [28] | 6.98 | 0.98 | 97.78 | 0.14 | 79.34 | 0.11 | 97.53 | 0.51 | 2.75 | 0.31 |
| | IU [29] | 37.22 | 5.71 | 99.17 | 0.21 | 80.01 | 1.81 | 85.41 | 2.41 | 13.54 | 2.08 |
| | BS [6] | 15.71 | 0.33 | 98.47 | 0.04 | 76.02 | 3.74 | 96.05 | 0.18 | 5.86 | 0.18 |
| | $L_1$-sparse [30] | 89.02 | 4.67 | 94.18 | 0.05 | 78.89 | 0.20 | 18.67 | 4.36 | 41.64 | 2.20 |
| | SalUn [11] | 79.00 | 6.07 | 79.92 | 1.05 | 71.55 | 0.51 | 44.18 | 9.96 | 39.42 | 3.62 |
| | SCRUB [37] | 93.28 | 2.10 | 99.25 | 0.98 | 79.18 | 0.48 | 18.45 | 3.55 | 46.13 | 2.37 |
| | **TARF (ours)** | 0.00 | 0.00 | 95.80 | 0.79 | 79.55 | 0.57 | 100.00 | 0.00 | **1.61** | 0.05 |

