# OpenReview forum: "Decoupling the Class Label and the Target Concept in Machine Unlearning"
_ICLR.cc/2025/Conference — Submitted to ICLR 2025_

### Official Review · Reviewer_GTmm · 2024-10-23

**Soundness:** 3
**Presentation:** 3
**Contribution:** 3
**Rating:** 8
**Confidence:** 4

**Summary:**

This paper proposes a TARget-aware Forgetting (TARF) framework for machine unlearning, which decouples the class label and the target concept and investigates three problems beyond conventional all-matched forgetting. The authors conduct experiments on various datasets to demonstrate the effectiveness of TARF.

**Strengths:**

1. The authors introduce new settings in machine unlearning by decoupling the class label and the target concept, and investigate the label domain mismatch in class-wise unlearning, which makes sense and contributes to the community of machine unlearning.

2. The proposed TARF framework is novel in addressing the problem of three typical machine unlearning mismatch (target mismatch, model mismatch, data mismatch), and has the potential to improve the flexibility and practicality of machine unlearning.

3. The experimental results are comprehensive and show that TARF outperforms several baseline methods in different unlearning scenarios. The experiments cover a wide range of datasets and models, which adds to the credibility of the results.

4. This work systematically reveals the challenges of restrictive unlearning with the mismatched label domains, and give a new insight into machine unlearning that the representation gravity in forgetting dynamics is critical for achieving the forgetting target in the new tasks.

**Weaknesses:**

1. The keyword "gravity" is relatively unfamiliar to readers. It should be introduced as much as possible in advance or given a brief introduction to enhance the readability of the article. This would help readers better understand the context and significance of the term within the machine unlearning framework presented in the paper.

2. Under the three proposed mismatch scenarios, the metrics for measuring the algorithm's effectiveness are not very clear. For example, in the case of model mismatch where the model only outputs superclasses, it is not clear how the accuracy on the unlearning set is calculated. Appendix C only introduces the specific implementation of the baselines and seems to have omitted the specific calculation method of the metrics. This lack of clarity makes it difficult for readers to fully understand and evaluate the performance of the proposed method in different scenarios. It is recommended that the authors provide a more detailed and explicit explanation of how these metrics are calculated in each scenario to enhance the transparency and reproducibility of the research.

**Questions:**

1. The article proposes three mismatch scenarios and then establishes a unified framework. What are the commonalities among the three mismatch scenarios? Please give more explanations about how to build a unified framework to deal with these scenarios.

---

> ### Author Response · Authors · 2024-11-20
> **Response to Reviewer GTmm [1/2]**
>
> Sincerely thank you for your supportive review and your constructive suggestions. Here are our detailed replies to your comments and questions.
>
> > **W1:** The keyword "gravity" is relatively unfamiliar to readers. It should be introduced as much as possible in advance or given a brief introduction to enhance the readability of the article. This would help readers better understand the context and significance of the term within the machine unlearning framework presented in the paper.
>
> **A1:** Thank you for the constructive comments! We will follow the reviewer's advice to add its explanation as early as possible for clarity.
>
> Generally, the keyword **"gravity", originating in physics**, refers to **a fundamental force of nature that causes objects with mass to attract each other.** It is a significant interaction between objects at the macroscopic scale and determines the motion of planets, stars, and galaxies. In machine learning, **data points can be regarded as having "attraction" to each other** based on their similarity, and the closer points have strong influence.
>
> **In our research framework**, we use **"gravity"** to serve as **an intuitive analogy and inspiration** for **describing the mutual influence/"attraction" between two parts of data**, specializing in the learning dynamics of model behavior (e.g., forgetting degree) on these two parts during the forgetting phase. Intuitively, the **strong "gravity" refers to a strong interaction force**, making the model behavior change on the one part be similar to the other part, while the **weak "gravity" refers to a weak interaction force**, making the model behavior change on the one part of data has limited effect on the other one. It is observed to be affected by the representation similarity of two data parts, as demonstrated in our Figure 3.
>
> > **W2:** Under the three proposed mismatch scenarios, the metrics for measuring the algorithm's effectiveness are not very clear. For example, in the case of model mismatch where the model only outputs superclasses, it is not clear how the accuracy on the unlearning set is calculated. Appendix C only introduces the specific implementation of the baselines and seems to have omitted the specific calculation method of the metrics. This lack of clarity makes it difficult for readers to fully understand and evaluate the performance of the proposed method in different scenarios. It is recommended that the authors provide a more detailed and explicit explanation of how these metrics are calculated in each scenario to enhance the transparency and reproducibility of the research.
>
> **A2:** Thank you for pointing this out! Generally, in the evaluation phase, **we adopt the same labels used in pre-training to measure the unlearned model**.  For example, the unlearn accuracy is calculated on the data belonging to the target concept to be forgotten. **In the model mismatch forgetting**, as the model is trained with superclass labels, **the Unlearning Accuracy (UA) is calculated using the superclass label**.
>
> To give a detailed explanation, we summarize the following list and tables of the metrics (following the previous work [1,2]) and adopted labels in different scenarios, as well as an example in CIFAR-100,
>
> - Unlearning Accuracy **(UA)**: the accuracy of the unlearned model $\theta_\text{u}$ on the dataset of target concept $D_\text{t}$.
> - Retaining  Accuracy **(RA)**:  the accuracy of the unlearned model $\theta_\text{u}$ on retaining dataset $D_\text{r}$.
> - Testing Accuracy **(TA)**: the accuracy of the unlearned model $\theta_\text{u}$ on test dataset $D_\text{test}$ excluding the data belonging to the target concept to be forgotten.
> - Model Inversion Attack **(MIA)**:  the MIA success rate by a confidence-based MIA predictor of the model $\theta_\text{u}$ on the dataset of target concept $D_\text{t}$. We follow [1] to implement it to find how many samples in $D_\text{t}$ can be correctly predicted as non-training sample by the MIA predictor against $\theta_\text{u}$. Specifically, we sample a balanced dataset from the retaining dataset $D_\text{r}$ and the test dataset excluding the forgetting data to train the MIA predictor, then it is used to calculate the rate of true negative predictions on $D_\text{t}$.

---

> ### Author Response · Authors · 2024-11-20
> **Response to Reviewer GTmm [2/2]**
>
> **Table 1. The label used in evaluation metrics on different forgetting scenarios.**
>
> | Used Label           | All Matched | Target Mismatch | Model Mismatch | Data Mismatch |
> | -------------------- | ----------- | --------------- | -------------- | ------------- |
> | UA ($D_\text{t}$)    | Class       | Class           | Superclass     | Superclass    |
> | RA ($D_\text{r}$)    | Class       | Class           | Superclass     | Superclass    |
> | TA ($D_\text{test}$) | Class       | Class           | Superclass     | Superclass    |
> | MIA ($D_\text{t}$)   | Class       | Class           | Superclass     | Superclass    |
>
> **Table 2. The evaluation data (label number) of different forgetting scenarios with CIFAR-100.**
>
> | CIFAR-100 Example    | All Matched        | Target Mismatch                           | Model Mismatch                                               | Data Mismatch           |
> | -------------------- | ------------------ | ----------------------------------------- | ------------------------------------------------------------ | ----------------------- |
> | UA ($D_\text{t}$)    | "boy", "girl" (2)  | "boy", "girl", "man", "woman", "baby" (5) | part of "people" (1), which is data of "boy" and "girl" but with superclass label. | "people" (1)            |
> | RA ($D_\text{r}$)    | Other classes (98) | Other classes (95)                        | other part of "people" (1) with the rest superclasses (19)   | Other superclasses (19) |
> | TA ($D_\text{test}$) | Other classes (98) | Other classes (95)                        | other part of "people" (1) with the rest superclasses (19)   | Other superclasses (19) |
> | MIA ($D_\text{t}$)   | "boy", "girl" (2)  | "boy", "girl", "man", "woman", "baby" (5) | part of "people" (1), which is data of "boy" and "girl" but with superclass label. | "people" (1)            |
>
> We will add the above in our revised version to enhance the metric explanation.
>
> [1] Model Sparsity Can Simplify Machine Unlearning. NeurIPS 2023.
>
> [2] SalUn: Empowering Machine Unlearning via Gradient-based Weight Saliency in Both Image Classification and Generation. ICLR 2024.
>
> > **Q1:** The article proposes three mismatch scenarios and then establishes a unified framework. What are the commonalities among the three mismatch scenarios? Please give more explanations about how to build a unified framework to deal with these scenarios.
>
> **A3:** Thanks for the thoughtful questions! **The three mismatch scenarios**, i.e., target mismatch forgetting, data mismatch forgetting, and model mismatch forgetting, **share the common challenge on representation mismatch** between the pre-trained model, the identified forgetting data, and the target concept to be forgotten. It breaks the assumption in all-matched scenarios that the three are matched and can result in extra-/ineffective- forgetting in unlearning tasks. **Like the intuitive illustration in Figure 2**, in the target/data mismatch forgetting, the target concept can be wider than the identified forgetting data; while in the model mismatch forgetting, it can be smaller than the coarse-grained model representation.
>
> To build a unified framework like TARF, **it requires considering** the aforementioned **two issues**, i.e., **insufficient representation** in target/data mismatch, and **decomposition lacking** in the model mismatch. The former requires a flexible controller for forgetting strength while the latter requires a simultaneous consideration on forgetting and retaining. Thus, **based on the general equation in Eq. (5), we set two sub-objectives** (annealed forgetting and target-aware retaining) to **decompose the learned representation** and **control the forgetting strength** by the instance-wise weighting mechanism which selects the targeted-aware forgetting data. Then, TARF becomes a unified framework for the three mismatched scenarios. Note that in our presentation, TARF is illustrated with three phases to better explain its functionality, while it is not three independent parts but unified in a general objective with specific parameters.

---

> ### Author Response · Authors · 2024-11-22
>
> Dear Reviewer GTmm,
>
> Thank you for your great efforts in reviewing our work and for your valuable comments.
>
> In the rebuttal, we have tried our best to address the concerns, and provided detailed responses as well as the revision summarization. Would you mind checking our responses and confirming if there is any unclear point so that we should/could further clarify?
>
> Best regards,
>
> Authors of Submission 4171

---

> ### Author Response · Authors · 2024-11-24
> **Looking forward Further Reply**
>
> Dear Reviewer GTmm,
>
> Thank you very much for your time and insightful comments. We have provided detailed responses and conducted the revision (summarized in General Response) following your valuable suggestions.
>
> As the discussion phase is approaching to end, would you mind confirming if there is any additional questions or concerns regarding our response? We will do our best to address them promptly. Additionally, if you find our responses helpful, we would also greatly appreciate it if you could acknowledge the efforts we have made.
>
> Best regards,
>
> Authors of Submission 4171

---

> > ### Comment · Reviewer_GTmm · 2024-11-25
> >
> > Thank you for your response! My concern has been addressed.

---

> > > ### Author Response · Authors · 2024-11-25
> > > **Thank you for the positive acknowledgment in the feedback!**
> > >
> > > We sincerely appreciate Reviewer GTmm's recognition and support of our work! We are glad to hear that our response addressed your concerns and will make sure to incorporate all suggestions in the revision.

---

### Official Review · Reviewer_GXDf · 2024-10-30

**Soundness:** 3
**Presentation:** 2
**Contribution:** 2
**Rating:** 3
**Confidence:** 3

**Summary:**

The paper presents a new formalization of unlearning based on label mismatches between ground truth, the learned hypotheses, and the unlearning data, define new unlearning problems as a result of this formalization, and present an algorithm and results for these problems.

**Strengths:**

- The developed viewpoint that the forgetting data may have their own labeling is nice, and I think important.

**Weaknesses:**

- The problems appear a bit artificial to me. All explanations are based on the concept hierarchy in the CIFAR data. I miss a clear real-world motivation for the setting. For example, I would have thought of a problem as follows: Given a learned classifier from CIFAR, remove all images that have a certain characteristics (e.g., remove all images from copyrighted sources, etc.). This obviously has real-world implications.
I can imagine that this can be formulated in the setting studied by the authors, but it is not clear how, because they always refer to existing classes (and there is no class for "images from copyrighted sources", I guess). Probably what they do is fine for an evaluation, but one should provide a clearer motivation, and then discuss why the restricted CIFAR setting can be used for evaluating the setting discussed in the motivation.

- I really had a hard time in understanding the < notation (this should be a curly <). Maybe this is for similar reasons as discussed above, because I was really looking for an intuitive understanding, not for an explanation that specifies in what way that affects the CIFAR data. Also terms like "false retaining set" and "true retaining set" I found hard to grasp, even though they are formally defined.

- The proposed method has three phases, which are apparently applied sequentially. Even though there is a blended phase in the middle, I would expect that a somewhat tighter integration of these phases would be better.

MINOR CONCERNS:

- I don't think that computation is the main problem with "exact unlearning", it is more that you have to have access to all the training data, i.e., you cannot simply store the model and unlearn new data from it.

- Table 1 is too small. Some of the subscripts cannot be discriminated from each other.

- The paper lists 63 references, many of which are not cited in the paper. I suppose they are cited in the appendix, but then I think the appendix should have a separate bibliography.

- Personally I find it preposterous and disrespectful of the reviewers' commitment to submit a 40 page document to a conference with a 10-page limit. As a result, I did not look at any of the supplementary material, and only read the work as a stand-alone paper. If you have a journal-length paper, do submit it to a journal, and not to a conference where reviewing time is necessarily limited. I did my best to not let this influence my assessment of the paper, but, if anything, the effect was certainly not positive.

**Questions:**

I don't need any clarifications at this point (I did not understand everything, see above, but I think this is a problem with the writing, so an explanation would not change my assessment of the paper).

---

> ### Author Response · Authors · 2024-11-20
> **Response to Reviewer GXDf [1/3]**
>
> Sincerely thank you for your time devoted to reviewing this paper and your constructive suggestions. Here are our detailed replies to your comments and questions.
>
> > **W1:** The problems appear a bit artificial to me. All explanations are based on the concept hierarchy in the CIFAR data. I miss a clear real-world motivation for the setting. For example, I would have thought of a problem as follows: Given a learned classifier from CIFAR, remove all images that have a certain characteristics (e.g., remove all images from copyrighted sources, etc.). This obviously has real-world implications. I can imagine that this can be formulated in the setting studied by the authors, but it is not clear how, because they always refer to existing classes (and there is no class for "images from copyrighted sources", I guess). Probably what they do is fine for an evaluation, but one should provide a clearer motivation, and then discuss why the restricted CIFAR setting can be used for evaluating the setting discussed in the motivation.
>
> **A1:** Thanks for your example! However, we would like to highlight that the research setting is **motivated by a general assumption perspective**. As introduced in the L42-52, the key problem we considered is that **the practical unlearning request may violate the taxonomy of the pre-training tasks**, which may generally exist in removing all images with certain characteristics (like IP conflicts/copyright issues [1], bias information [2], privacy/safety concern [3,4], **as long as it mismatches with the pre-training taxonomy**). **Such as you mentioned** removing all images from the copyrighted sources, the copyrighted image may not always be exactly the identified forgetting data or the one class of the pre-training task. When the images from the copyrighted sources only occupy a sub-set of a class, it is similar to **model mismatch forgetting** as illustrated in the left bottom of Figure 1; When we consider removing harmful images containing NSFW (Not Safe For Work) content (related to the concept of "nudity"), we are not only to forget the identified/reported data but all of the data belonging to the target concept, which similar to the **target/data mismatch forgetting** as illustrated in the right of Figure 1. Our Figure 1, using the CIFAR dataset, provides an overall illustration of controllable mismatched forgetting scenarios with the Venn diagrams.
>
> We believe using the **benchmark CIFAR datasets** (also commonly used in previous unlearning literature [2,5]) **serves an important purpose in exploring** the newly considered unlearning settings **under the controlled conditions**. With the officially provided class and superclass information, it can naturally provide flexible research basic to construct the unlearning request with **the possible scenarios of target concept covering more than identified forgetting data or only a sub-set of pre-training classification unit**, and help us to conduct an investigation on evaluating unlearning challenges under various scenarios. **We will highlight it and add more practical significance and correspondence to our research setting in the Section 3.1.**
>
> [1] On the Opportunities and Risks of Foundation Models. arXiv 2021.
>
> [2] Machine Unlearning: A Survey. ACM Survey 2023.
>
> [3] AI model disgorgement: Methods and choices. PNAS 2024.
>
> [4] Machine Unlearning in Generative AI: A Survey. ACM Survey 2024.
>
> [5] Model Sparsity Can Simplify Machine Unlearning. NeurIPS 2023.

---

> ### Author Response · Authors · 2024-11-20
> **Response to Reviewer GXDf [2/3]**
>
> > **W2:** I really had a hard time in understanding the < notation (this should be a curly <). Maybe this is for similar reasons as discussed above, because I was really looking for an intuitive understanding, not for an explanation that specifies in what way that affects the CIFAR data. Also terms like "false retaining set" and "true retaining set" I found hard to grasp, even though they are formally defined.
>
> **A2:** The **intuitive illustrations can refer to Figure 1 and Figure 2**. First, **the notation $\prec$** is defined at L73-74, which is **used to describe label domain relations instead of a way that affects the data**, e.g., $L_1\prec L_2$ indicates $L_1$ is the subclass domain of $L_2$. **It is instantiated** as class (represented by yellow area) and superclass (represented by green area) **in Figure 1** to **illustrate the mismatch** among target concept, forgetting data, and the model output. **To give an example in your mentioned** copyright sources removing, given the pre-trained CIFAR-100 model and one class identified forgetting data, imaging there are more than one class data has the copyright issue need to be removed, then target concept "image with copyright" can be regarded as a superclass of the identified forgetting data. To sum up, this notation can be generally understood as **an indicator of the smallest taxonomy unit of one label domain is smaller than that of another label domain.** As for the term of **"false/true retaining set"**, the intuitive illustration can refer to **the Venn diagrams in Figure 2**. If the target concept is larger than the **identified forgetting data** of "boy" and "girl",  e.g., we want to forget the concept of "people", those non-identified data of "man", "woman", and "baby" **are false retaining data** as indicated Figure 2, then **rest data** in CIFAR-100/CIFAR-20 which not exist in the Venn diagram are **true retaining data**.
>
>
> > **W3:** The proposed method has three phases, which are apparently applied sequentially. Even though there is a blended phase in the middle, I would expect that a somewhat tighter integration of these phases would be better.
>
> **A3:** We would like to point out that **the proposed method is a unified framework that can be characterized by Eq. (5)**. It is **presented as three phases** intensively as we would like **to clearly explain each part's functionality and rationality** for the previously identified challenge in mismatch forgetting. The implementation is achieved by assigning time-related parameters in Eq. (5), to realize target identification, separation and approximate unlearning.
>
> > **M1:** I don't think that computation is the main problem with "exact unlearning", it is more that you have to have access to all the training data, i.e., you cannot simply store the model and unlearn new data from it.
>
> **A4:** First, we never claim the "computation" is the "main problem" with "exact unlearning" and it is not the major issue related to our research work. Second, the similar statement of "intensive compuational cost" with approximate learning also existed in previous literature [1,2,3].
>
> [1] Model Sparsity Can Simplify Machine Unlearning. NeurIPS 2023.
>
> [2] SalUn: Empowering Machine Unlearning via Gradient-based Weight Saliency in Both Image Classification and Generation. ICLR 2024.
>
> [3] Machine Unlearning: A Survey. ACM Survey 2023.
>
> > **M2:** Table 1 is too small. Some of the subscripts cannot be discriminated from each other.
>
> **A5:** Thanks for the comments! We will adjust Table 1 to make the subscripts larger to be discriminated.

---

> > ### Comment · Reviewer_GXDf · 2024-11-25
> >
> > Thanks for your elaborate reply.
> >
> > I have been informed that you take issues with me not asking any questions. Why should I? Even though I did not understand every detail, the paper was clear enough for me to come to a conclusion.
> >
> > For me, this is primarily a minor advancement on a set of benchmark tasks, on which other people have worked on. I don't think that this is particularly exciting, and given the stiff competition in this conference, this is, in my opinion, not enough to justify acceptance.  So I did not need any clarification, and your response has not changed my opinion.
> >
> > I do see that my co-reviewers disagree, and are more on the accept side. I acknowledge that, but I also see that they bring up some of the same criticism that I had as well. For example, reviewer  DRij also criticisizes the unclear motivation, and the lack of real-world applicability. I think we essentially agree on the problems with this paper, but not on how strongly we should weight them. For me, these problems are apparently more significant than for my co-reviewers, but the problems are clear and apparent.
> >
> > In any case, I trust the area chair will find an appropriate conclusion, which, either way, will be fine with me, of course.

---

> ### Author Response · Authors · 2024-11-20
> **Response to Reviewer GXDf [3/3]**
>
> > **M3 & M4:** **M3** The paper lists 63 references, many of which are not cited in the paper. I suppose they are cited in the appendix, but then I think the appendix should have a separate bibliography. **M4** Personally I find it preposterous and disrespectful of the reviewers' commitment to submit a 40 page document to a conference with a 10-page limit. As a result, I did not look at any of the supplementary material, and only read the work as a stand-alone paper. If you have a journal-length paper, do submit it to a journal, and not to a conference where reviewing time is necessarily limited. I did my best to not let this influence my assessment of the paper, but, if anything, the effect was certainly not positive.
>
> **A6:** Thanks for the comments! However, given the depth and complexity of the research setting and problem we studied, providing comprehensive supplementary material in the appendix is a common practice in the research community [1,2,3,4]. We will consider to streamlining it in the final version if there is widespread demand.
>
> [1] SWE-bench: Can Language Models Resolve Real-World GitHub Issues?. ICLR 2024 oral. [51 pages]
>
> [2] Finetuning Text-to-Image Diffusion Models for Fairness. ICLR 2024 oral. [43 pages]
>
> [3] Mastering Memory Tasks with World Models. ICLR 2024 oral. [37 pages]
>
> [4] Pareto Invariant Risk Minimization: Towards Mitigating the Optimization Dilemma in OOD Generalization. ICLR 2023. [50 pages]

---

> ### Author Response · Authors · 2024-11-29
>
> Dear Reviewer,
>
> Thanks for your elaborate response. However, we respectfully disagree with your assessments of our work. **Regarding the contribution, we believe our work goes substantially beyond "minor" improvements, and is recognized by the other three reviewers**:
>
> - introduce new settings of unlearning that extend the conventional all matched forgetting (zoDC), which is a novel problem (DRij) that makes sense and contributes to the community of machine unlearning (GTmm).
> - propose a novel and general framework TARF (zoDC), providing an interesting solution (DRij) for unlearning in three typical mismatch scenarios, and has the potential to improve its flexibility and practicality (GTmm).
> - systematically reveals the challenges of restrictive unlearning with mismatched label domains (GTmm), gives new insights, and demonstrates the promising results of the method with sufficient (zoDC) and extensive (DRij) experiments.
>
> **Regarding the motivation, we believe it is clear on the consideration that unlearning request may have different taxonomy from the original tasks**, and we don't think your concern is the same as Reviewer DRij suggested "further clarify how unlearning can enhance trustworthiness", in which the latter is a clarification requirement on "trustworthiness", while not affecting the rationality of our introduced label domain mismatch.
>
> **Regarding real-world applicability, it is recognized by Reviewer GTmm and zoDC for extending the conventional unlearning paradigm and improving its practicality**, and we don't think your concern is the same as Reviewer DRij suggested "further discuss unlearning with limited class information", as the latter is for discussion of our method in tackling a more challenging scenario in target mismatch forgetting (one of the mismatched scenarios), instead of the criticism on the practical significance of our introduced problem.
>
> While we understand your position of holding your opinion on the impact assessment, we believe the support from other reviewers reflects the substantial value of our contributions.
>
> Best regards,
>
> Authors of Submission 4171

---

### Official Review · Reviewer_zoDC · 2024-10-31

**Soundness:** 3
**Presentation:** 2
**Contribution:** 3
**Rating:** 6
**Confidence:** 4

**Summary:**

Machine unlearning aims to enable a model to forget specific data without retraining, making it behave as if the data had never been used during training. This paper introduces a broader machine unlearning framework that extends the conventional all-matched forgetting scenarios. The proposed method performs target identification to recognize false remaining data. Then, the proposed method conducts forgetting on given forgetting data and identified false remaining data, and conducts retaining on identified true remaining data. Finally, extensive experiments validate the effectiveness of the proposed method.

**Strengths:**

1.This paper extends conventional all matched forgetting and proposes a general framework for machine unlearning.

2.The experiments are sufficient to demonstrate the effectiveness of the proposed method.

**Weaknesses:**

1.In the model mismatch scenario, the performance of the proposed method is significantly lower than that of existing methods that do not consider model mismatch.

2.Compared to other approximate unlearning methods, the proposed method has a higher time cost.

3.The organization of the paper needs revision to improve its readability. Additionally, is such an extensive appendix necessary for the completeness of the paper? The current version seems somewhat unfriendly to readers.

**Questions:**

1.In the target identification phase, the authors use the prediction loss before and after forgetting to reflect representation similarity, thereby selecting false remaining data. This operation is based on empirical observations from experiments. How should the parameter \beta be controlled during implementation to balance the amount of data being forgotten?

2.Compared to existing methods, the proposed approach adds a target identification phase. Does this phase cause performance degradation in the all-matched scenario? In the experiments, the proposed method shows less performance advantage in the all-matched scenario.

3.In Figure 3, what does "class-aligned data" refer to?

4.In the evaluation metrics, on which dataset is TA applied?

5.In Table 1, for the retrained method, why are the values of the metric UA not equal to 0 in the model mismatch scenario?

6.In the model mismatch scenario, the proposed method performs significantly worse on UA compared to existing methods such as GA, IU, BS, and SalUn. Please analyze the reasons behind this phenomenon. Furthermore, existing methods do not consider the model mismatch scenario, while the proposed method provides a specific solution for this scenario.

7.Since a smaller UA value is better, a downward arrow should be annotated.

---

> ### Author Response · Authors · 2024-11-20
> **Response to Reviewer zoDC [1/2]**
>
> Sincerely thank you for your supportive review and your constructive suggestions. Here are our detailed replies to your comments and questions.
>
> > **W1 & Q5 & Q6 & Q7 :** **W1** In the model mismatch scenario, the performance of the proposed method is significantly lower than that of existing methods that do not consider model mismatch. **Q5** In Table 1, for the retrained method, why are the values of the metric UA not equal to 0 in the model mismatch scenario? **Q6** In the model mismatch scenario, the proposed method performs significantly worse on UA compared to existing methods such as GA, IU, BS, and SalUn. Please analyze the reasons behind this phenomenon. Furthermore, existing methods do not consider the model mismatch scenario, while the proposed method provides a specific solution for this scenario. **Q7** Since a smaller UA value is better, a downward arrow should be annotated.
>
> **A1:** We should point out that **the performance of TARF that is generally closest to that of Retrained (Ref.) in model mismatch scenario and is better than other methods, according to the "Gap$\downarrow$" in Table 1**. Besides, the **unlearning accuracy (UA) is not the smaller the better** as it is evaluated with superclass label, **but the closer to the UA of Retrained (Ref.) the better**. In this spirit, **we also do not expect the UA in the model mismatch scenario to approach 0 as forgetting data** ("boy" and "girl") **have the same superclass label (i.e., "people") with a part of retaining data** (e.g., "man", "woman", and "baby"). Regarding the definition of unlearning target, TARF achieves the closest approximation on UA with the Retrained reference. We will add a detailed explanation in our revised version for all the evaluation metrics, and a special explanation for the model mismatch scenario in Section 4.2 to make it clearer.
>
> > **W2:** Compared to other approximate unlearning methods, the proposed method has a higher time cost.
>
> **A2:** We agree with the reviewer that TARF may require more time in unlearning compared with some methods like GA. However, we would like to clarify that **the metric "TIME$\downarrow$" originally means to avoid some methods that consume too much time compared with that of Retrained (Ref.)**. From this perspective, these current methods and TARF actually fall in the acceptable time range, and the efficiency gap between existing explorations in that range is indeed not a bottleneck based on Table 1. We appreciate the reviewer's concern about this point and will add more discussion in the revision for clarity.
>
>
> > **W3:** The organization of the paper needs revision to improve its readability. Additionally, is such an extensive appendix necessary for the completeness of the paper? The current version seems somewhat unfriendly to readers.
>
> **A3:** Thanks for your constructive comments! **We will add some introductory content at the beginning of each Section to improve the organization.** The appendix are some reference information and further analytical results about our work, which are auxiliary information about main results and can enhance the transparency and reproducibility of our research. **We will also add the introduction part in appendix and improve its table of contents** to help readers better understand the content and can directly narrow down to the subsections of their interest.
>
> > **Q1:** In the target identification phase, the authors use the prediction loss before and after forgetting to reflect representation similarity, thereby selecting false remaining data. This operation is based on empirical observations from experiments. How should the parameter \beta be controlled during implementation to balance the amount of data being forgotten?
>
> **A4:** Thanks for the question! In the implementation, **$\beta$ value is estimated with the ranked loss difference of each class and the prior information** about the target concept to control the forgetting strength in unlearning. For the task feasibility, we will generally **assume the amount of false remaining data or classes is known** at our target/data mismatch forgetting, **following a similar setup in learning from label noise** where the noise rate can be estimated and utilized as prior information [1,2]. We will add the implementation details in Appendix E and cross-refer it in the main text to make it clearer.
>
> [1] Co-teaching: Robust Training of Deep Neural Networks with Extremely Noisy Labels. NeurIPS 2018.
>
> [2] Machine Learning from Weak Supervision. MIT Press 2022.

---

> ### Author Response · Authors · 2024-11-20
> **Response to Reviewer zoDC [2/2]**
>
> > **Q2:** Compared to existing methods, the proposed approach adds a target identification phase. Does this phase cause performance degradation in the all-matched scenario? In the experiments, the proposed method shows less performance advantage in the all-matched scenario.
>
> **A5:** Thanks for the thoughtful question! **From the methodology view**, there is no need for the target identification part to identify extra forgetting data in the all-matched scenario as the target concept matches the forgetting data, then our TARF degenerates into a general framework using the given forgetting data to forget and the rest to retain. **From the experimental view**, we would like to clarify that our performance on the all-matched scenario is comparable to the existing best counterpart like SCRUB regarding the "Gap$\downarrow$" in Table 1. **It can be found that the overall performance of the unlearned models have already closely approximated the Retrained reference.** Furthermore, since our TARF is a general framework for unlearning, we can also adopt the KL divergence loss with the original model as designed in SCRUB to further improve the performance, for which we present the following comparison.
>
> **Table 1. Performance comparison in all matched scenario.**
>
> |                  | CIFAR-10 |       |       |        |                 | CIFAR-100 |       |       |        |                 |
> | ---------------- | -------- | ----- | ----- | ------ | --------------- | --------- | ----- | ----- | ------ | --------------- |
> | All matched      | UA       | RA    | TA    | MIA    | GAP$\downarrow$ | UA        | RA    | TA    | MIA    | GAP$\downarrow$ |
> | Retrained (Ref.) | 0.00     | 99.51 | 94.69 | 100.00 | -               | 0.00      | 97.85 | 76.03 | 100.00 | -               |
> | FT               | 1.07     | 98.62 | 92.36 | 100.00 | 1.07            | 0.67      | 96.32 | 72.34 | 100.00 | 1.47            |
> | SCRUB            | 0.00     | 99.94 | 91.00 | 100.00 | 1.03            | 0.00      | 99.98 | 76.75 | 100.00 | 0.71            |
> | TARF (with CE)   | 0.00     | 98.23 | 91.95 | 100.00 | 1.01            | 0.00      | 96.90 | 72.53 | 100.00 | 1.11            |
> | TARF (with KL)   | 0.00     | 98.81 | 93.33 | 100.00 | **0.52**        | 0.00      | 96.95 | 74.98 | 100.00 | **0.49**        |
>
> We will add the discussion and experimental results in our revised version.
>
> > **Q3:** In Figure 3, what does "class-aligned data" refer to?
>
> **A6:** The "class-aligned data" in Figure 3 refers to **other data under the same label in the original task as forgetting data**, which can correspond to "affected retaining data" (as illustrated in the left Venn diagram of Figure 2) to the model mismatch forgetting scenario. For example, given the model trained by the superclass, the class-aligned data can be "man","woman", and "baby" while the forgetting data are "boy" and "girl" under the superclass label "people". Since in Figure 3, we present the representation similarity regardless of the specific unlearning scenario, we utilize the "class-aligned data" instead of "affected retaining data".
>
> > **Q4:** In the evaluation metrics, on which dataset is TA applied?
>
> **A7:** TA is measured using **the test dataset without the data belonging to the target concept**. The test datasets are the whole test datasets of those benchmark datasets like CIFAR-10/100, and we will adjust them in each specific mismatched forgetting scenario to remove the data belonging to the target concept to be forgotten, then use the rest for evaluation.

---

> ### Author Response · Authors · 2024-11-22
>
> Dear Reviewer zoDC,
>
> Thank you for your great efforts in reviewing our work and for your valuable comments.
>
> In the rebuttal, we have tried our best to address the concerns, and provided detailed responses as well as the revision summarization. Would you mind checking our responses and confirming if there is any unclear point so that we should/could further clarify?
>
> Best regards,
>
> Authors of Submission 4171

---

> ### Author Response · Authors · 2024-11-24
> **Looking forward Further Reply**
>
> Dear Reviewer zoDC,
>
> Thank you very much for your time and insightful comments. We have provided detailed responses and conducted the revision (summarized in General Response) following your valuable suggestions.
>
> As the discussion phase is approaching to end, would you mind confirming if there is any additional questions or concerns regarding our response? We will do our best to address them promptly. Additionally, if you find our responses helpful, we would also greatly appreciate it if you could acknowledge the efforts we have made.
>
> Best regards,
>
> Authors of Submission 4171

---

> ### Author Response · Authors · 2024-11-25
> **Would you mind checking our responses and confirming whether you have any further questions?**
>
> Dear Reviewer zoDC,
>
> Thanks again for your time in reviewing and insightful comments. As the discussion phase is approaching to end, we hope to further discuss with you if there is any additional questions regarding our detailed response, and will do our best to address them promptly following your constructive suggestions. if you find our responses helpful, we would also sincerely appreciate it if you could acknowledge the efforts we have made.
>
> Best regards,
>
> Authors of Submission 4171

---

> ### Author Response · Authors · 2024-11-27
> **Would you mind checking our responses and confirming whether you have any further questions?**
>
> Dear Reviewer zoDC,
>
> We sincerely thank you for your efforts in reviewing! We have provided detailed responses and conducted the revision (summarized in General Response) following your valuable suggestions. Kindly please let us know if there is any unclear point so that we should/could further clarify. if you find our responses helpful, we would also greatly appreciate it if you could acknowledge the efforts we have made.
>
> Best regards,
>
> Authors of Submission 4171

---

> ### Author Response · Authors · 2024-11-29
>
> Dear Reviewer,
>
> Thank you once again for your constructive feedback and **positive recognition of our work, e.g., extended problem setting, the general framework for unlearning, sufficient experiments, and effective results**. We appreciate the time you have taken to review our paper, and your insights have been instrumental in improving our work.
>
> In response to your questions, we have **clarified the evaluation details especially for the model mismatch scenario**, and apologize for any confusion it may induce to understand our results (related to most of your questions). Due to the label domain mismatch, the **unlearning accuracy (e.g., UA) is not expected to be equal to 0 since it is evaluated with superclass label** (e.g., unlearning a subset of "people" while retaining another part). We have also provided a detailed section to explain all the evaluation metrics in different scenarios in Appendix C.2. We hope the explanation can clarify that **our TARF is more effective in approximating the Retrained Reference in the model mismatch forgetting compared with other methods**, regarding the metric of "GAP".
>
> Given the other constructive suggestions, **we sincerely appreciate your thoughtful comments** on the performance of conventional all-matched scenarios, for which we provide a comparison of both methodology and empirical view (in Appendix F.3) to show the satisfactory performance of all the methods and compatablity of our general framework to achieve superior performance; on the computational cost, for which we discuss the reasonable comparison with Retrained Reference for achieving more challenging unlearning requests in Appendix E.2; on the $\beta$ controlling, for which we explain the threshold estimation on the ranking values in Appendix E; **those help us to make the details to be clearer and improve the organization via more structural revision**.
>
> Given the improvements we’ve made, we would greatly appreciate if you could consider revising your score, considering our contributions and refinements incorporated into the revised version. Should you have any remaining concerns or require additional clarification, we are more than happy to discuss. We remain fully committed to addressing any additional questions or concerns you may have, during the extended discussion period. Thank you again for your time and valuable feedback!
>
> Best regards,
>
> Authors of Submission 4171

---

### Official Review · Reviewer_DRij · 2024-11-03

**Soundness:** 2
**Presentation:** 3
**Contribution:** 2
**Rating:** 6
**Confidence:** 3

**Summary:**

Machine unlearning is an emerging field focused on enabling models to "forget" specific knowledge derived from training data. While existing research demonstrates that removing knowledge of certain classes can be achieved through techniques like gradient ascent, these
approaches typically assume a perfect alignment between class labels and target concepts,  limiting their applicability to broader cases. This work expands on this by addressing misalignments in label, model, and data domains by proposing TARget-aware Forgetting  (TARF), a framework that selectively forgets target concepts by combining gradient ascent and descent methods, ensuring that unaffected data remains preserved. Experiments highlight TARF's effectiveness in achieving targeted forgetting.

**Strengths:**

1.The experiments are extensive and the results are promising.
2.The research question is novel, and the proposed solution is interesting.
3. The overall presentation is easy to follow.

**Weaknesses:**

While TARF shows promise, the experiments are comprehensive and thorough, this study faces several limitations:
- Study Motivation: The motivation behind this work requires further clarity and discussion. In Section 3.1, there is a mention of "increasing concerns about trustworthiness," but this lacks detail. Expanding on how unlearning can enhance trustworthiness would strengthen this section. Additionally, existing unlearning research is primarily motivated by a desire to protect data owners' right to withdraw from the learning process. Please discuss how this work differs in motivation and approach from previous research.
- Experimental Design: In the experimental section, it is mentioned that superclass information from the CIFAR-10 dataset was used. However, CIFAR-10 does not natively include superclass information. Please clarify if artificial labels were applied to CIFAR-10. Additionally, the main results heavily rely on the appendix for image generation outcomes. Consider reorganizing this part to make key findings more accessible.
- Limitations in Real-World Application: Based on Equation 5 and Phase 1, TARF demonstrates effectiveness primarily when complete class information is available. If class information (such as subclasses or super classes) is unavailable or implicit, TARF may be  limited in real-world applications. Consider discussing this limitation in more depth.

**Questions:**

1.What kind of class (superclass) information have been introduced on the CIFAR-10 datasets for supporting the experiments?
 2. Whether the class information should all be available for the proposed TARF?

---

> ### Author Response · Authors · 2024-11-20
> **Response to Reviewer DRij [1/2]**
>
> Sincerely thank you for your supportive review and your constructive suggestions. Here are our detailed replies to your comments and questions.
>
> > **W1:** Study Motivation: The motivation behind this work requires further clarity and discussion. In Section 3.1, there is a mention of "increasing concerns about trustworthiness," but this lacks detail. Expanding on how unlearning can enhance trustworthiness would strengthen this section. Additionally, existing unlearning research is primarily motivated by a desire to protect data owners' right to withdraw from the learning process. Please discuss how this work differs in motivation and approach from previous research.
>
> **A1:**  Thanks for the comments! Machine unlearning was originally introduced in response to data regulations [1,2], primarily driven by the need to safeguard data owners' rights to withdraw their data from the learning process. However, **given its technical nature** of mitigating the data influence from a trained model [3], **unlearning is actually now given broader significance in the context of trustworthy AI** [4,5]. **For example, 1)** unlearning can be applied to mitigate the model bias or unfairness by erasing inaccurate or biased data in training [6], similar to our model mismatch scenario to forget a sub-set of data; **2)** unlearning can support model safety by erasing the vulnerable or sensitive information [5,7] (e.g., outlier or human-related data), where the target mismatch scenario may exist; **3)** unlearning can erase harmful concept (e.g., "nudity" in NSFW content) in image generation to ensure proper usage of generative AI [8,9], which is similar to data mismatch forgetting as sometimes we can only identified limited forgetting cases.
>
> **We appreciate the reviewer's suggestion and have expanded the discussion to clarify this claim about trustworthiness in the revision. We also emphasize that our motivation is compatible with both the original data protection and trustworthiness, which considers that the unlearning request may have a different taxonomy from the pre-trained tasks**.
>
> [1] Towards making systems forget with machine unlearning. S&P 2015.
>
> [2] The european union general data protection regulation: what it is and what it means. Information & Communications Technology Law 2019.
>
> [3] Machine Unlearning: A Survey. ACM Survey 2023.
>
> [4] Supporting Trustworthy AI Through Machine Unlearning. Electrical Engineering 2023.
>
> [5] AI model disgorgement: Methods and choices. PNAS 2024.
>
> [6] Machine Unlearning Challenge. Google research 2023.
>
> [7] The WMDP Benchmark: Measuring and Reducing Malicious Use With Unlearning. arXiv 2024.
>
> [8] Machine Unlearning in Generative AI: A Survey. ACM Survey 2024.
>
> [9] Erasing Concepts from Diffusion Models. ICCV 2023.

---

> ### Author Response · Authors · 2024-11-20
> **Response to Reviewer DRij [2/2]**
>
> > **W2 & Q1:** **W2** Experimental Design: In the experimental section, it is mentioned that superclass information from the CIFAR-10 dataset was used. However, CIFAR-10 does not natively include superclass information. Please clarify if artificial labels were applied to CIFAR-10. Additionally, the main results heavily rely on the appendix for image generation outcomes. Consider reorganizing this part to make key findings more accessible. **Q1** What kind of class (superclass) information have been introduced on the CIFAR-10 datasets for supporting the experiments?
>
> **A2:** Thanks for the suggestion! To support the experiment, we adopt artificial labels to the CIFAR-10 dataset for the superclass information. The **detailed label information is provided in Table 11 (Table 13 at revised version) and Appendix D.4.** We are sorry to make this part hard to find and will **link Appendix D.4 in the main text obviously to provide a clear clarification**. As for the image generation results, we will follow the reviewer's advice to present more key findings in the main text, but leave more visualization in the appendix due to the space limitation.
>
> > **W3 & Q2:**  **W3** Limitations in Real-World Application: Based on Equation 5 and Phase 1, TARF demonstrates effectiveness primarily when complete class information is available. If class information (such as subclasses or super classes) is unavailable or implicit, TARF may be limited in real-world applications. Consider discussing this limitation in more depth. **Q2** Whether the class information should all be available for the proposed TARF?
>
> **A3:** Thanks for your thoughtful comments! **In our experimental setup, the class information should be available in TARF**. Regarding the unavailable or implicit class information, we will add more discussion about the potential limitations. Specifically, **first**, if the class (i.e., the subclasses w.r.t. target concept) is not available, TARF may also utilize the model prediction to obtain the pseudo labels to conduct the task; **Second**, if the extra forgetting target beyond the identified data is not restricted as classes, it may require that given forgetting data can well represent the target concept (e.g., the false retaining data should be easier affected than the other retaining data). We acknowledge that both scenarios would lead to a larger performance gap with Retrained reference, as it is a generally more challenging scenario affecting the task achievability to all of the approximate unlearning methods. **We believe it is worth future effort to explore and will add the discussion in our updated version.**

---

> ### Author Response · Authors · 2024-11-22
>
> Dear Reviewer DRij,
>
> Thank you for your great efforts in reviewing our work and for your valuable comments.
>
> In the rebuttal, we have tried our best to address the concerns, and provided detailed responses as well as the revision summarization. Would you mind checking our responses and confirming if there is any unclear point so that we should/could further clarify?
>
> Best regards,
>
> Authors of Submission 4171

---

> ### Author Response · Authors · 2024-11-24
> **Looking forward Further Reply**
>
> Dear Reviewer DRij,
>
> Thank you very much for your time and insightful comments. We have provided detailed responses and conducted the revision (summarized in General Response) following your valuable suggestions.
>
> As the discussion phase is approaching to end, would you mind confirming if there is any additional questions or concerns regarding our response? We will do our best to address them promptly. Additionally, if you find our responses helpful, we would also greatly appreciate it if you could acknowledge the efforts we have made.
>
> Best regards,
>
> Authors of Submission 4171

---

> ### Author Response · Authors · 2024-11-25
> **Would you mind checking our responses and confirming whether you have any further questions?**
>
> Dear Reviewer DRij,
>
> Thanks again for your time in reviewing and insightful comments. As the discussion phase is approaching to end, we hope to further discuss with you if there is any additional questions regarding our detailed response, and will do our best to address them promptly following your constructive suggestions. if you find our responses helpful, we would also sincerely appreciate it if you could acknowledge the efforts we have made.
>
> Best regards,
>
> Authors of Submission 4171

---

> > ### Comment · Reviewer_DRij · 2024-11-27
> >
> > Thank you for your thoughtful response. I appreciate the effort you put into addressing my concerns. After careful consideration, I will keep my original score.

---

> > > ### Author Response · Authors · 2024-11-27
> > > **Thank you for the acknowledgment and for keeping positive support in the feedback!**
> > >
> > > Dear Reviewer DRij,
> > >
> > > Thank you for your acknowledgment and for keeping positive on our work! Following your constructive comments, we have incorporated all your suggestions in the revision. Please don't hesitate to let us know if there are any remaining concerns, we will do our best to address them.
> > >
> > > Best regards,
> > >
> > > Authors of Submission 4171

---

> ### Author Response · Authors · 2024-11-29
>
> Dear Reviewer,
>
> Thank you once again for your constructive feedback and **positive recognition of our work, e.g., novel problem, interesting solution, extensive experiments, and easy-to-follow presentation**. We appreciate the time you have taken to review our paper, and your insights have been instrumental in improving our work.
>
> In response to your suggestions, we have **extended the discussion on how unlearning can enhance trustworthiness**, which strengthens the practical significance of our introduced problem setting. We believe **our clear motivation** will not affect its compatibility with previous pursuit on data protection, since it **is general on the assumption perspective** of machine unlearning (which is elaborated in L43-L52 in the introduction and detailed discussed in Section 3.1 and Appendix D.5), **i.e., considering the unlearning tasks may have different taxonomy from the original pre-training task.** Thus, we decouple the target concept from the class label, to explore the **feasibility of forgetting that is not limited to the original task class (i.e., all matched scenario)**. It is reasonable and practical when the unlearning request is beyond the original classification tasks like concerns on copyright, bias, or safety issues as elaborated in our submission and responses.
>
> Regarding the **experimental setups and a more challenging scenario on limited class information**, we have clarified the details of superclasses information in Appendix D.4 and provided the discussion on the requirement of class information with the unlearning feasibility in Appendix E.3. **We appreciate your thoughtful comments** and your pursuit of more challenging settings, **which we believe indicates a recognition of considering more flexible setting for machine unlearning and the potential impact of our introduced label domain mismatch.** We sincerely hope that our step towards the practical and flexible achievement of machine unlearning brings new insights into the research problem and the start can be recognized.
>
> Given the improvements we’ve made, we would greatly appreciate if you could consider revising your score, considering our contributions and refinements incorporated into the revised version. Should you have any remaining concerns or require additional clarification, we are more than happy to discuss. We remain fully committed to addressing any additional questions or concerns you may have, during the extended discussion period. Thank you again for your time and valuable feedback!
>
> Best regards,
>
> Authors of Submission 4171

---

### Author Response · Authors · 2024-11-20
**General Response to All Reviewers**

We appreciate all the reviewers for their thoughtful comments and suggestions on our paper.

**Recapping our work**, we introduce new settings in machine unlearning by decoupling the class labels and the target concept, considering the practical unlearning request may violate the taxonomy of original tasks. With the instantiated unlearning tasks in controllable benchmark datasets, we systematically explore and reveal the challenges in target mismatch, model mismatch, and data mismatch forgetting scenarios, providing analytics from the view of representation gravity in forgetting dynamics on the achievability in these mismatch scenarios. Based on that, we propose a general framework, TARF, which consists of annealed forgetting and target-aware retaining to realize the restrictive approximate unlearning. We present comprehensive experimental results to demonstrate its effectiveness and characteristics.

**We are very glad to see that the reviewers find** our **introducing new settings** to extend the conventional all matched machine unlearning, where the **research question is novel**, the developed **viewpoint is nice and important**, and would **contributes to the community** of machine unlearning (Reviewer DRij, zoDC, GXDf, GTmm); The proposed TARF is **a general framework** for machine unlearning with mismatch, which is **novel**, **interesting**, and has the potential to **improve the flexibility and practicality** (Reviewer DRij, zoDC, GTmm); The experiments are **extensive, comprehensive, and sufficient** to demonstrate the effectiveness of the method,  and **results are promising**, which give **a new insight into machine unlearning** with the representation gravity in forgetting dynamics (Reviewer DRij, zoDC, GTmm); We are also pleased that the reviewers find our overall **presentation is good** and **easy to follow** (Reviewer DRij, GTmm).

We have tried our best to address the reviewers' comments and concerns in **individual responses to each reviewer** with further explanation and justification. Those valuable comments allowed us to improve our draft and the contents added in the **revised version** are summarized below:

**From Reviewer DRij**

- Expand the "untrustworthiness" concerned in unlearning. (in Section 3.1 and Appendix D.5)
- Clarify the superclass in CIFAR-10. (in Section 2 and Appendix D.4)
- Discuss TARF with limited class information for identification. (in Appendix E.3)

**From Reviewer zoDC**

- Detail the explanation of evaluation metrics in different scenarios. (in Section 4.1 and Appendix C.2)
- Clarify the details of $\beta$ controlling. (in Appendix E)
- Add the discussion and comparison in the all matched scenario. (in Appendix F.3)
- Discuss the computational cost of TARF. (in Appendix E.2)
- Add more introductory content and improve the table of contents of the appendix.

**From Reviewer GXDf**

- Highlight the practical significance and correspondence of the research settings (in Introduction and Section 3.1)
- Explicitly link the specific term with intuitive examples. (with Figures 1 and 2)
- Simplify and scale Table 1. (in Section 2)

**From Reviewer GTmm**

- Add the introduction of "gravity" in advance. (in Section 3.2)
- Detail the explanation of evaluation metrics in different scenarios. (in Section 4.1 and Appendix C.2)
- Discuss the commonalities of mismatch scenarios and how to construct a unified framework. (in Appendix D.6)

**We appreciate your valuable comments and time!** We have tried our best to address your concerns and revised the paper following the suggestions. **Would you mind checking it and confirming if there are any unclear parts?** We look forward to continuing the valuable discussion and your constructive feedback during the discussion period!

---

### Meta-Review · Area_Chair_QBCw · 2024-12-19

**Metareview:**

This paper present a machine unlearning method adapted in domain mismatch context. It proposes a general framework called TARget-aware Forgetting (TARF) , which decouples the class label and the target concept and investigates three problems beyond conventional all-matched forgetting. Some experiments on various datasets are proposed to demonstrate the effectiveness of the approach.

Strengths:
- an interesting research questions,
- a general framework for machine unlearning,
- good experimental study,
- promising results.

Weaknesses
- the motivation of the work should be better justified ,
- the problem may appear artificial,
- some experimental settings were not convincing,
- the method has a high running time cost,
- the method shows some limitations for real applications,
- the paper presentation could be improved, notably with respect to the clarity of the evaluation metrics.

During the rebuttal, authors have provided a long and detailed answers some clarifications and additional results. Reviewers have raised some divergences on the quality of their answers.
Reviewers have appreciated the effort done by the authors during the rebuttal and the interesting contribution proposed in the paper.
During the discussion, it appears that a strong majority of the reviewers aligns on the fact that the paper needs significant improvements in terms of motivation (clarity of the motivation, novelty of the work), paper organization, experimental details and the limited real-world capabilities.

Based on this evaluation I propose then rejection.
Nevertheless, this paper has a potential and I encourage the authors to improve their submission for other venues.

**Additional Comments On Reviewer Discussion:**

There was a divergence between opinions and evaluations during the evaluation of the paper.
Reviewer GXDf was not open to discussion and to take into account authors' feedback. Authors complained on his attitude. As a consequence, I noticed his reservations, but I did not take his point of view to propose a decision on the paper acceptance.

Reviewer DRij appreciated the authors' effort but mention that this paper has still significant  weaknesses: (i) on the study Motivation: which still lacks clarity, particularly regarding its novelty compared to prior work and its impact on trustworthiness; (ii) on Real-World Applicability, the framework's effectiveness appears limited to settings where complete class information is available, which reduces its practical significance.
Reviewer zoDC still believes after rebuttal that this paper needs significant improvements in motivation, organization, and experimental details.
Reviewer GTmm was very positives and mentioned that his concerns were addressed, but did not support more the paper.


Overall, there is a significant majority for saying that the paper needs improvement on important weaknesses, so I propose rejection.

---

### Decision · Program_Chairs · 2025-01-22

Reject